# ARTICLES

# Comparative roadmaps of reprogramming and oncogenic transformation identify Bcl11b and Atoh8 as broad regulators of cellular plasticity

A. Huyghe [1,14 ✉], G. Furlan[1,13,14], J. Schroeder[2], E. Cascales[1], A. Trajkova[1], M. Ruel[1], F. Stüder [3], M. Larcombe[2], Y. Bo Yang Sun[2], F. Mugnier[1], L. De Matteo[1], A. Baygin[1], J. Wang[4], Y. Yu[4], N. Rama [5], B. Gibert [5], J. Kielbassa[6], L. Tonon[6], P. Wajda[1], N. Gadot[7], M. Brevet [8], M. Siouda[9], P. Mulligan[9,10], R. Dante[5], P. Liu [4], H. Gronemeyer [11], M. Mendoza-Parra[3], J. M. Polo [2,12] and F. Lavial [1 ✉]

Coordinated changes of cellular plasticity and identity are critical for pluripotent reprogramming and oncogenic transformation. However, the sequences of events that orchestrate these intermingled modifications have never been comparatively dissected. Here, we deconvolute the cellular trajectories of reprogramming (via Oct4/Sox2/Klf4/c-Myc) and transformation (via Ras/c-Myc) at the single-cell resolution and reveal how the two processes intersect before they bifurcate. This approach led us to identify the transcription factor Bcl11b as a broad-range regulator of cell fate changes, as well as a pertinent marker to capture early cellular intermediates that emerge simultaneously during reprogramming and transformation. Multiomics characterization of these intermediates unveiled a c-Myc/Atoh8/Sfrp1 regulatory axis that constrains reprogramming, transformation and transdifferentiation. Mechanistically, we found that Atoh8 restrains cellular plasticity, independent of cellular identity, by binding a specific enhancer network. This study provides insights into the partitioned control of cellular plasticity and identity for both regenerative and cancer biology.

During development, cells progressively differentiate with phenotypically distinct fates. These cellular identities, established by cell type-specific gene expression programs, are sustained over cell divisions throughout an organism's lifespan. However, this view of irreversible identity has been challenged by the discovery that somatic cells display a certain degree of plasticity in numerous contexts, including pluripotent reprogramming (hereafter called reprogramming) and oncogenic transformation (hereafter called transformation)[1,2]. Here, cellular plasticity is defined as the capability of cells to change identity outside normal development and tissue homeostasis[3].

During reprogramming, the transcription factors (TFs) Oct4, Sox2, Klf4 and c-Myc (OSKM) trigger widespread reconfiguration of chromatin and TF occupancy, which orchestrates a gradual gain of cellular plasticity and a concomitant loss of cellular identity in mouse embryonic fibroblasts (MEFs)[4]. Activation of the pluripotent network occurs later on, leading to the generation of induced pluripotent stem cells (iPS cells)[5–10]. Despite the description of reprogramming roadmaps

of diverse cell types[6,9,11–15], the molecular mechanisms coordinating the stepwise gain of plasticity and loss of identity remain largely unknown, yet they are critical for acquiring pluripotency.

Transformation shares features with reprogramming: both processes are constrained by oncogenic barriers and subjected to significant latencies[16–20]. Moreover, premature termination of reprogramming facilitates cancer development[21,22]. Gain of plasticity and loss of identity are also critical in various transformation contexts[1,20,23]. Cancer formation frequently relies on the activation of developmental programs that increase cellular plasticity, thus fuelling tumour heterogeneity[1,24]. Recent findings point to a crucial role for the oncogenic variant of the K-ras gene, which harbors a substitution of glycine for aspartic acid at codon 12 (K-ras$^{G12D}$), in triggering such changes. When combined with c-Myc exogenous expression and p53 depletion, K-ras$^{G12D}$ drives changes of cellular plasticity early during MEF transformation[18,25,26]. In the lung and pancreas, K-ras$^{G12D}$ alters the identity of specific cell types and increases their plasticity, fostering early tumorigenesis[27–29].

[1]Cellular Reprogramming, Stem Cells and Oncogenesis Laboratory, Equipe Labellisée la Ligue Contre le Cancer, LabEx Dev2Can, Université de Lyon, Université Claude Bernard Lyon 1, INSERM 1052, CNRS 5286, Centre Léon Bérard, Centre de Recherche en Cancérologie de Lyon, Lyon, France. [2]Department of Anatomy and Developmental Biology, Monash University, Melbourne, Clayton, Australia. [3]Génomique Métabolique, Genoscope, Institut François Jacob, CEA, CNRS, Université d'Évry, Université Paris-Saclay, Évry, France. [4]Wellcome Trust Sanger Institute, Cambridge, UK. [5]Apoptosis, Cancer and Development Laboratory, Université de Lyon, Université Claude Bernard Lyon 1, INSERM 1052, CNRS 5286, Centre Léon Bérard, Cancer Research Center of Lyon, Lyon, France. [6]Gilles Thomas Bioinformatics Platform, Centre Léon Bérard, Cancer Research Center of Lyon, Lyon, France. [7]Research Pathology Platform, Department of Translational Research and Innovation, Centre Léon Bérard, Lyon, France. [8]Department of Pathology, HCL Cancer Institute and Université Claude Bernard Lyon 1, Lyon, France. [9]Epigenetics and cancer Laboratory - Lyon University, Université Claude Bernard Lyon 1, INSERM 1052, CNRS 5286, Centre Léon Bérard, Cancer Research Center of Lyon, Lyon, France. [10]Institut NeuroMyoGene, Division PGNM, Universite Claude Bernard Lyon 1, Universite de Lyon, INSERM U1315, CNRS UMR5261, Lyon, France. [11]Institut de génétique et de biologie moléculaire et Cellulaire, CNRS UMR 7104 INSERM, Strasbourg, France. [12]Adelaide Centre for Epigenetic, and the South Australian Immunogenomics Cancer Institute, Faculty of Medicine Nursing and Medical Sciences, The University of Adelaide, Adelaide, Australia. [13]Present address: Lunenfeld–Tanenbaum Research Institute, University of Toronto, Toronto, Ontario, Canada. [14]These authors contributed equally: A. Huyghe, G. Furlan. ✉e-mail: aurelia.huyghe@lyon.unicancer.fr; fabrice.lavial@lyon.unicancer.fr

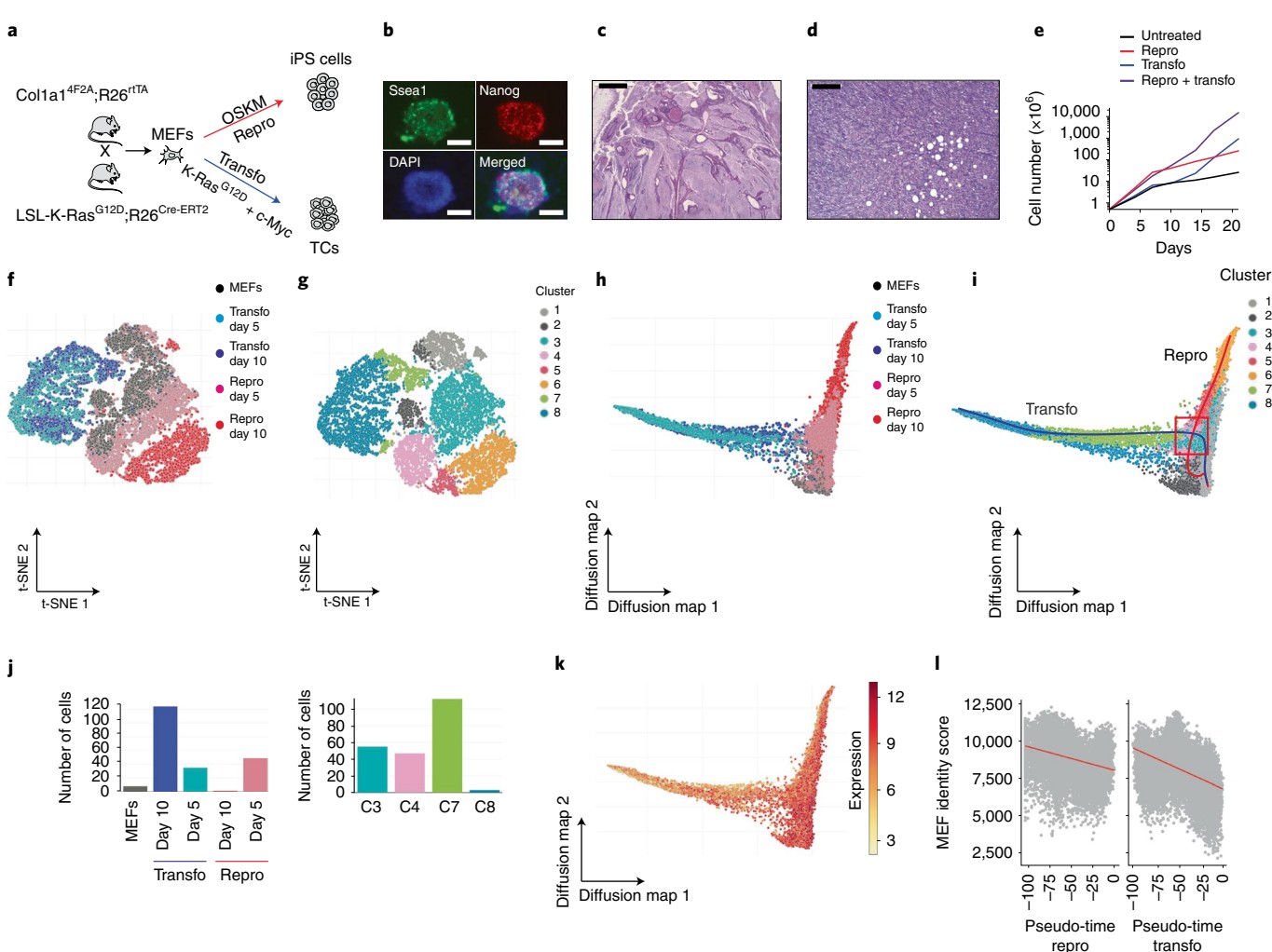

**Fig. 1 | Comparing single-cell trajectories of reprogramming and transformation. a**, Schematic of the repro-transformable mouse model. Reprogramming (repro; doxycycline-induced OSKM expression) or transformation (transfo; tamoxifen-induced K-ras[G12D] expression combined with c-Myc overexpression) gave rise to iPS cells or transformed cells (TCs), respectively. **b**, Immunofluorescence staining of repro-induced iPS cells for Ssea1 and Nanog. Scale bar, 100 μm. **c**, Histological analysis of teratomas derived from iPS cells. Scale bar, 1 mm. **d**, Tumour generated by transformed cells injected into nude mice. Scale bar, 0.2 mm. **e**, Proliferation curves of MEFs upon induction of repro, transfo and repro plus transfo. The data from one representative experiment out of two are shown. **f,g**, T-SNE visualization of scRNA-Seq profiles integrating the replicate values of 30,146 preprocessed cells (individual dots), corresponding to two biological replicates run in one sequencing experiment. The cells are coloured by sample (**f**) or by cluster (**g**). **h,i**, Diffusion maps of scRNA-Seq profiles where the cells are coloured by sample (**h**) or by cluster (**i**). The trajectories defined by Slingshot are represented by red (repro) and blue lines (transfo). The intersection area is indicated by a red box. **j**, Composition of samples in the intersection area. **k,l**, Patterns of the MEF identity signature score using gene lists from Schiebinger et al.[13], with the score represented on the diffusion map (**k**) or on the calculated pseudotime trajectories (**l**).

Altogether, these findings indicate that coordinated changes of cellular plasticity and identity are crucial for reprogramming and transformation. However, the sequence of events that orchestrate these modifications, as well as their degree of interdependency, have never been comparatively dissected. Tackling these questions is instrumental for the safe design of regenerative strategies based on in vivo reprogramming[30,31] but also to discover regulators of cancer cell plasticity[1,24]. Here, we combined a variety of single-cell, multiomics and phenotypic assays to address these questions. First, by defining the single-cell trajectories of reprogramming and transformation, we unveiled that both processes intersect early before they bifurcate. Next, we identified the TF B cell leukaemia/lymphoma 11B (Bcl11b) as a crucial regulator of reprogramming and transformation but also as a pertinent marker that, when combined with thymus cell antigen 1 theta (Thy1), delineates an ordered sequence of cellular intermediates emerging during reprogramming and transformation. Multiomics characterization led us to unveil a

regulatory axis, centred on the TF atonal bHLH transcription factor 8 (Atoh8), that acts as a broad-range lock of cellular plasticity during reprogramming, neuron transdifferentiation and transformation.

## Results

**Deciphering and comparing reprogramming and transformation.** We developed a mouse model, entitled repro-transformable, to conditionally induce reprogramming or transformation in the same population of cells (Fig. 1a). OSKM was selected as the prototypical cocktail of reprogramming[4,30–32]. The cooperation between K-ras[G12D] and c-Myc was chosen as it triggers MEF transformation[18,25,26]. R26[rtTA];Col1a1[4F2A] mice[33] carrying an inducible OSKM cassette were crossed with LSL-K-ras[G12D];R26[cre-ERT2] mice harbouring an excisable K-ras[G12D] allele[34] and MEFs were derived (Fig. 1a). Doxycycline treatment led to the formation of iPS colonies (15 days (d); efficiency = 0.21 ± 0.1%) expressing Nanog and Ssea1 (Fig. 1b) and capable of undergoing multilineage differentiation in teratomas

(Fig. 1c). Tamoxifen treatment to induce K-ras$^{G12D}$, combined with c-Myc expression (Fig. 1a), triggered transformation after serial passaging (30 d). Foci assays indicated clonal loss of contact inhibition (efficiency = 0.66 ± 0.3%) (Extended Data Fig. 1a). Soft agar assays revealed the acquisition of anchorage-independent growth potential (Extended Data Fig. 1b). Injection of transformed cells (TC) into mice led to the formation of liposarcoma-like tumours (Fig. 1d).

This model provides a unique opportunity to compare reprogramming and transformation in a genetically matched manner. We showed that MEF proliferation increased in response to both processes and this effect was cumulative (Fig. 1e). Next, we evaluated the impact of 3 d of reprogramming or transformation on DNA damage. As expected, K-ras$^{G12D}$/c-Myc triggered the formation of γH2AX phosphorylation foci in 45.1 ± 10.0% of cells (Extended Data Fig. 1c,d) and similar results were obtained with other oncogenic events, including p53 depletion and H-ras$^{G12V}$ expression (Extended Data Fig. 1e). Conversely, reprogramming did not significantly induce γH2AX foci. Moreover, when both processes were simultaneously induced, OSKM significantly prevented γH2AX foci formation triggered by K-ras$^{G12D}$/c-Myc (Extended Data Fig. 1c,d). A preventive effect of OSKM was also observed on the changes of cell cycle features but not on apoptosis induced by K-ras$^{G12D}$/c-Myc (Extended Data Fig. 1f–i). Altogether, we identified similar and divergent responses to reprogramming and transformation, as well as a preventive action of OSKM on cell cycle and DNA damage induced by K-ras$^{G12D}$/c-Myc.

**Single-cell trajectories of reprogramming and transformation.** Next, we compared the cellular trajectories of reprogramming and transformation. Single-cell RNA sequencing (scRNA-Seq) was conducted on MEFs either left untreated or induced for 5 or 10 d of reprogramming or transformation, as well as on fully reprogrammed (iPS) and transformed cells (TCs). After preprocessing 30,146 cells, principal component analysis (PCA) and t-distributed stochastic neighbour embedding (t-SNE) defined 12 clusters of cells (Extended Data Fig. 1j,k). To focus on the early dynamics, we defined eight clusters by excluding the iPS and transformed cell samples (Fig. 1f,g). Diffusion maps[35] and Slingshot[36] were used to establish pseudo-temporal ordering of cells in a high-dimensional gene expression space and to infer the cellular trajectories (Fig. 1h,i). This unveiled that single reprogramming and transforming cells (mainly from clusters 3, 4, 7 and 8) intersect within a reprogramming–transformation area before they bifurcate (Fig. 1i,j), suggesting the existence of shared transcriptomic features. Single-sample gene set enrichment analysis[37] was used next to compute activity scores for different pathways. The use of two independent scores[12,13] revealed a progressive decrease in MEF identity in cells progressing

into reprogramming and transformation trajectories, as well as in iPS and transformed cells (Fig. 1k,l and Extended Data Fig. 1l–n). In contrast, proliferation was modulated mainly independently of the trajectories (Extended Data Fig. 1o). Collectively, we unveiled an early intersection between the trajectories of reprogramming and transformation that suggests the existence of molecular similarities in individual cells.

**Bcl11b hinders reprogramming, transformation and transdifferentiation.** The scRNA-Seq dataset constitutes a unique tool to identify somatic barriers. By computing marker genes of each cluster, we identified 150 genes expressed predominantly in MEFs (clusters 1 and 2 (C1 and C2, respectively)) (Fig. 2a and Supplementary Table 1). Gene set enrichment analysis (PantherDB) highlighted enrichment for embryo development and transcription regulation (Fig. 2b). Among them, we identified the glycoprotein Thy1, which has already been reported as a MEF marker during reprogramming[6]. We assessed whether Thy1 levels correlated with reprogramming and immortalization potential. For reprogramming, Thy1$^{low}$ and Thy1$^{high}$ cells were sorted by fluorescence-activated cell sorting (FACS) after 5 d of OSKM induction and replated at similar densities. Thy1$^{low}$ cells formed significantly more alkaline phosphatase-positive (AP$^+$) iPS colonies than Thy1$^{high}$ cells, as reported previously[6,15] (Extended Data Fig. 2a,b). For transformation, with a similar sorting (5 d post K-ras$^{G12D}$/c-Myc induction), Thy1$^{low}$ cells formed fourfold more foci than Thy1$^{high}$ cells (Extended Data Fig. 2c,d). Even if the observed differences are limited, Thy1 can be used to slightly enrich fractions of cells prone to reprogramming and transformation.

Among the identified candidates, we selected the TF Bcl11b, which was previously described as a cellular identity gatekeeper in haematopoiesis, for further investigation (Fig. 2c)[38]. We showed that Bcl11b expression is high in MEFs, specifically decreased in Thy1$^{low}$ cells during reprogramming and transformation, and silenced in iPS and transformed cells (Fig. 2d and Extended Data Fig. 2e). Interrogation of published datasets broadened *Bcl11b* downregulation to keratinocyte reprogramming (Extended Data Fig. 2f).

First, we investigated Bcl11b function during transformation. Bcl11b downregulation by RNA interference (Bcl11b knockdown (KD)) (Extended Data Fig. 2g,h), before the induction of transformation, significantly increased the efficiency of soft agar colony formation. In contrast, Bcl11b overexpression (Bcl11b OE) severely hindered the process, indicating that a tight Bcl11b level safeguards MEFs from transformation (Fig. 2e,f). Similar results were obtained in foci assays (Extended Data Fig. 2i,j). Next, we assessed Bcl11b function during reprogramming. Bcl11b KD significantly improved the efficiency of generation of AP$^+$ but also Pou5f1-GFP$^+$ iPS colonies (Fig. 2g–i). Similar results were obtained using Bcl11b

**Fig. 2 | Bcl11b broadly constrains cell fate changes. a**, Patterns of the gene signature score composed of 150 genes enriched in C1 and C2 on the diffusion map. The graph integrates the 30,146 preprocessed cells of two biological replicates that were run in one sequencing experiment. **b**, Statistical over-representation assays conducted with PantherDB on the gene signature. **c**, Patterns of *Thy1* and *Bcl11b* transcript levels on the diffusion map. **d**, Western blot for Bcl11b in MEFs, Thy1$^{low}$ and Thy1$^{high}$ cells after 5 d of reprogramming and transformation. **e**, Top, experimental design. Bottom, pictures of soft agar colonies, representative of four independent experiments. **f**, Colony quantification (*n* = 4 independent experiments). **g**, Top, experimental design. Bottom, pictures of iPS colonies stained for AP. Dox, doxycycline. **h**, Colony quantification (*n* = 3 independent experiments). **i**, Pou5f1$^+$ colony quantification (*n* = 3 independent experiments). **j**, Pictures depicting the histological analysis of teratomas. Two independent teratomas were analysed per cell line. Scale bars, 1 mm. **k**, Top, experimental design. Bottom, pictures depicting mouse T cells and iPS cells obtained following reprogramming. cKO, conditional knockout; Tamox, tamoxifen. Scale bars, 120 μm. **l**, Colony quantification (*n* = 2 independent experiments). **m**, Top, experimental design. Bottom, pictures of MAP2$^+$ neural progenitors. Scale bars, 100 μm. **n**, MAP2$^+$ cell quantification (*n* = 2 independent experiments). **o**, Venn diagram showing the numbers of differentially expressed genes in control versus Bcl11b KD MEFs (orange) and control versus Bcl11b OE MEFs (blue) (log$_2$[FC] < −0.5 or >0.5; adjusted *P* value < 0.05). **p**, Distribution of endogenous Bcl11b peaks in relation to genes. UTR, untranslated region. **q**, Distribution of Bcl11b peaks in relation to the TSS. kb, kilobases. TSS, transcription start site. **r**, Most enriched DNA-binding motifs associated with Bcl11b derived from a de novo motif analysis (MEME). **s**, Graph presenting the distribution of Bcl11b peaks on genes deregulated by Bcl11b modulation in MEFs (differentially expressed genes in Bcl11b KD and Bcl11b OE versus control MEFs). **t**, Western blot depicting Bcl11b and ERK1/2 levels in Control and Bcl11b KD MEFs. In **f**, **h** and **i**, the data represent means ± s.d. Statistical significance was determined by Fisher's exact two-sided test (**b**) or two-tailed Student's *t*-test (**f**, **h** and **i**).

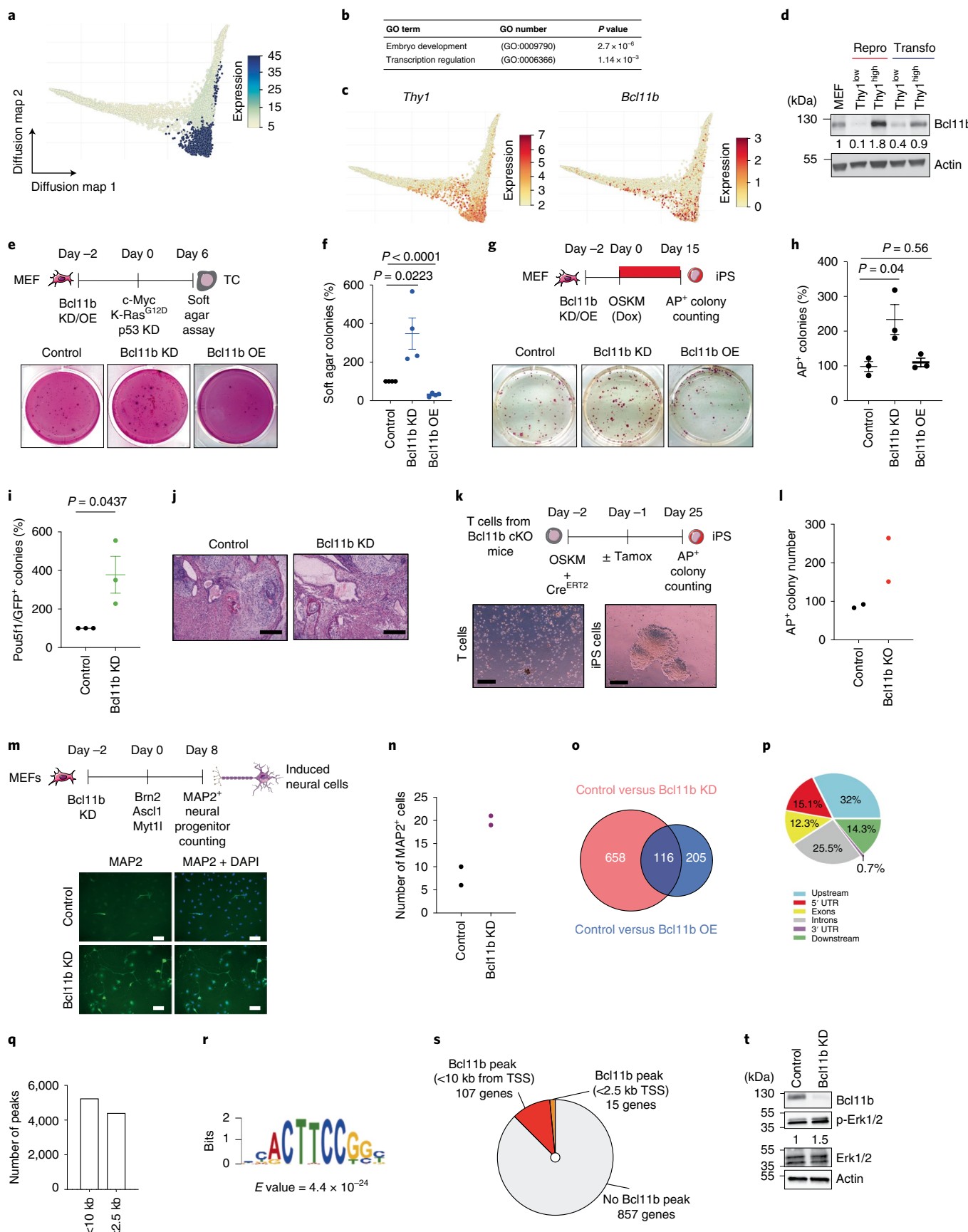

conditional KO MEFs (Extended Data Fig. 2k,l). However, Bcl11b OE did not negatively impact the reprogramming efficiency (Fig. 2g,h). Of note, Bcl11b KD iPS cell lines were capable of forming three germ layers in teratoma (Fig. 2j), indicating that Bcl11b loss is compatible with the acquisition of multilineage differentiation potential. Because Bcl11b is expressed in T lymphocytes[39], T cells isolated from mice conditional KO for Bcl11 were induced to reprogramme (Fig. 2k). Bcl11b depletion triggered the formation of twofold more AP+ iPS colonies (Fig. 2k,l). In addition, Bcl11b KD, before the induction of MEF transdifferentiation into neurons, significantly improved the efficiency of generation of MAP2+ cells (Fig. 2m,n)[40].

Next, we combined RNA-Seq and chromatin immunoprecipitation sequencing (ChIP-Seq) assays to identify the gene regulatory network (GRN) controlled by Bcl11b. Transcriptomic analyses of Bcl11b KD and Bcl11B OE MEFs led, respectively, to the identification of 774 and 321 deregulated genes compared with control MEFs (adjusted $P$ value < 0.05; $\log_2$[fold change (FC)] > 0.5 or < −0.5) (Fig. 2o). Bcl11b ChIP-Seq identified 7,430 specific peaks located mainly in the vicinity of genes (<10 kilobases from the transcription start site (TSS)) with enrichment for an Elk motif (Fig. 2p–r). Among the 979 genes deregulated by Bcl11b, 122 (12.4%) presented a Bcl11b-specific peak (Fig. 2s). Moreover, while MEF identity was not significantly impacted by Bcl11b deregulation (Extended Data Fig. 2m), we noticed that several Bcl11b targets were associated with the Mapk pathway, such as Calponin-1 and Bmf[41,42]. In line with this, we unveiled that Bcl11b constrains phospho-Erk1/2 levels in MEFs, potentially explaining its barrier role during reprogramming[43] (Fig. 2t). Altogether, we demonstrated that Bcl11b regulates reprogramming, transformation and transdifferentiation, as well as a specific GRN and phospho-Erk1/2 levels.

**Bcl11b faithfully indicates reprogramming and transforming potential.** Next, we investigated whether Bcl11b could be used as a marker to track cells changing fate using Bcl11b-tdTomato reporter MEFs (Extended Data Fig. 2n)[38]. FACS analysis confirmed that the majority of MEFs expressed Bcl11b-tdTomato. However, after 5 d of reprogramming or transformation, a subset of Bcl11b[low] cells emerged (Extended Data Fig. 2o). Bcl11b[low] cells, sorted at day 5 of reprogramming, formed sevenfold more AP+ iPS colonies than Bcl11b[high] cells (Extended Data Fig. 2p,q). Bcl11b[low] cells, sorted at day 5 of transformation, formed immortalized foci with a tenfold higher efficiency than Bcl11b[high] cells (Extended Data Fig. 2r,s). Collectively, these results identify Bcl11b as a MEF marker whose downregulation faithfully reflects the ability of cells to engage into pluripotency or immortalization paths.

**Capture of early cellular intermediates using Bcl11b and Thy1.** Our scRNA-Seq analysis did not allow the interrogation of the functional features of individual cells[4,18]. Therefore, we attempted to design a strategy to isolate early cellular intermediates using FACS. Most reprogramming strategies combined the downregulation of a MEF marker with the activation of a pluripotent factor[6,15,44–46]. However, as we aimed to capture cells emerging during both reprogramming and transformation, the use of pluripotent markers was not possible. We noticed, in contrast, that the downregulation of *Bcl11b* and *Thy1* was not occurring in the same cells during reprogramming and transformation (Fig. 2c) and in published OSK-mediated reprogramming dataset (Extended Data Fig. 3a)[14]. This finding was confirmed by the visualization of four subpopulations of cells (Bcl11b[high]/Thy1[high] (BHTH), Bcl11b[high]/Thy1[low] (BHTL), Bcl11b[low]/Thy1[high] (BLTH) and Bcl11b[low]/Thy1[low] (BLTL)) on the diffusion map or trajectories (Fig. 3a and Extended Data Fig. 3b). This result prompted us to investigate whether the combined downregulation of Bcl11b and Thy1 can be used to capture cellular intermediates by FACS. We profiled Bcl11b and Thy1 changes during reprogramming and transformation using Bcl11b-tdTomato MEFs (Fig. 3b). To begin with a homogeneous population, BHTH MEFs were FACS sorted to purity. In the absence of reprogramming or transformation, MEFs stably maintained a BHTH phenotype (Extended Data Fig. 3c,d). By day 17 of both processes, most cells displayed Bcl11b and Thy1 downregulation, as expected (Fig. 3b). However, rare BLTL cells emerged as early as 3 d after the induction of reprogramming (R-BLTL) and transformation (T-BLTL) (Fig. 3b). We demonstrated that R-BLTL and T-BLTL cells were respectively highly prone to forming iPS (Fig. 3c) or immortalized (Fig. 3d) colonies compared with R-BHTH and T-BHTH cells that remained heavily refractory. Next, we assessed the emergence of BLTL cells with alternative molecular cocktails that did not rely on c-Myc. BLTL cells emerged during reprogramming induced by Sall4-Nanog-Esrrb-Lin28 (ref. [5]) or by Oct4-Sox2-Klf4-Wnt inhibitor IWP2 (ref. [47]). BLTL cells also emerged during transformation induced by cyclin E, H-Ras[G12V] and p53 depletion (Fig. 3e and Extended Data Fig. 3e). In addition, BLTL cells were found to emerge during reprogramming and transformation of mouse adult ear fibroblasts (Fig. 3f and Extended Data Fig. 3f) and to be more efficient at forming pluripotent colonies than BHTH cells (Fig. 3g,h).

Next, we assessed whether T-BLTL cells acquired increased aggressiveness compared with T-BHTH cells[48,49] by comparing the functional features of transformed cell lines generated from these subsets of cells. Practically, T-BHTH and T-BLTL cells were FACS sorted 5 d after the induction of transformation, replated and serially passaged to establish independent polyclonal cell lines. While the cell lines presented similar growth curves when grown in two

**Fig. 3 | Sequence of intermediates during reprogramming and transformation. a**, Representation of *Bcl11b* and *Thy1* expression in single cells. The thresholds were as follows: Bcl11b < 1 and Thy1 < 2 for BLTL; Bcl11b > 2 and Thy1 < 2 for BHTL; Bcl11b < 1 and Thy1 > 4 for BLTH; and Bcl11b > 2 and Thy1 > 4 for BHTH. **b**, Expression of Bcl11b-tdTomato and Thy1 during reprogramming and transformation. KI: knock-in. **c**, Left, pictures of iPS colonies from a representative experiment. Right, quantification of AP+ colonies (*n* = 6 independent experiments). **d**, Left, pictures of foci assays from a representative experiment. Right, foci quantification (*n* = 5 independent experiments). **e**, Emergence of Bcl11b-tdTomato[low]/Thy1[low] cells. The graph represents the distribution of BHTH, BLTH, BHTL and BLTL cells. IWP2, Wnt inhibitor; OSKM, Oct4, Sox2, Klf4, c-Myc; SNEL, Sall4, Nanog, Esrrb, Lin28. BHTH cells were FACS sorted before reprogramming/transformation. **f**, Emergence of BLTL cells from mouse adult ear fibroblasts. The settings were similar to those for **e**. **g**, Left, pictures of iPS colonies following reprogramming induced by OSK + IWP2, taken from a representative experiment. Right, quantification of AP+ colonies (*n* = 2 independent experiments). **h**, Left, pictures of iPS colonies from mouse adult ear fibroblasts, taken from of a representative experiment. Right, quantification of AP+ colonies (*n* = 3 independent experiments). **i**, Schematic of the experimental design. **j**, Left, brightfield images of tumours in a chick (as indicated by the dashed lines). Right, quantification of the tumours (*n* = 23 for T-BLTL cells; *n* = 27 for T-BHTH cells). **k**, Left, tumour growth curves. Right, survival curves of the mice (*n* = 6 animals per group). **l**, Left, FACS profiles. Cells harboring various levels of Bcl11b and Thy1 were FACS sorted at day 5 of reprogramming, plated back in culture and analyzed 2 days later. Right, quantification of AP+ colonies (*n* = 5 independent experiments). **m**, Left, FACS profiles. Cells harboring various levels of Bcl11b and Thy1 were FACS sorted at day 5 of transformation, plated back in culture and analyzed 2 days later. Right, quantification of foci (*n* = 6 independent experiments). **n**, Schematic of the sequence of intermediates. The corresponding efficiencies are indicated using arbitrary units. In **h**, **j** and the left panel of **k**, the data represent means ± s.d. Statistical significance was determined by two-tailed Student's *t*-test (**c**, **d**, **h** and **j**), two-way ANOVA combined with Šidák's multiple comparisons test (left panel in **k**), Gehan–Breslow–Wilcoxon test (right panel in **k**) or one-way ANOVA followed by Tukey's post-hoc test (**l** and **m**).

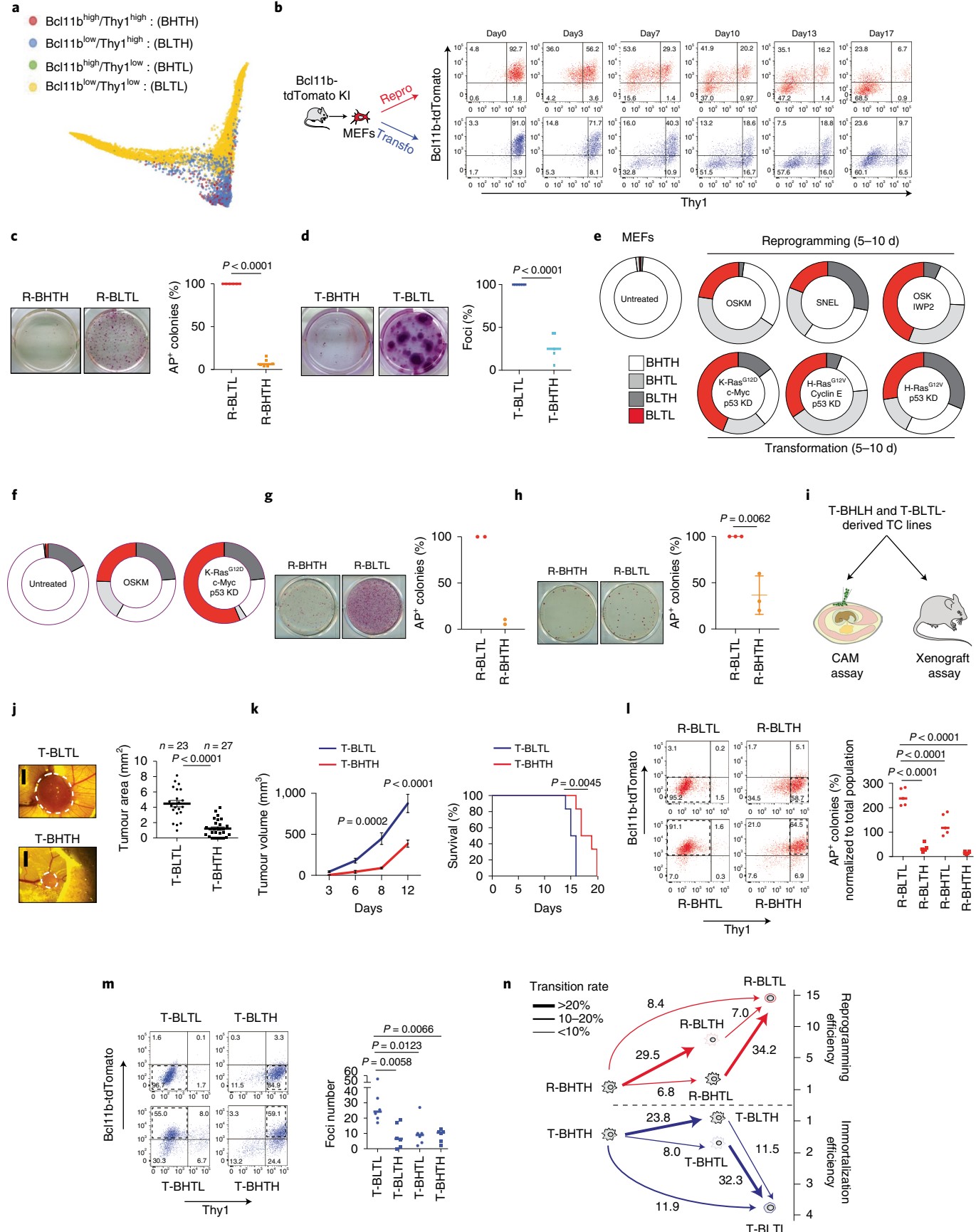

dimensions in vitro (Extended Data Fig. 3g), T-BLTL-derived lines formed sevenfold more soft agar colonies than T-BHTH-derived ones (Extended Data Fig. 3h,i). Next, we performed chick chorioal-lantoic membrane and mouse xenograft assays as in vivo models of tumorigenesis (Fig. 3i). The size of the tumours generated in chick embryos by T-BLTL-derived cells was significantly higher than by T-BHTH (Fig. 3j). An accelerated growth of T-BLTL-derived tumours and reduced survival were also observed in immunocom-promised mice (Fig. 3k). These data indicate that Thy1 and Bcl11b loss broadly delineate early intermediates highly amenable to form-ing pluripotent or tumorigenic derivatives.

**Sequence of intermediates during reprogramming and transformation.** Next, we sought to characterize the sequential emergence of intermediates during reprogramming and transformation. To ensure that changes in Bcl11b/Thy1 (Fig. 3b) reflected the transi-tion of individual cells from one stage to the next, and not merely the loss of one major population and expansion of another, each fraction was sorted after 5 d of reprogramming then replated for 48 h before FACS analysis. The progression of cellular intermedi-ates revealed the routes induced by OSKM. First, we observed that R-BLTL cells were stable as they did not transit efficiently into other states. R-BHTH and R-BHTL cells generated R-BLTL cells at a very low rate while R-BLTH cells transited into R-BLTL cells efficiently (35%) (Fig. 3l), suggesting that Bcl11b downregulation is a rate-limiting step of reprogramming, in line with our previous results (Fig. 2). Importantly, these cellular progressions were cor-related with the capacity of the intermediates to form AP+ colonies (Fig. 3l and Extended Data Fig. 3j). For transformation, the T-BLTL state was also relatively stable and prone to forming immortalized foci (Fig. 3m). T-BHTH and T-BLTH cells were poorly efficient at generating T-BLTL while T-BHTL cells efficiently reached this state (Fig. 3m and Extended Data Fig. 3k). On this basis, we generated the functional roadmaps presented in Fig. 3n.

**Chromatin reconfigurations in early intermediates.** We next dissected the reconfigurations of chromatin accessibility by con-ducting assay for transposase-accessible chromatin sequencing (ATAC-Seq) on cellular intermediates captured at day 5 of repro-gramming or transformation. PCA analysis showed that R-BLTL and T-BLTL cells segregated together on the x axis (principal com-ponent 1) (Fig. 4a) and towards the direction of the iPS/transformed cells (Extended Data Fig. 4a), suggesting the existence of com-mon changes of chromatin accessibility, as exemplified with *Thy1* (Fig. 4b). To test this, we classified the peaks in clusters defining regions that were accessible in MEFs but not in both R-BLTL and T-BLTL cells (C1); became accessible in both R-BLTL and T-BLTL cells over MEFs (C2); specifically lost (C3) or gained (C4) acces-sibility in R-BLTL cells over MEFs and T-BLTL cells; or specifically lost (C5) or gained (C6) accessibility in T-BLTL cells over MEFs and

R-BLTL cells (Fig. 4c). We also generated an unsupervised heatmap to visualize peak intensities for differential loci (Extended Data Fig. 4b). To uncover TFs possibly driving these changes, we per-formed a DNA motif enrichment on clusters (Fig. 4d and Extended Data Fig. 4c). A subset of regulatory elements changing accessibil-ity in R-BLTL and T-BLTL cells were enriched in the FosL1 motif (C2, C3 and C5), suggesting relocation of this TF. We assessed whether Fosl1 functionally regulates both processes. FosL1 deple-tion (Extended Data Fig. 4d,e) led to a fourfold reduction in the number of immortalized foci (Extended Data Fig. 4f,g) and an aver-age sixfold increase in reprogramming efficiency (Extended Data Fig. 4h,i). Hence, ATAC-Seq shed light on changes in chromatin accessibility that occur specifically or commonly during repro-gramming and transformation, as well as identified FosL1 as a common but antagonistic regulator.

**Transcriptomic changes in early intermediates.** PCA conducted on RNA-Seq data of day 5 cellular intermediates revealed that R-BLTL and T-BLTL cells segregated together on the x axis, suggest-ing common changes (Fig. 4e). In addition, a significant number of the genes deregulated by Bcl11b modulation in MEFs (Fig. 2o) were also impacted in both R-BLTL and T-BLTL cells (Fig. 4f). Next, we exploited published datasets to characterize R-BLTL and T-BLTL cells. MEF identity scores[12,13] were not downregulated in R-BLTL and T-BLTL cells (Extended Data Fig. 4j), indicating that these cells constituted early intermediates that gained plasticity but did not yet downregulate identity, in contrast with previously isolated interme-diates[8]. In line with this, R-BLTL and T-BLTL cells did not induce *CD73* and *CD49d*, which delineate late intermediates (Extended Data Fig. 4k)[45]. R-BLTL and T-BLTL cells harboured some mod-erated reductions in stromal markers (*Csf1*, *Prrx1* and *Id3*) but no concomitant inductions of mesenchymal-to-epithelial transition (MET) markers (*Fut9* and *Zic3*), reinforcing the notion that they are not yet fully engaged on a MET trajectory (Extended Data Fig. 4l)[13]. These findings demonstrate that a gain of cellular plas-ticity is not correlated with, but rather precedes, a loss of cellular identity or engagement into MET. Next, we attempted to position R-BLTL and T-BLTL cells on the trajectories of reprogramming and transformation by defining activity scores. Cells with high R-BLTL and T-BLTL score activity emerged early along the respective tra-jectories and increased in number during progression towards plu-ripotency and malignancy (Fig. 4g).

To identify molecular regulators of cellular plasticity, we next interrogated the specific transcriptomic features of R-BLTL and T-BLTL cells. Some 410 genes were differentially expressed between R-BLTL and R-BHTH cells and 1,389 genes were differentially expressed between T-BLTL and T-BHTH cells, with a shared sig-nature of 301 genes (adjusted P value < 0.05; log2[FC] > 1 or < −1) (Fig. 4h) enriched in stem cell differentiation but also immunity (Fig. 4i)[8]. Interestingly, the modulation of these genes also occurred

---

**Fig. 4 | Chromatin and transcriptome reconfigurations in cellular intermediates. a**, PCA conducted on ATAC-Seq data. Untreated MEFs (black), BLTL and BHTH cells FACS sorted after 5 d of reprogramming (red) or transformation (blue) are represented. **b**, Example of ATAC chromatin sites at the *Thy1* locus. **c**, Definition of the clusters described in the main text (n = 2 independent experiments). Central lines represents medians, box edges represent upper and lower quartiles and whiskers show the highest and lowest values, excluding outliers (at most 1.5× the interquartile range above or below the upper and lower quartile). **d**, Enrichment in TF motifs. Each point represents significant enrichment in the motif (x axis) for the cluster (y axis). The point size represents the proportion of sequences in the cluster featuring the motif and the colour gradient represents the enrichment significance. **e**, PCA conducted on RNA-Seq data. **f**, Venn diagram showing the numbers of differentially expressed genes in MEFs versus R-BLTL cells (red), MEFs versus T-BLTL cells (blue) and control MEFs versus Bcl11b KD MEFs (green) (log2[FC] < −0.5 or > 0.5; base mean < 40; adjusted P value < 5 × 10⁻²). **g**, Visualization of R-BLTL and T-BLTL score activities on single-cell trajectories. **h**, Venn diagram showing the numbers of differentially expressed genes in T-BLTL versus T-BHTH cells (blue) and R-BLTL versus R-BHTH cells (red) (log2[FC] < −1 or > 1; adjusted P value < 5 × 10⁻²). **i**, Statistical over-representation assays conducted with PantherDB. Statistical significance was determined by Fisher's exact two-sided test. **j**, Top, patterns of the downregulated genes signature plotted on the diffusion map. Bottom, patterns of the upregulated genes signature plotted on the diffusion map. **k**, Heatmap clustering the 301 commonly deregulated genes. The MEF sample was excluded from the representation. **l**, Western blot depicting Atoh8, Id4, Twist2 and Gapdh levels in cellular intermediates. **m**, Western blot depicting Atoh8, Id4, Twist2 and Actin levels in MEFs, iPS and transformed cells.

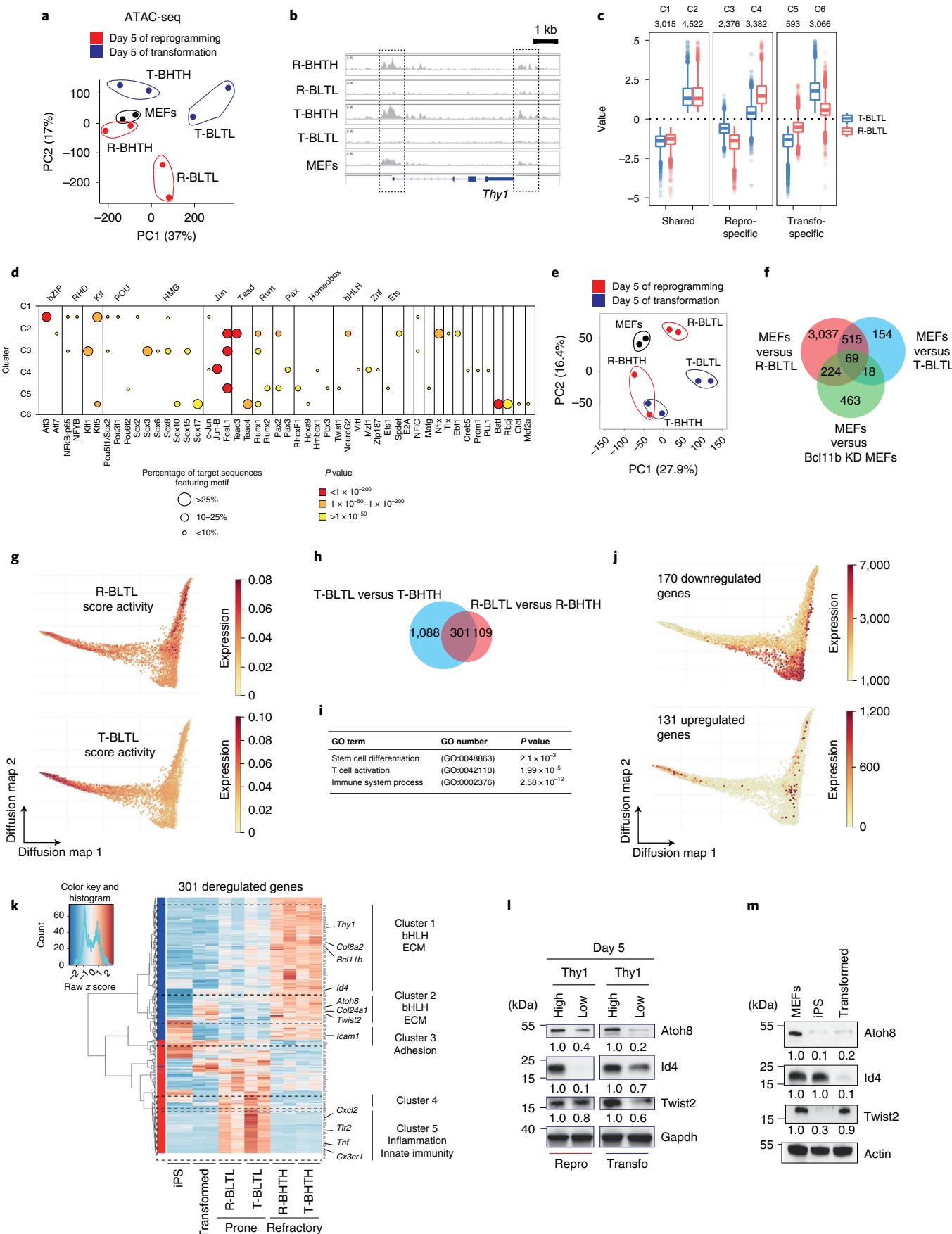

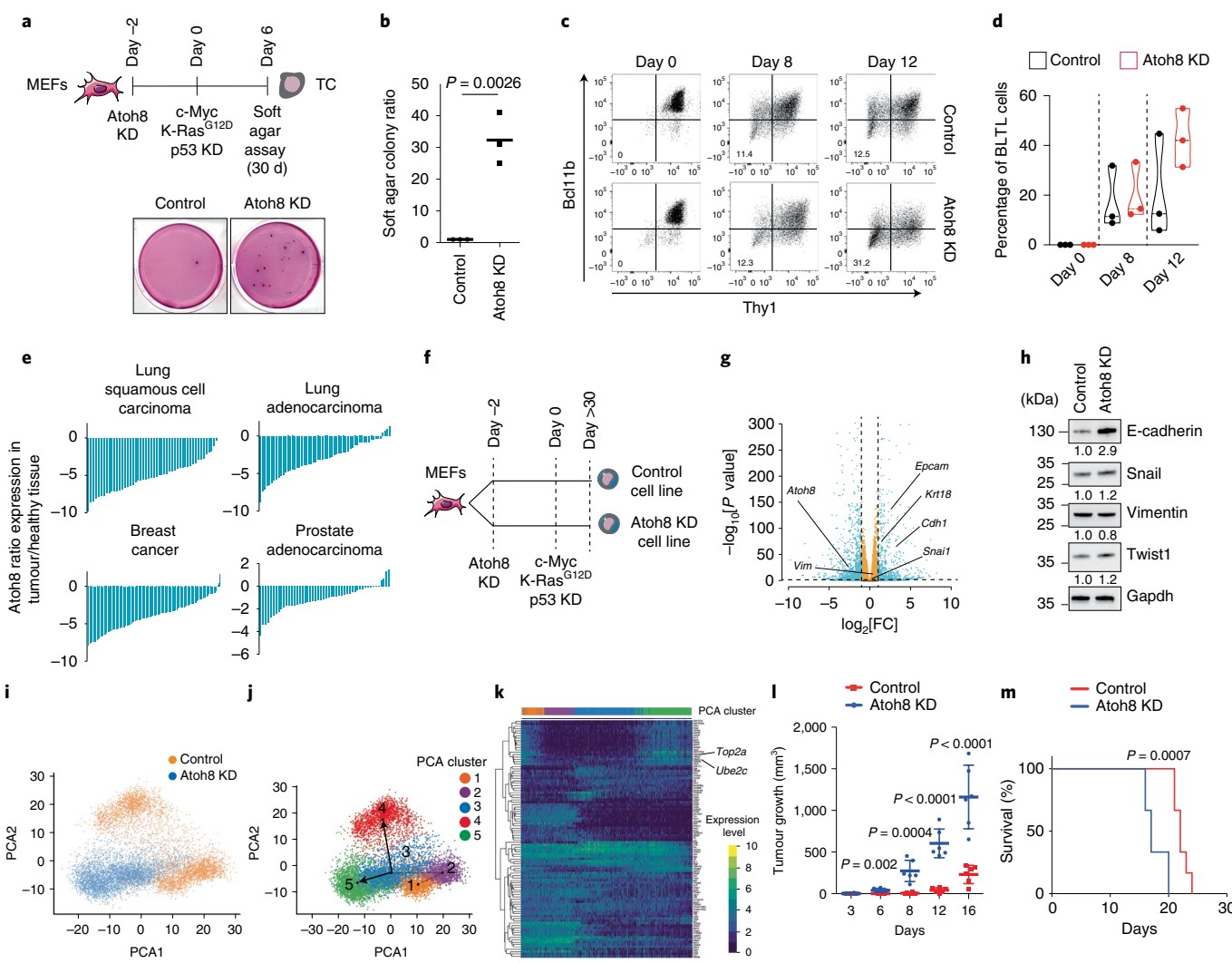

**Fig. 5 | Atoh8 regulates the acquisition of transformed features. a**, Top, experimental design. Bottom, picture of soft agar colonies, representative of three independent experiments. **b**, Colony quantification (*n* = 3 independent experiments). **c**, FACS analysis showing the emergence of Bcl11b-tdTomato^low^/Thy1^low^ (BLTL) cells during transformation in Control and Atoh8 KD backgrounds. **d**, Graph depicting the emergence of BLTL cells (*n* = 3 independent experiments). **e**, Histograms depicting *Atoh8* transcript levels in patients. The data are presented as the log₂[ratio of Atoh8 fragments per kilobase of transcript per million mapped reads (FPKMs)] between malignant and healthy tissues in paired samples. **f**, Schematic of the experimental design. Cells were split for at least ten passages (30 d) before subsequent analyses. **g**, Volcano plot showing differentially expressed genes in Control versus Atoh8 KD transformed line. Each dot corresponds to a transcript. Blue dots represent log₂[FC] > 1 or < −1 and adjusted *P* value < 0.00001. Benjamini–Hochberg-adjusted *P* values of the comparisons were computed using the limma-voom workflow modified two-sided *t*-test. **h**, Western blot for E-cadherin, Snail, Vimentin, Twist1 and Gapdh in Control and Atoh8 KD transformed cell lines. **i**, PCA of the scRNA-Seq data. **j**, Single-cell pseudotime trajectories. **k**, Heatmap of the top temporally expressed genes based on the Atoh8 KD cell trajectory. **l**, Xenograft tumour volume over time (*n* = 6 independent mice per group). **m**, Survival plot of the mice from **l** (*n* = 6 independent mice). In **b**, **d** and **l**, the data are presented as means ± s.d. Statistical significance was determined by two-tailed Student's *t*-test (**b**, **d** and **l**) or Kaplan–Meyer test (**m**).

in the single-cell dataset (Fig. 4j). Next, we conducted heatmap analysis to identify clusters of genes permanently or transiently deregulated in R-BLTL and T-BLTL cells (Fig. 4k). Cluster 1 corresponded to genes permanently repressed during reprogramming and transformation, including *Thy1* and *Bcl11b*. Clusters 2 and 3 encompassed genes transiently repressed in R-BLTL and T-BLTL cells but reactivated respectively in transformed or iPS cells such as *Icam1* (ref. 44). Of importance, we noticed that a network of basic helix–loop–helix (bHLH) TFs, encompassing the *Atoh8*, *Id4* and *Twist2* transcripts, was downregulated in R-BLTL and T-BLTL cells (Fig. 4k). We confirmed the downregulation of Atoh8, Id4 and Twist2 proteins in cellular intermediates (Fig. 4l) but Atoh8 silencing was solely maintained in iPS and transformed cells (Fig. 4m).

Overall, these results identify shared transcriptional modifications occurring during reprogramming and transformation.

**Atoh8 regulates the acquisition of malignant features.** We addressed whether Atoh8 functionally controls cellular plasticity during transformation. Atoh8 KD, before the induction of transformation, significantly increased the ability of MEFs to grow independent of anchorage (Extended Data Fig. 5a–c) and form immortalized foci (Extended Data Fig. 5d–f). Comparable results were obtained by CRISPR–Cas9 Atoh8 KO (Extended Data Fig. 5g–j), demonstrating that Atoh8 constrains transformation. To assess whether Atoh8 controls the pace at which MEFs acquire malignant features, control and Atoh8 KD MEFs were subjected to soft agar

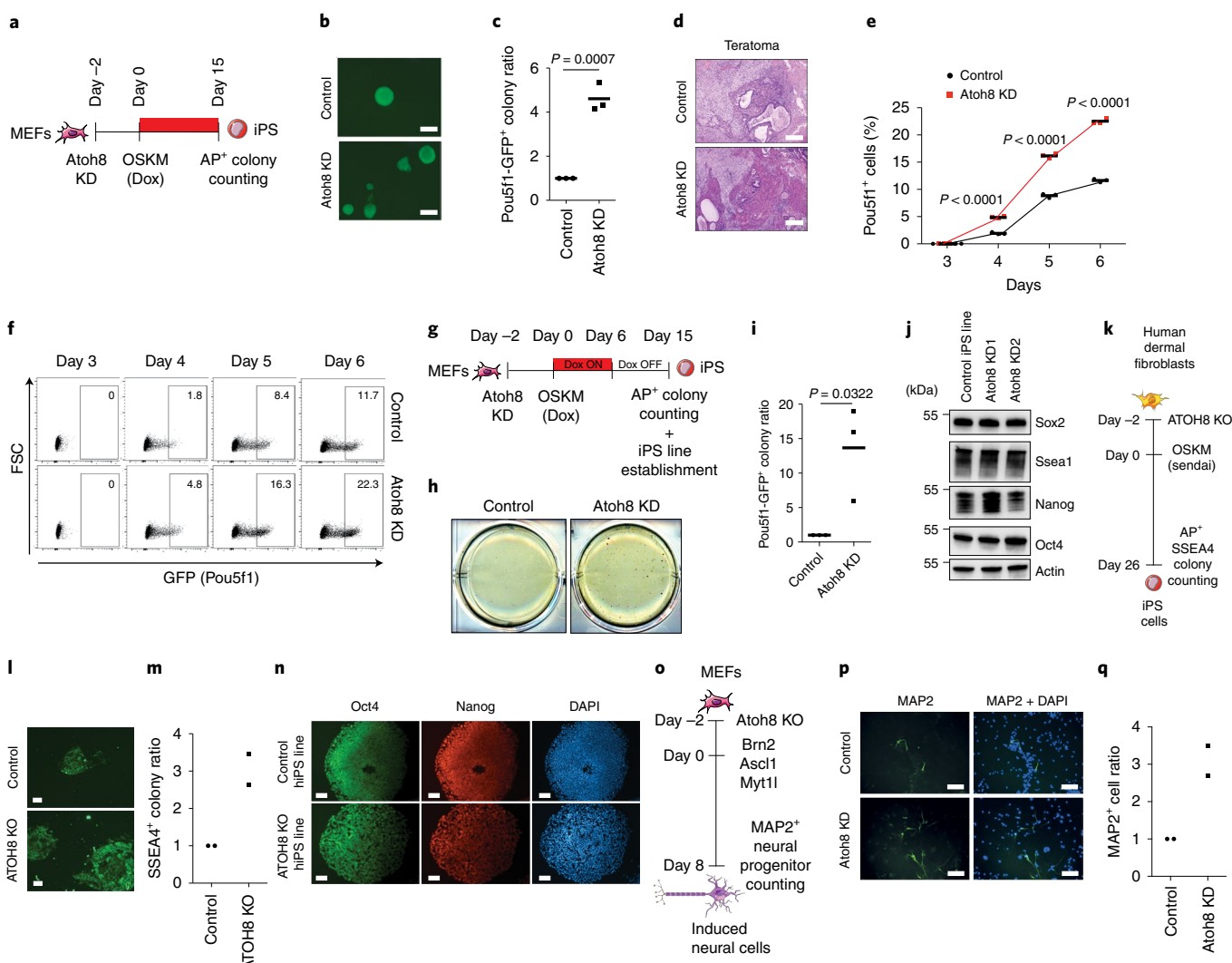

**Fig. 6 | Atoh8 constrains cellular plasticity during reprogramming and transdifferentiation. a**, Experimental set-up. **b**, Immunofluorescence of Pou5f1-GFP⁺ colonies at day 15 of reprogramming. Scale bars, 100 μm. **c**, Pou5f1⁺ colony quantification (*n* = 3 independent experiments). **d**, Histological analysis of teratomas. Two independent teratomas were analysed per cell line. Scale bars, 1 mm. **e**, Graph depicting the percentage of Pou5f1-GFP⁺ cells (*n* = 3 independent experiments). **f**, FACS analysis showing the emergence of Pou5f1-GFP⁺ cells. **g**, Experimental design. **h**, Image representing AP⁺ colonies at day 15 of reprogramming, representative of three independent experiments. **i**, Colony quantification (*n* = 3 independent experiments). **j**, Western blot for Sox2, Ssea1, Nanog, Oct4 and Actin in Control and Atoh8 KD-derived iPS cell lines. **k**, Schematic of human induced pluripotent stem cell reprogramming. **l**, Picture representing SSEA4⁺ colonies at day 26 of reprogramming, representative of two independent experiments. Scale bars, 100 μm. **m**, Colony quantification (*n* = 2 independent experiments). **n**, Immunofluorescence for Oct4 and Nanog. hiPS, human induced pluripotent stem. Scale bars, 100 μm. **o**, Schematic depicting MEF-to-neuron transdifferentiation. **p**, MAP2 immunostaining. Scale bars, 100 μm. **q**, MAP2⁺ cell quantification per field (*n* = 2 independent experiments). In **c**, **e** and **i**, the data represent means ± s.d. and statistical significance was determined by two-tailed Student's *t*-test.

assays as early as 6 d after transformation induction (Fig. 5a). Atoh8 KD cells succeeded in forming colonies, while control cells largely failed (Fig. 5a,b), demonstrating that Atoh8 constrains the temporal acquisition of anchorage-independent growth properties. In line with this, Atoh8 depletion was found to accelerate the emergence of T-BLTL cells (Fig. 5c,d).

In line with a putative function in tumorigenesis, data from The Cancer Genome Atlas (TCGA) revealed significant downregulation of *Atoh8* expression in malignant tissues compared with paired peritumoral tissues in various cancers (Fig. 5e)[50]. While the role of Atoh8 in established cancer cells has been addressed, its function in cellular plasticity during transformation remains unknown[50]. We addressed this question by establishing polyclonal cell lines transformed in the presence or absence of Atoh8 (Fig. 5f). Bulk RNA-Seq led to the identification of 803 differentially expressed genes with

a significant induction of epithelial markers (*Cdh1* and *Epcam*) (log₂[FC] > 1 and <1; adjusted *P* value < 10⁻⁵) (Fig. 5g and Extended Data Fig. 5k), indicating that Atoh8 impacts the establishment of the transformed transcriptome. We confirmed the significant increase in the E-cadherin protein but also the steady expression of Vimentin or Snail in Atoh8 KD-derived cells (Fig. 5h and Extended Data Fig. 5l). Next, we conducted scRNA-Seq to assess whether these differences corresponded to the emergence of alternative cellular states. Among five clusters of cells, PCA and t-SNE analyses revealed that C3 was found in both Control and Atoh8 KD populations (Fig. 5i,j). In contrast, C5 was composed nearly exclusively of Atoh8 KD-derived cells (Extended Data Fig. 5m). Pseudotime calculations inferred two main trajectories: C1–C2–C3–C4 and C1–C2–C3–C5 (Fig. 5j). These data demonstrate that Atoh8 depletion diverts cells during transformation towards a new cluster enriched in genes

linked to cancer cell invasion such as *Ube2c*[51] (Fig. 5k). Based on this, we assessed whether Atoh8 regulates tumorigenicity. Atoh8 KD-derived lines were found to be significantly more prone to growing under non-adherent conditions (Extended Data Fig. 5n–s). Injection into immunocompromised mice showed an increased growth of Atoh8 KD-derived tumours and a significant reduction in the overall survival of mice (Fig. 5l,m). Collectively, these data indicate that Atoh8 constrains cellular plasticity and the emergence of highly aggressive tumour cells.

**Atoh8 hinders reactivation of the pluripotent network.** Interrogation of reprogramming datasets[6,9,12,13,52,53] confirmed Atoh8 expression in MEFs and downregulation in reprogramming intermediates (Extended Data Fig. 6a). In contrast, mouse adult ear fibroblasts barely expressed Atoh8, as reported (Extended Data Fig. 6b,c)[54]. Atoh8 KD (Fig. 6a and Extended Data Fig. 6d,e) led to a fourfold increase in the number of AP+ (Extended Data Fig. 6f,g) and Pou5f1-GFP+ iPS colonies (Fig. 6b,c). Similar improvements were observed with Atoh8 CRISPR–Cas9 KO (Extended Data Fig. 6h–j), demonstrating that Atoh8 constrains mouse reprogramming. In contrast, Atoh8 OE was not sufficient to constrain reprogramming or transformation (Extended Data Fig. 6k–n). Atoh8 KD established iPS lines expressed similar Oct4, Sox2 and Nanog levels to controls (Extended Data Fig. 6o,p) and differentiated into three germ layers in teratoma (Fig. 6d). During reprogramming, cells activate the endogenous pluripotency network and become independent of the OSKM transgenes[15]. We showed that the emergence of Pou5f1-GFP+ cells is significantly accelerated by Atoh8 depletion (Fig. 6e,f). To evaluate transgene independency, OSKM doxycycline-inducible MEFs were exposed to doxycycline for 6 d and iPS colony emergence was monitored (Fig. 6g). Atoh8 KD cells succeeded in forming AP+ iPS colonies while control cells failed (Fig. 6h,i). Of note, iPS cell lines derived from Pou5f1-GFP+ Atoh8 KD cells FACS sorted at day 6 of reprogramming expressed similar levels of pluripotency markers to bona fide iPS lines (Fig. 6j and Extended Data Fig. 6q,r). These findings demonstrate that Atoh8 hinders reactivation of the endogenous *Pou5f1* gene and acquisition of transgene independency.

**Atoh8 constrains human reprogramming and mouse transdifferentiation.** We assessed Atoh8 function in human dermal fibroblast (HDF) reprogramming, during which it is also rapidly downregulated (Fig. 6k and Extended Data Fig. 7a)[11,55]. CRISPR–Cas9 ATOH8 KO significantly increased the number of AP+ (Extended Data Fig. 7b,c) and SSEA4+ iPS colonies (Fig. 6l,m). Control and ATOH8 KO iPS cell lines expressed comparable levels of pluripotency markers (Fig. 6n and Extended Data Fig. 7d), as well as differentiation genes in embryoid bodies (Extended Data Fig. 7e). Next, we evaluated

whether Atoh8 also hinders MEF-to-neuron transdifferentiation (Fig. 6o)[40]. Atoh8 depletion led to a threefold increase in the number of MAP2+ induced neuronal cells (Fig. 6p,q), demonstrating that Atoh8 broadly constrains TF-mediated cell conversions.

**Atoh8 fine tunes WNT signalling via Sfrp1.** Assuming that TF binding determines its function, we assessed Atoh8 genomic distribution in MEFs (Extended Data Fig. 7f). ChIP-Seq led to the identification of 1,826 peaks, principally distributed in upstream and downstream gene regions and introns (Fig. 7a,b). The specificity of reads was confirmed with a mock sample (Extended Data Fig. 7g)[56]. Motif analysis showed that Atoh8 bound preferentially to the CAGCTG motif (E-box), as expected for a bHLH TF, even if the AP-1 motif was also enriched (Fig. 7c). In contrast with the bHLH TFs Ascl1 and MyoD, which mainly bind to inaccessible regions as pioneer factors, Atoh8 binds preferentially in accessible (ATAC-Seq-positive) enhancer regions enriched in H3K27Ac/H3K4Me1, distant from the TSS[9,57] (Fig. 7d and Extended Data Fig. 7h–j). We noticed a significant overlap in Atoh8 and Bcl11b binding, especially on the *Atoh8* locus itself (Fig. 7e and Extended Data Fig. 7k).

Next, we conducted ATAC-Seq on MEFs after 5 d of reprogramming or transformation to assess the dynamic behaviour of Atoh8-bound regions. They remained overall largely accessible except in iPS cells (Fig. 7f,g). To focus on reprogramming, we conducted a chromatin combinatorial state analysis using data for chromatin accessibility, histone marks and TFs in MEFs, after 48 h of OSKM expression and in pluripotent cells[9,58]. Atoh8-bound regions progressively lost H3K27Ac/H3K4Me1 while gaining H3K9Ac in pluripotent cells (Fig. 7h). A significant proportion of Atoh8 peaks were co-occupied by c-Myc in MEFs, but this fraction dropped significantly during reprogramming, indicating a gradual relocation of c-Myc, in parallel with a transient binding of Oct4 and, to a lesser extent, Sox2. Altogether, these data indicate highly dynamic reconfigurations of the Atoh8-bound regions during reprogramming and transformation.

Next, we defined the GRN controlled by Atoh8. RNA-Seq conducted on MEFs (Control) and after 5 d of Atoh8 KD identified 503 deregulated genes ($\log_2[FC] > 0.8$ and $< -0.8$; adjusted $P$ value $< 10^{-5}$) (Fig. 7i). At this time point, MEF identity scores were not impacted, suggesting that the Atoh8 primary function is not to safeguard identity (Extended Data Fig. 7l). Interestingly, Atoh8 depletion induced c-Myc (Fig. 7i and Extended Data Fig. 7m). Conversely, c-Myc bound to the Atoh8 promoter and repressed its expression (Fig. 7j,k), revealing a negative feedback loop. In addition, over-representation assays linked Atoh8 with WNT signalling and *Sfrp1* (Fig. 7i,l), in line with β-catenin upregulation in Atoh8 KD MEFs (Extended Data Fig. 7n). These results prompted us to evaluate whether Atoh8 constrains cellular plasticity by tuning

**Fig. 7 | Atoh8 restrains cellular plasticity by binding specific enhancers to limit WNT signalling activity. a**, Heatmap displaying Atoh8 read counts ± 1 kilobase around merged peak summits. **b**, Genomic distribution of Atoh8-specific peaks. **c**, The most enriched DNA-binding motifs associated with Atoh8, derived from a de novo motif analysis (MEME). **d**, Mean read count enrichment density associated with H3K27Ac and H3K4Me1. **e**, Venn diagram showing the numbers of Bcl11b versus Atoh8 peaks. **f**, Fraction of Atoh8 sites retrieved within open chromatin regions (ATAC-Seq) during reprogramming and transformation. **g**, Examples of open chromatin sites. Chr, chromosome. **h**, Atoh8-centred chromatin state analysis. ES, Embryonic stem cells. **i**, Volcano plot showing differentially expressed genes in Atoh8 KD versus control MEFs. Samples were collected 5 d after RNA interference induction. Each dot corresponds to a transcript. Benjamini–Hochberg-adjusted $P$ values of the comparisons were computed using the limma-voom workflow modified two-sided $t$-test. **j**, Top, schematic depicting the c-Myc binding site (BS) on the Atoh8 promoter. Bottom, real-time qPCR showing the levels of DNA immunoprecipitated. The data are represented as a percentage of Atoh8 DNA levels in ChIP input and are from two independent experiments. **k**, Western blot for c-Myc, Atoh8 and Gapdh in MEFs in response to c-Myc exogenous expression. **l**, Graph depicting the fold-enrichment as determined by statistical over-representation analysis. RTK, Receptor tyrosine kinase. **m**, Top, pictures representing AP+ colonies. Bottom, pictures representing cresyl violet foci. **n**, Quantification ($n = 3$ independent experiments). **o**, Quantification of AP+ colonies and foci ($n = 3$ independent experiments). **p**, Top, pictures representing AP+ colonies. Bottom, pictures representing cresyl violet foci. **q**, Quantification ($n = 6$ independent experiments for the left panel and $n = 3$ for the right panel). **r**, Western blot of Atoh8 expression in MEFs treated with recombinant Wnt3a for 48 h. **s**, Schematic recapitulating the main findings of the study. In **n**, **o** and **q**, the data represent means ± s.d. and statistical significance was determined by two-tailed Student's $t$-test.

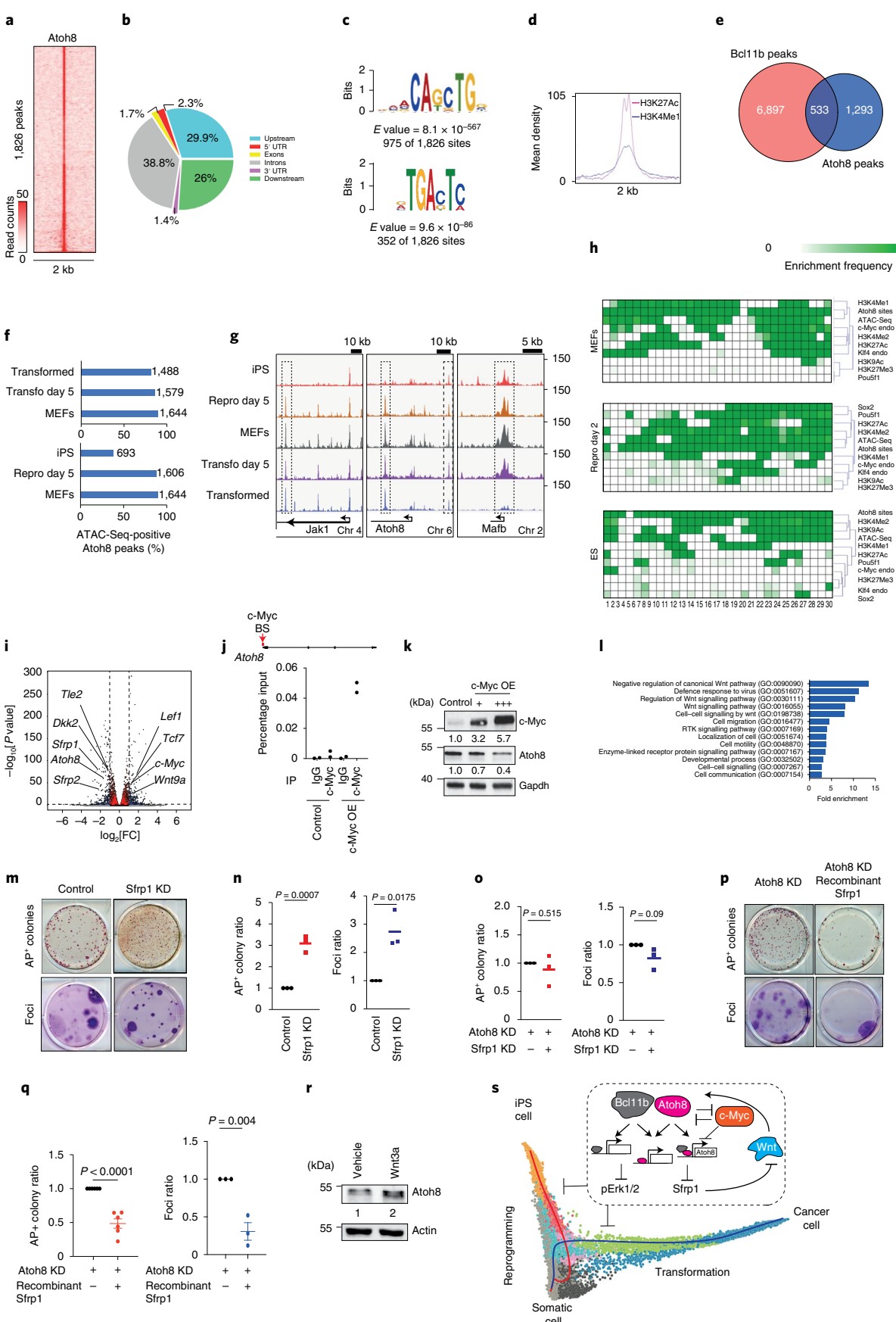

WNT signalling via Sfrp1. We first found that *Sfrp1* is downregulated in R-BLTL and T-BLTL cells (Extended Data Fig. 7o). Next, we assessed whether Sfrp1 depletion mimics the effects of Atoh8 KD. Sfrp1 KD (Extended Data Fig. 7p,q) led to a significant increase in the numbers of AP[+] colonies and immortalized foci (Fig. 7m,n). Moreover, simultaneous suppression of Atoh8/Sfrp1 improved reprogramming/transformation efficiencies in a similar range to Atoh8, indicating that Atoh8 exerts its function via a regulatory axis involving Sfrp1 (Fig. 7o). In line with this, we demonstrated that the effect of Atoh8 on reprogramming/transformation is significantly reduced by recombinant Sfrp1 (Fig. 7p,q). WNT signalling activation also induces Atoh8 expression, highlighting a negative feedback loop (Fig. 7r). Finally, we evaluated whether Bcl11b, Atoh8, FosL1 and Sfrp1 regulate the expression of drivers of reprogramming/transformation and of each other. We showed by RNA interference that they do not regulate the expression of Oct4 and K-ras[G12D] (Extended Data Fig. 7r). However, putative interplays between FosL1 and Sfrp1 were identified (Extended Data Fig. 7s). Thus, we showed that Atoh8 is a target of WNT and c-Myc that constrains cellular plasticity by tuning WNT activity via Sfrp1.

## Discussion

Coordinated changes of cellular plasticity and identity emerged as critical for reprogramming and transformation[1,20,23,59]. However, the cellular and molecular roadmaps that orchestrate these intermingled modifications, as well as their degree of analogy and coupling, have never been comparatively dissected despite constituting crucial topics for regenerative medicine[60]. Because genetic models enabling the conduction of such analyses were lacking, in this study we generated repro-transformable mice to rigorously compare the responses to reprogramming and transformation in a genetically matched manner. In complement to reports defining trajectories of reprogramming[6,11,13,14,45], we provide a high-resolution analysis of the early cellular intermediates that emerge simultaneously during both processes. First, by inferring the single-cell trajectories of reprogramming and transformation, we identified the TF Bcl11b as a broad regulator of cell fate, enlarging initial findings in haematopoietic cells[38], but also as a faithful indicator of the propensity of cells to change fate. This reconstruction also unveiled how cells intersect in a reprogramming–transformation area, in which the transcriptomes of reprogramming and transforming cells share analogies. The combined use of Bcl11b and Thy1 confirmed the existence of cellular intermediates sharing transcriptomic but also epigenomic traits, therefore providing a concrete demonstration of the previously proposed correlative analogies between cancer and reprogramming[21,22,59]. Of interest, we underlined cell-surface markers (CD14, CD53, CD72 and CD84) that are transiently upregulated in intermediates and might help to refine their isolation. We also unveiled that a gain of cellular plasticity and a loss of cellular identity are uncoupled in these intermediates. The acquisition of plasticity in R-BLTL and T-BLTL cells indeed precedes the significant loss of MEF identity. While both phenomena are coupled in the literature[6,11,13,14,45], our results rather suggest the existence of partitioned regulatory networks controlling identity or plasticity.

Next, we exploited this finding to uncover molecular determinants of cellular plasticity. We identified the TF Atoh8—initially described in neurodevelopment and later on as a cellular context-dependent regulator of reprogramming[50,54,61]—as a MEF-specific barrier of cellular plasticity. Genome-wide binding analysis revealed a unique location to highly dynamic enhancer regions compared with other bHLH TFs (that is, c-Myc, Ascl1 and MyoD)[9,57]. Moreover, Atoh8 was found to control a specific GRN that tunes WNT signalling—a pathway well described to promote plasticity in regeneration and cancer[62,63]. Therefore, we revealed the existence of a c-Myc/Atoh8/Sfrp1 regulatory axis specifically

constraining plasticity without impacting identity, reinforcing the concept that both processes are uncoupled and regulated by specific networks of genes. The control of cellular plasticity and identity has emerged as one of the most crucial research topics for regenerative medicine and cancer biology[60]. By providing a single-cell comparative deconvolution of reprogramming and transformation, and by identifying molecules uncoupling the gain of plasticity and the loss of identity in various biological processes (Fig. 7s), our work provides a conceptual framework that opens fascinating perspectives for regenerative medicine and cancer biology.

## Online content

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

## Methods

**Mice and MEFs.** The animals were maintained in a specific pathogen-free animal facility P-PAC (Plateforme du petit animal du CRCL) at the Cancer Research Center of Lyon and the Center Léon Bérard, Lyon, France. All of the experiments were performed in accordance with the animal care guidelines of the European Union and were validated by the local Animal Ethics Evaluation Committee (C2EA-15 agreed by the French Ministry of High School and Research). R26$^{rtTA}$;Col1a1$^{4F2A}$ (ref. [64]), LSL-K-ras$^{G12D}$ (ref. [34]), R26-CRE$^{ERT2}$, Oct4-EGFP and Bcl11b-tdTomato and Bcl11b$^{flox/flox}$[38] mice were housed under standard conditions and bred in accordance with French national guidelines. Genotyping was carried out on genomic DNA derived from adult and embryonic tails using the DirectPCR Lysis Reagent (102-T; Viagen Biotech) and EconoTaq Plus Green 2X Master Mix (Lucigen). Supplementary Table 2 lists the primers that were used.

MEFs were isolated from E13.5 embryos after removal of the head and internal organs. The remaining tissues were physically dissociated and incubated in trypsin at 37 °C for 10 min, after which the cells were resuspended in MEF medium. When indicated, MEFs were treated with 1 μg ml⁻¹ recombinant Sfrp1 (9019-SF-025; R&D Systems) or 100 ng ml⁻¹ mouse Wnt3a (315-20 Peprotech).

**Teratoma.** Teratoma formation assays were performed by injecting 1 × 10$^6$ iPS cells into the testes of 7-week-old severe combined immunodeficient (SCID) mice (CB17/SCID; Charles River Laboratories). After 3–4 weeks, the mice were euthanized and lesions were surgically removed and fixed in 4% paraformaldehyde for sectioning and haematoxylin and eosin staining.

**Plasmids and constructs.** pMXS-Oct4, pMXS-Sox2, pMXS-Klf4, pMXS-Myc, pLKO.1 and pWPXLd plasmids were purchased from Addgene. Short hairpin RNAs (shRNAs) against Trp53, FosL1, Atoh8 and Sfrp1 were designed using the MISSION shRNA library from Sigma–Aldrich and ligated using the Rapid DNA ligation kit (Sigma–Aldrich) into the pLKO.1 vector digested with AgeI and EcoRI. Supplementary Table 2 lists the shRNA sequences. Atoh8 and Bcl11b complementary DNA was amplified from MEFs and cloned into the pWPXLd expression vector at the BamH1 restriction site. For Atoh8 ChIP-Seq, AM-Tag was added at the carboxy terminal. Single guide RNA targeting Atoh8 (designed using the CRISPOR program) was cloned into the lentiCRISPRv2 plasmid at a BsmBI restriction site. The single guide RNA sequences are listed in Supplementary Table 2. The pWPIR H-ras G12V and cyclin E plasmids were kindly supplied by the laboratory of A. Puisieux. Tet-O-FUW-Brn2, Tet-O-FUW-Ascl1, Tet-O-FUW-Myt1l, FUW-TetO-Sall4, FUW-TetO-Nanog, FUW-TetO-Esrrb, FUW-TetO-Lin28 and FUdeltaGW-rtTA plasmids were purchased from Addgene.

**Cell culture and viral production.** MEF, mouse adult ear fibroblast and HDF medium consisted of DMEM supplemented with 10% foetal bovine serum (FBS), 100 U ml⁻¹ penicillin–streptomycin, 1 mM sodium pyruvate, 2 mM ʟ-glutamine, 0.1 mM non-essential amino acids (NEAAs) and 0.1 mM β-mercaptoethanol. T lymphocytes from the spleen of Bcl11b conditional KO mice were isolated using the Pan T Cell Isolation Kit (Miltenyi Biotec) according to the manufacturer's instructions after removal of red blood cells by NH₄Cl treatment. T cells were grown in RPMI medium supplemented with 10% FBS, 100 U ml⁻¹ penicillin–streptomycin, 1 mM sodium pyruvate, 2 mM ʟ-glutamine, 0.1 mM NEAAs, 10 mM HEPES, 0.1 mM β-mercaptoethanol, 10 ng ml⁻¹ interleukin-2 and anti-CD3/CD28.

pMXs-based retroviral vectors were generated with Plat-E cells (a retroviral packaging cell line constitutively expressing *gag*, *pol* and *env* genes). Briefly, calcium phosphate transfection of the vectors was performed with the CalPhos Mammalian Transfection kit (Ozyme) in 10-cm dishes. The medium was replaced with 10 ml MEF medium after 7 h of incubation. The lentivirus-containing supernatants were collected 48 h later and stored at −80 °C. 293FT cells, grown in MEF medium, were used to produce lentiviral particles. The vectors were transfected along with plasmids encoding the envelope G glycoprotein of the vesicular stomatitis virus and Gag-Pol.

**Pluripotent reprogramming.** For doxycycline-induced reprogramming, reprogrammable R26$^{rtTA}$;Col1a1$^{4F2A}$;Pou5f1-EGFP MEFs within three passages were plated in six-well plates at 80,000–100,000 cells per well in MEF medium. The following day, cells were infected overnight with shRNA- or single guide RNA (sgRNA)-carrying lentiviral stocks in the presence of 8 μg ml⁻¹ polybrene. The medium was then replaced with fresh medium with 2 μg ml⁻¹ doxycycline. MEFs were reseeded 72 h after infection on 0.1% gelatin-coated plates in iPSC medium (DMEM containing 15% KnockOut Serum Replacement, 1,000 U ml⁻¹ leukaemia inhibitory factor, 100 U ml⁻¹ penicillin–streptomycin, 1 mM sodium pyruvate, 2 mM ʟ-glutamine, 0.1 mM NEAAs and 0.1 mM β-mercaptoethanol) at equal densities for each condition to normalize the potential effect of differential MEF proliferation on reprogramming efficiency. Several densities were tested (15,000–68,000 cells per cm²). Every day, the medium was either replaced or supplemented with doxycycline-containing fresh medium. Once the iPS colonies were macroscopically visible, Pou5f1-EGFP⁺ colonies were counted under an Axiovert 200M microscope and AP staining was performed using the Leukocyte Alkaline Phosphatase Kit (Sigma–Aldrich). Alternatively, MEFs were co-infected with OSKM retroviral vectors 48 h after lentiviral infections and cultured identically

thereafter. SNEL infection (Sall4, Nanog, Esrrb and Lin28) or OSK + IWP2 (2 μM for the first 3 d of reprogramming) were also alternatively used to induce pluripotent reprogramming.

For human pluripotent reprogramming, HDFs (Sigma–Aldrich) were cultivated in MEF medium and infected with lentiviral sgRNA particles in the presence of 8 μg ml⁻¹ polybrene. The following day, the medium was replaced with fresh medium. Two days after sgRNA infection, HDFs were infected with OSKM Sendai particles (CytoTune-iPS Sendai Reprogramming Kit, Life technologies) and the medium was replaced with fresh medium the following day and every other day until day 9. After 9 d, the cells were split onto vitronectin and the medium was changed to mTeSR medium (STEMCELL Technologies). After approximately 26 d, the colonies were SSEA4 live stained (GloLIVE Human Pluripotent Stem Cell Live Cell Imaging Kit, R&D Systems) and counted under an Axiovert 200M microscope. Alternatively, AP staining was performed with the Leukocyte Alkaline Phosphatase Kit (Sigma–Aldrich).

**Oncogenic transformation.** For oncogenic transformation, the LSL-K-ras$^{G12D}$;R26-CRE$^{ERT2}$ MEFs were similarly infected overnight with shRNA- or sgRNA-carrying lentiviral stocks in the presence of 8 μg ml⁻¹ polybrene. After 48 h, the cells were co-infected overnight with sh*Trp53*- and *Myc*-carrying viruses concomitantly with 4-hydroxitamoxifen treatment (1 μM) to induce K-ras$^{G12D}$ expression. Alternatively, the co-infection of sh*Trp53*-, *Myc*- and H-ras$^{G12V}$-carrying viruses was used in wild-type MEFs to initiate transformation. MEFs were reseeded 48 h post-infection in six-well plates at low density (500, 1,000 or 2,000 cells per well) in focus medium (MEF medium with 5% FBS) for the foci formation assay. The medium was then changed twice a week. After several passages of the cells derived from oncogenic transformation, soft agar assays were performed. Transformed cells were plated on an agarose-containing MEF medium layer at a density of 25,000–50,000 cells per six-well plate. Foci and soft agar colonies were stained 25–30 d later with a 0.5% cresyl violet solution in 20% methanol. sh*Trp53*, H-ras$^{G12V}$ and cyclin E-expressing plasmids were also alternatively used in different combinations to induce oncogenic transformation.

**Xenografts.** Some 3 × 10$^6$ transformed cells were prepared in 100 μl PBS supplemented with 100 μl Matrigel and injected subcutaneously into immunocompromised SCID mice (*n* = 6 for each group). The volume of the tumour was then measured every 3 d until day 16.

**Chick chorioallantoic membrane assay.** Some 2.5 × 10$^6$ transformed cells were inoculated on the chorioallantoic membrane in the eggs of chick embryos at E11, where they formed a primary tumour. The size of the tumour was evaluated after 7 d. Numbers of replicates are indicated in the figure captions.

**MEF-to-neuron transdifferentiation.** Wild-type MEFs were co-infected with FUdeltaGW-rtTA and shRNA or sgRNA (control or targeting Atoh8 or Bcl11b) lentiviral plasmids at day −2 in the presence of 8 μg ml⁻¹ polybrene. At day 0, the cells were co-infected with Tet-O-FUW-Brn2, -Ascl1 and -Myt1l lentiviral plasmids. The day after, the medium was replaced with fresh MEF medium supplemented with 2 μg ml⁻¹ doxycycline. At day 3, the medium was replaced with fresh N3 medium consisting of DMEM/F12, 100 U ml⁻¹ penicillin–streptomycin, 2.5 μg ml⁻¹ insulin, 50 μg ml⁻¹ apo-transferrin, 86.5 μg ml⁻¹ sodium selenite, 6.4 ng ml⁻¹ progesterone and 16 μg ml⁻¹ putrescine supplemented with 2 μg ml⁻¹ doxycycline. The medium was changed daily until day 7–8.

**T lymphocyte pluripotent reprogramming.** T cells were infected with OSKM retroviral vectors in the presence of 8 μg ml⁻¹ polybrene the day after isolation for two consecutive days. At 4 h after infection, the medium was replaced with fresh T cell medium. At 3 d after the second infection, the cells were plated onto irradiated MEFs. The day after, the medium was replaced with iPSC medium supplemented with 10 ng ml⁻¹ interleukin-2 and Dynabeads Human T-Activator CD3/CD28 (Life Technologies). The medium was changed every other day.

**Immunofluorescence.** Cells were fixed with 4% paraformaldehyde for 10 min at room temperature, washed three times with PBS, permeabilized with 0.1% Triton X-100 for 30 min at room temperature and blocked with 1% bovine serum albumin for 1 h. After incubation with primary antibodies overnight at 4 °C, the cells were washed three times with PBS and incubated with fluorophore-labelled appropriate secondary antibodies (Life Technologies). Live SSEA4 immunostaining was carried out with the GloLIVE Human Pluripotent Stem Cell Live Cell Imaging Kit (SC023B; R&D Systems). Acquisition was done with Axiovision 4.8.2 software. Supplementary Table 2 lists the antibody dilutions and secondary antibodies used.

**RNA extraction and real-time quantitative PCR.** Total RNAs were extracted using TRIzol reagent and 1 μg RNA was reverse transcribed with the RevertAid H Minus First Strand cDNA Synthesis Kit (Life Technologies). The quantitative PCR (qPCR) was performed with the LightCycler 480 SYBR Green I Master mix (Roche) on the LightCycler 96 machine (Roche) and LightCycler 4.1 software. *Gapdh* and *Rplp0* were used as housekeeping genes. The qPCR primers are listed in Supplementary Table 2.

**Chromatin immunoprecipitation.** MEFs were infected with lentiviral particles carrying AM-tagged Atoh8. After 3 d, DNA was extracted, precipitated and purified using the Tag-ChIP-IT kit (53022; Active Motif). qPCRs were performed as described above. Bcl11b ChIP-Seq was performed using a combination of two antibodies on MEFs exogenously expressing Bcl11b cDNA.

**Protein extraction and western blot.** Cells were harvested in RIPA buffer (150 mM NaCl, 1% Triton, 0.5% deoxycholate, 0.1% SDS and 50 mM Tris (pH 8.0)) supplemented with protease inhibitors and phosphatase inhibitors. After 30 min on ice, lysis by sonication and then centrifugation for 10 min at 15,000g, supernatants were collected and proteins were denatured for 10 min at 95 °C in Laemmli sample buffer, separated on 4–15% polyacrylamide gel and transferred onto a nitrocellulose membrane. The membrane was blocked with 5% milk in Tris-buffered saline with 0.1% Tween 20 for 1 h then incubated with primary antibody at 4 °C overnight and secondary antibodies for 1 h at room temperature. Antigens were detected using ECL reagents. Data were acquired using Bio-Rad Image Lab Software. Band intensities were quantified using ImageJ and normalized to actin levels. The western blot antibodies are listed in Supplementary Table 2.

**FACS.** The following antibody was used: anti-mouse CD90.2 (Thy-1.2) APC (17-0902; eBioscience). Analysis was performed on a BD LSRFortessa with the FACSDiva version 8.0 and FlowJo version 10 software. Sorting was performed on a BD FACSAria. Apoptosis was measured using the FITC Annexin V/Dead Cell Apoptosis Kit (V13242; Invitrogen). For cell cycle analysis, the cells were fixed in 70% ethanol and stained with 40 µg ml⁻¹ propidium iodide supplemented with 2 mg ml⁻¹ RNase. The antibodies used in the study are listed in Supplementary Table 2.

**Next-generation sequencing analyses.** For bulk RNA-Seq, RNA quality was analysed using a 2100 Bioanalyzer (Agilent). RNA-Seq libraries were constructed and sequenced on an Illumina HiSeq 2000 by the cancer genomics platform on site. Fastq files were quality control checked with FASTQC (version 0.11.5). Reads from fastq files were mapped to a reference genome (GRCm38; Gencode) with STAR. The aligned reads were then converted to counts with STAR (version 2.5.2b). RNA-Seq analyses were done with the DESeq2 (version 1.30.1) package in R (version 4.0.3). For the MEF identity score calculation, we computed the fragments per kilobase of transcript per million mapped reads gene values with the DESeq2 package. Then, we used ssGSEA (GSVA R package version 1.44.0) analysis to generated the MEF identity score. Bulk ATAC-Seq and ChIP-Seq data were generated by the Active Motif company. For native Bcl11b ChIP-Seq, a combination of two antibodies was used: ab18465 (Abcam) and A300-385A (Bethyl Laboratories). For scRNA-Seq, cells were resuspended in PBS with 0.04% bovine serum albumin and the number of live cells was determined with a NucleoCounter NC-3000 (ChemoMetec) to obtain an expected cell recovery population of 5,000 cells per channel, loaded on a 10X chip and run on the Chromium Controller system (10X Genomics) according to the manufacturer's instructions. scRNA-Seq libraries were generated with the Chromium Single Cell 3′ v3.1 assay (PN-1000121; 10X Genomics) and sequenced on the NovaSeq 6000 platform (S2 flowcell; Illumina) to obtain around 60,000 reads per cell. The Cell-Ranger Single-Cell Software Suite (version 3.0.2) was used to perform sample demultiplexing, alignment to the mouse genome, barcode assignment for each cell and gene counting by unique molecular identifier counts. Standard procedures for filtering, normalization, variable gene selection, dimensionality reduction and clustering were performed using R software version 4.0.3 (R packages SingleCellExperiment version 1.12.0 (ref. [65]), scater version 1.18.6 (ref. [66]) and scran version 1.18.5 (ref. [67])). Cells having <2,500 genes or a mitochondrial content >15% were excluded from the analysis. Counts were log normalized. The 1,500 most variable genes were used to reduce the dimensionality of the dataset by PCA. Based on the plot of variance explained, we kept the first six principal components for further analyses and summarized them using t-SNE. Clustering was conducted using a shared nearest neighbours graph (the buildSNNGraph function of the R package scran). Cluster-specific markers were computed with a t-test from the finderMarkers() function of the scran package[67], where the combined P value of a gene is the maximum of all P values from all pairwise comparisons (pval = "all"). Single-cell pseudotime trajectories were constructed with Slingshot[36] and temporally expressed genes were identified using a general additive model (R package gam version 1.20). To create diffusion maps of the data, we used the function runDiffusionMap of the R package scater. Single-sample gene set enrichment activity scores (ssSSEA[68]) of the pathways were computed with the GSVA[37] R package version 1.38.2. Calculation of the MEF identity score was conducted using two independent signatures from refs. [12,13]. Some figures have been created using the Cerebro visualization tool version 1.3 (ref. [69]).

For ATAC-Seq unsupervised hierarchical clustering, we used the pheatmap package to create a heatmap from the differentially accessible regions used to form the six clusters described above. Peak intensity labels are scaled row-wise and hierarchical clustering was performed using complete clustering. We have annotated each peak with the assigned cluster label for reference.

Atoh8 ChIP-Seq and ATAC-Seq datasets were aligned to the mouse reference genome assembly mm10 using Bowtie 2.1.0 using default parameters. Peak calling was performed using MACS 2.1.1. For experiments with replicates, BED files

obtained after alignment were concatenated before MACS peak calling processing. Atoh8-specific sites were obtained by subtracting binding sites observed within the AM-tag processed control dataset (BEDTools 2.29.2). Enrichment heatmaps and mean density plots were obtained with seqMINER version 1.3.4 (ref. [70]). De novo motif analysis was performed with MEME-ChIP (MEME Suite version 5.4.1)[71]. Read count enrichment signals were visualized with the IGV genome browser (version 2.4.15). Atoh8-centred chromatin state analysis was performed by intersecting Atoh8-specific binding sites with those associated with public data[9,72] then inferring co-occuring events with ChromHMM version 1.14 (ref. [73]).

**Statistics and reproducibility.** No statistical methods were used to predetermine sample sizes but our sample sizes are similar to those reported in previous publications[74]. Data distribution was assumed to be normal but this was not formally tested. No randomization was used. Data collection and analysis were not performed blind to the conditions of the experiments. For the single-cell experiments presented in Figs. 1 and 2, two biological replicates were conducted and run together in a single sequencing experiment. Western blot quantifications were performed with ImageJ. Statistical analyses of mean and variance were performed with Prism 8 (GraphPad Software) and the statistical tests are indicated. No data points were excluded. For western blots, three independent experiments gave similar results.

**Reporting summary.** Further information on research design is available in the Nature Research Reporting Summary linked to this article.

## Data availability

Sequencing data that support the findings of this study have been deposited in the Gene Expression Omnibus under accession code GSE137050. Previously published data that were re-analysed here are available under accession codes GSE90895, GSE10871, GSE11074, GSE122662 and GSE62777 (Gene Expression Omnibus) and SRP046744 and SRP119979 (Sequence Read Archive). All other data supporting the findings of this study are available from the corresponding author upon reasonable request. Source data are provided with this paper.

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

## Acknowledgements

We are grateful to A. Lalande for technical assistance. We thank the Cancer Genomic Platform (Centre Léon Bérard, Cancer Research Center of Lyon) for performing the scRNA-Seq. This work was supported by institutional grants from INSERM/CNRS, ATIP-Avenir, Plan Cancer, Labex Dev2Can, Institut Convergence PLAsCAN (ANR-17-CONV-0002) and La Ligue Contre le cancer Nationale et Régionale (to F.L.), Institut National du Cancer (2019-L22 to F.L., M.M.-P., F.S. and H.G.), Fondation ARC (to F.L., G.F. and A.H.), Centre Léon Bérard (to F.L. and A.H.) and Fondation pour la Recherche Médicale (to A.T.). P.M. and M.S. were funded by Institut National du Cancer (PLBIO2016-180) and Fondation pour la Recherche Médicale (AJE20131128936). J.M.P was supported by an ARC Future Fellowship FT180100674 and NHMRC APP1104560.

## Author contributions

A.H. and G.F. performed most of the experiments. N.R., L.T., J.K., J.P., J.S., F.S., R.D., M.M.-P. and H.G. performed the bioinformatics analyses. A.T., M.R., M.L. and Y.B.Y.S. conducted the reprogramming experiments. F.M., L.D.M., A.B., J.W., Y.Y.,

P.W. and P.L. contributed to the generation of data for the study. N.G. and M.B. conducted the histologic analyses of teratoma and tumours. M.S. and P.M. conducted the c-Myc ChIP experiments. E.C. performed most of the FACS analyses. B.G. performed the CAM (chorioallantoic membrane) and xenograft assays. F.L., A.H. and G.F. designed the experiments and wrote the manuscript. F.L. designed and supervised the study. All authors approved of and contributed to the final version of the manuscript.

## Competing interests

The authors declare no competing interests.

## Additional information

**Extended data** is available for this paper at https://doi.org/10.1038/s41556-022-00986-w.

**Correspondence and requests for materials** should be addressed to A. Huyghe or F. Lavial.

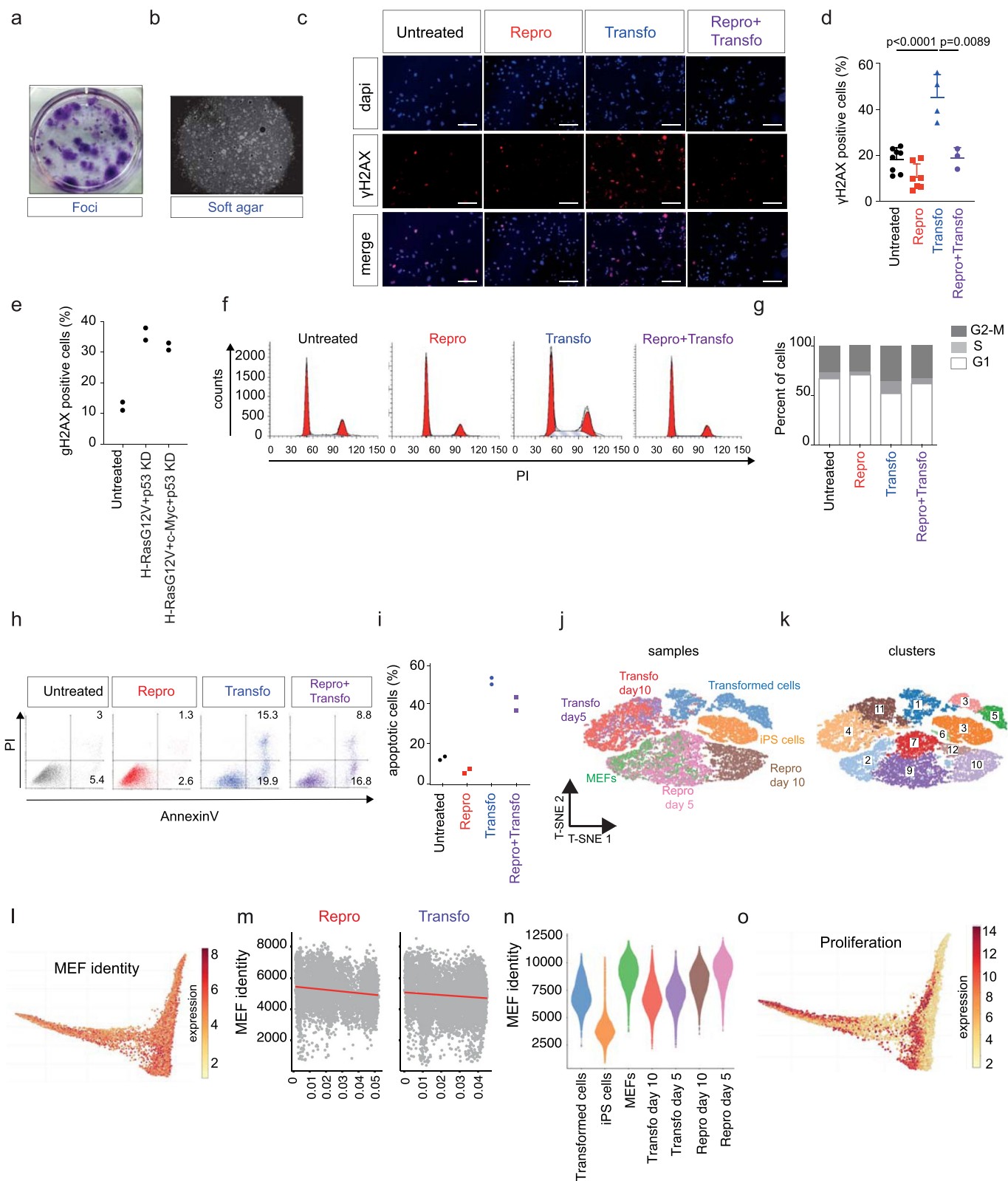

**Extended Data Fig. 1 | See next page for caption.**

**Extended Data Fig. 1 | Comparing single-cell trajectories of reprogramming & transformation.** (a) Foci obtained from repro-transformable MEFs colored with cresyl-violet. (b) Soft agar colonies. (c) Immunofluorescent staining. One representative experiment (from three independent experiments). Scale bar: 100 μm. (d) Counting of γH2AX-positive cells depicted in (c). n = 3 independent experiments. One-way ANOVA followed by a Tukey's post hoc test. (e) Counting of H2AX-positive cells after 3 days of transformation with alternative oncogenic cocktails. n = 2 independent experiments. (f) FACS of cell cycle analysis. (g) Countings. Data presents one experiment representative of three independent experiments. (h) FACS profile PI/AnnexinV after 3 days of Repro, Transfo or Repro+Transfo compared to control MEFs. One representative experiment (from two independent experiments). (i) Percentage of total apoptosis depicted in (h). n = 2 independent experiments. (j-k) T-distributed stochastic neighbor embedding (t-SNE) visualization of sc-RNA-seq profiles (individual dots). Cells are colored by samples (j) or by clusters (k). (l-m) Patterns of MEF identity signature score using gene lists from Nefzger et al., 2017. (l) Score represented on the diffusion map. (m) Score represented on the calculated pseudo-time trajectories. (n) MEF identity score during reprogramming and transformation. The score from Schiebinger et al., 2019 is assessed in MEFs, repro, transfo samples alongside iPS and transformed cells obtained from MEFs. (o) Patterns of gene signature scores on the diffusion map using gene lists.

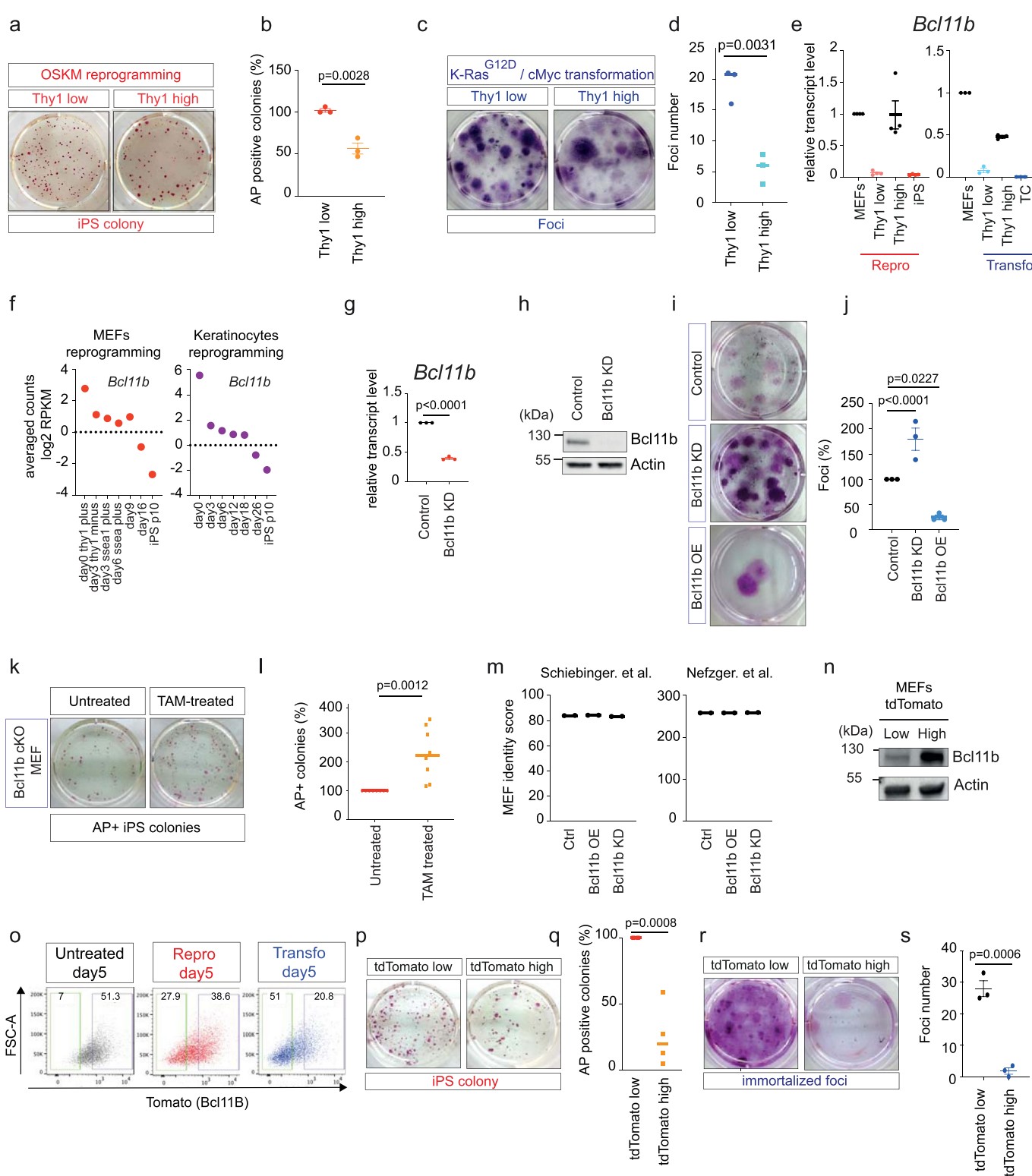

**Extended Data Fig. 2 | See next page for caption.**

**Extended Data Fig. 2 | Bcl11b constrains reprogramming & transformation.** (a) Alkaline Phosphatase (AP) staining of iPS colonies. One representative experiment (from three independent experiments). (b) Counting of AP-positive colonies depicted in (h). Data are the mean ± s.d. (n = 3 independent experiments). Two-tailed student's t-test was used. (c) Foci staining generated from Thy1low and Thy1high cells FACS-sorted at day 5 of Transfo. One representative experiment (from three independent experiments). (d) Counting of foci depicted in (c). Data are the mean ± s.d. (n = 3 independent experiments). Two-tailed student's t-test. (e) Relative transcript level of Bcl11b. Data are the mean ± s.d. (n = 4 independent experiments for left panel and 3 for right panel). (f) Bcl11b expression profiles during MEFs and keratinocytes reprogramming. Data are extracted from Nefzger et al., 2017. (g-h) Bcl11b knockdown efficiency. (g) Q-RTPCR of Bcl11b levels 3 days after infection. Data are the mean +/- sd of 3 independent experiments. Two-tailed student's t-test. (h) Western blot for Bcl11b in similar settings as (g). (i) Representative pictures of foci. (j) Counting of foci depicted in (i). Data are the mean ± s.d. (n = 3 independent experiments). Two-tailed Student's t-test. (k) Pictures of AP + iPS colonies generated from Bcl11b conditional KO MEFs treated or not with Tamoxifen. (l) Countings of colonies depicted in (k). Data are the mean ± s.d. (n = 8 independent experiments). Two-tailed student's t-test. (m) Patterns of MEF identity signature score using gene lists from Nefzger et al., 2017 and Schiebinger et al., 2019. (n) Western blot of Bcl11b in the tdTomato-high/low fractions. (o) FACS analysis of Bc11b-tdTomato. (p) Alkaline phosphatase (AP) staining of iPS colonies generated from tdTomatolow and tdTomatohigh cells. One representative experiment (from four independent experiments). (q) Counting of AP-positive colonies depicted in (p). n = 4 independent experiments. Two-tailed Student's t-test. (r) Foci staining of cells generated from tdTomatolow and tdTomatohigh cells. One representative experiment (from three independent experiments). (s) Counting of foci depicted in (r). Data are the mean ± s.d. (n = 3 independent experiments). Two-tailed Student's t-test.

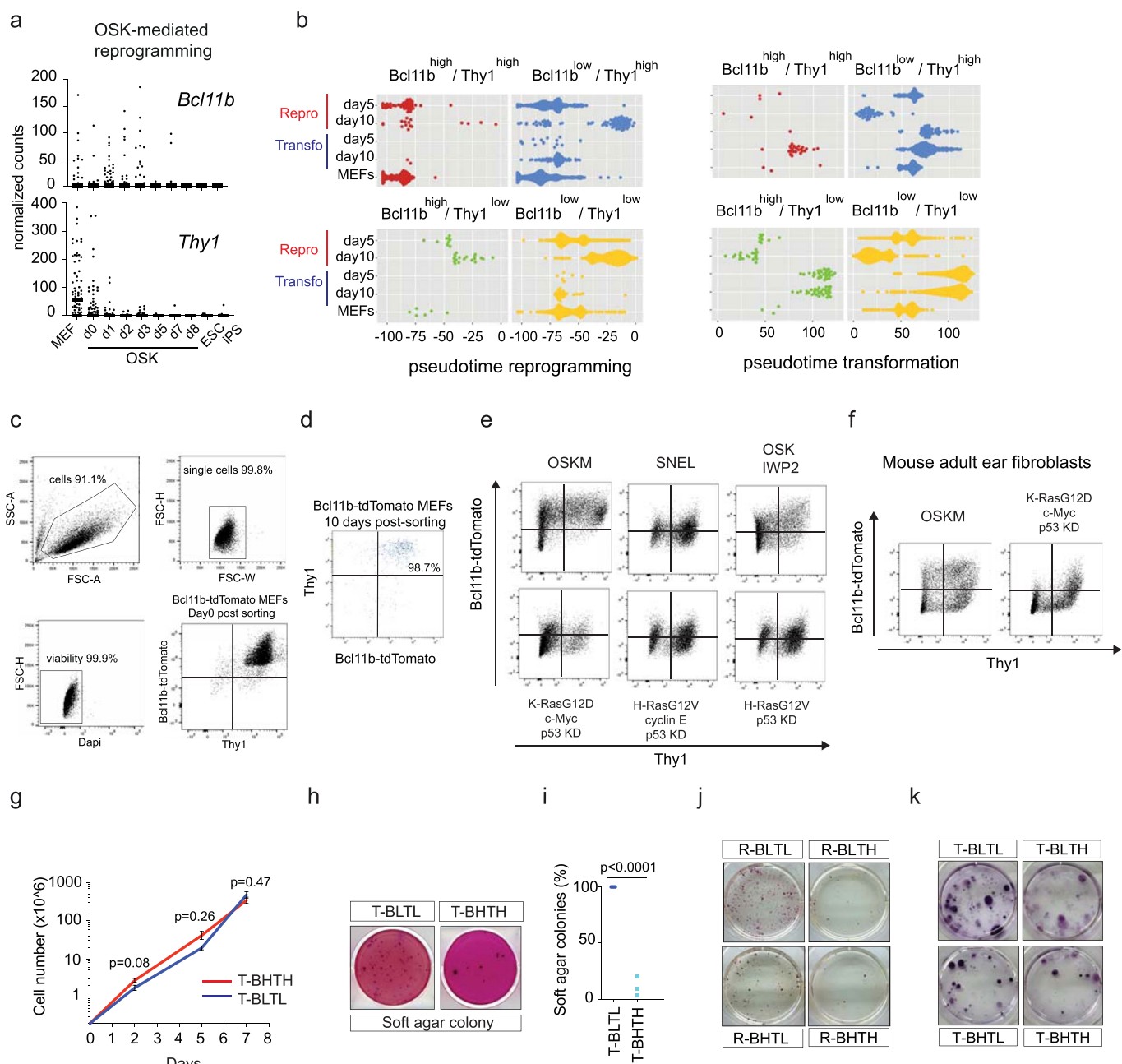

**Extended Data Fig. 3 | Sequence of intermediates during reprogramming & transformation.** (a) Bcl11b and Thy1 expression at the single cell level during OSK-mediated reprogramming in MEFs. Data are extracted from Guo et al., 2019. (b) Graph positioning the cells expressing different levels of Bcl11b and Thy1 on the reprogramming (left) and transformation (right) pseudotime trajectories. Each dot represents one cell. The thresholds are as follow: BLTL: Bcl11b < 1 and Thy1 < 2; BHTL: Bcl11b > 2 and Thy1 < 2; BLTH: Bcl11b < 1 and Thy1 > 4; BHTH: Bcl11b > 2 and Thy1 > 4. (c) Gating strategy for FACS analyses based on Thy1 and Bcl11b expression in MEFs. (d) FACS profile of Bcl11b-tdTomato MEFs FACS sorted for high expression of Bcl11b and Thy1 and re-analysed by FACS 10 days later. (e) Representative FACS plots for the emergence of Bcl11b-tdTomatoLow/Thy1Low (BLTL) cells. Data of one experiment representative of three independent experiments. (f) Representative FACS plots for the emergence of BLTL cells in MAEFs. Data of one experiment representative of three independent experiments. (g) Growth curves of transformed cell lines derived from Bcl11b-tdTomatoHigh/Thy1High (BHTH) and Bcl11b-tdTomatoLow/Thy1Low (BLTL) cells grown in 2D. Data are the mean ± s.d. (n = 3 independent experiments). Two-way ANOVA combined with Sidak multiple comparisons test was used. (h) Soft agar assays. (i) Counting of colonies from (h). n = 3 independent experiments. Two-tailed Student's t-test. (j) AP staining of iPS colonies generated from the different subpopulations FACS-sorted at day 5 of Repro. One representative experiment (from five independent experiments). (k) Foci assays generated from the different subpopulations FACS-sorted at day 5 of Transfo. One representative experiment (from six independent experiments).

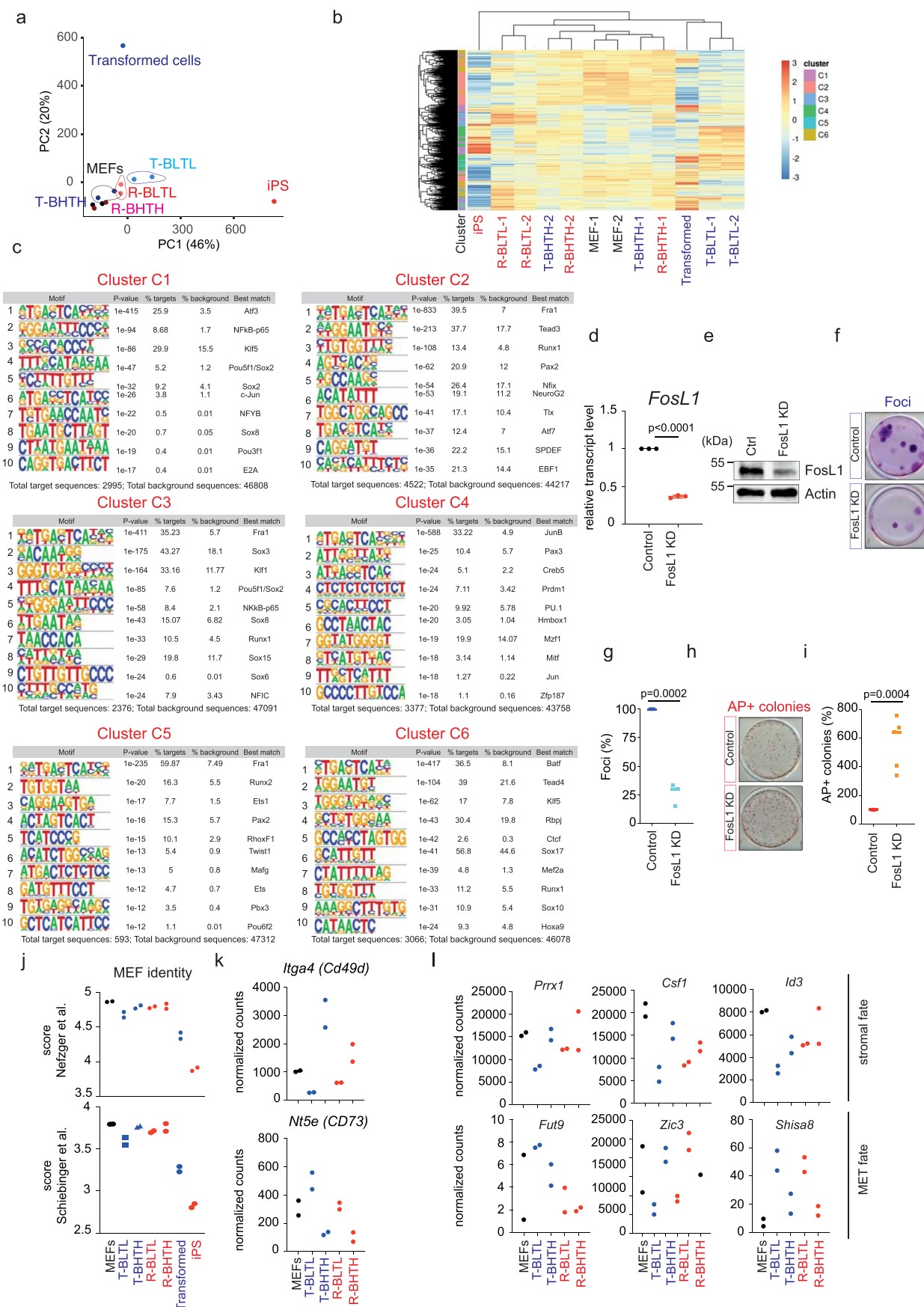

**Extended Data Fig. 4 | See next page for caption.**

**Extended Data Fig. 4 | Chromatin accessibility and transcriptomic changes during reprogramming & transformation.** (a) Principal component analysis of ATAC-seq signal of the control cells, cellular intermediates (BLTL and BHTH) and the final product of each process (iPS and transformed cells). (b) Heatmap of differentially accessible peaks in ATAC-seq data. Peaks are scaled per row and annotated by cluster labels. (c) Top 10 motifs enriched in the ATAC-seq clusters defined in the main text. (d) Q-RTPCR of FosL1 levels. Data, normalized to control, are the mean +/- sd of 3 independent experiments. Two-sided Student T-test. (e) Western blot for FosL1 in similar settings as (d). (f) Foci staining of immortalized cells. One representative experiment (from three independent experiments). (g) Counting of foci depicted in (f). n = 3 independent experiments. Two-tailed Student's t-test. (h) Alkaline phosphatase (AP) staining of iPS colonies. One representative experiment (from five independent experiments). (i) Counting of AP-positive colonies depicted in (h). n = 5 independent experiments. Two-tailed Student's t-test. (j) Ssgsea analysis of the MEF identity score using independent datasets from Nefzger et al., 2017 and Schiebinger et al., 2019. n = 2 independent experiments. (k) Itga4 and Nt5e expression in cellular intermediates. RNA-seq data from Fig. 4 are used. n = 2 independent experiments. (l) Gene expression levels in MEFs and cellular intermediates. Stromal and MET fate genes, as defined in Schiebinger et al., 2019, were analyzed in the RNA-seq dataset. n = 2 independent experiments.

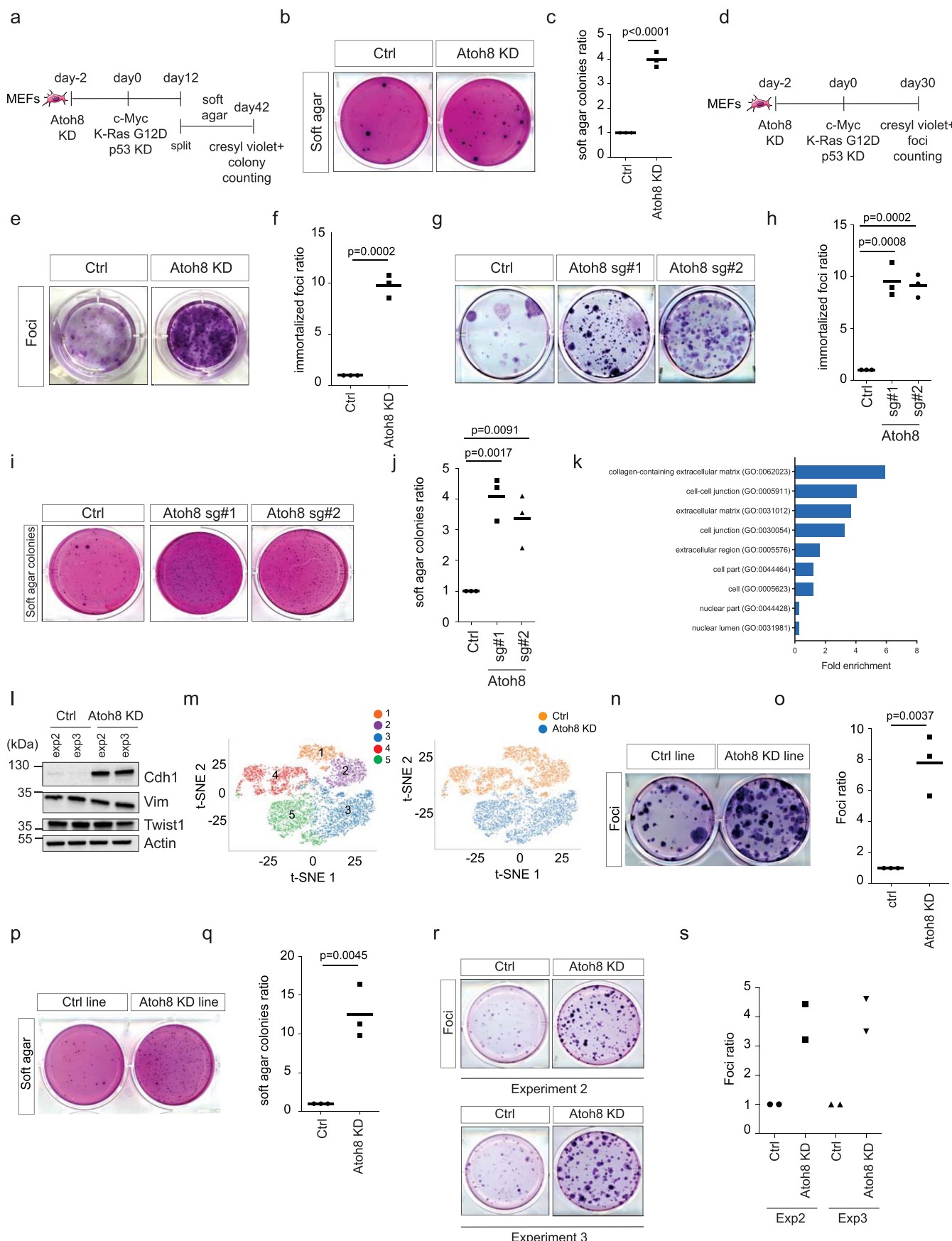

**Extended Data Fig. 5 | See next page for caption.**

**Extended Data Fig. 5 | Atoh8 regulates the acquisition of malignant features.** (a) Scheme of experimental design. (b) Picture representing soft-agar colonies, representative of three independent experiments. (c) Colony counting. Data are the mean ± s.d. (n = 3 independent experiments). Two-tailed student's t-test was used. (d) Experimental design. (e) Picture representing Cresyl-violet foci at day 30 of MEF immortalization, representative of three independent experiments. (f) Colony counting. Data are the mean ± s.d. (n = 3 independent experiments). Two-tailed student's t-test was used. (g) Picture representing Cresyl-violet immortalized foci. (h) Colony counting. Data are the mean ± s.d. (n = 3 independent experiments). Two-tailed student's t-test was used. (i) Picture representing Cresyl-violet transformed soft-agar colonies. (j) Colony counting. Data are the mean ± s.d. (n = 3 independent experiments). Two-tailed student's t-test was used. (k) Statistical overrepresentation analysis. (l) Western blot depicting Cdh1 induction during oncogenic transformation in absence of Atoh8. (m) t-SNE representation of single cell RNA-seq data. (n) Picture representing Cresyl-violet immortalized foci at day 30 of foci formation assay. (o) Colony counting. Data are the mean ± s.d. (n = 3 independent experiments). Two-tailed student's t-test was used. (p) Picture representing Cresyl-violet transformed colonies at day 30 of soft-agar assay. (q) Colony counting. Data are the mean ± s.d. (n = 3 independent experiments). Two-tailed student's t-test was used. (r) Picture representing Cresyl-violet immortalized foci at day 30 of foci formation assay starting from cell lines established from 2 independent experiments in control- or Atoh8-KD settings. (s) Colony counting. Data are the mean ± s.d. (n = 2 independent experiments).

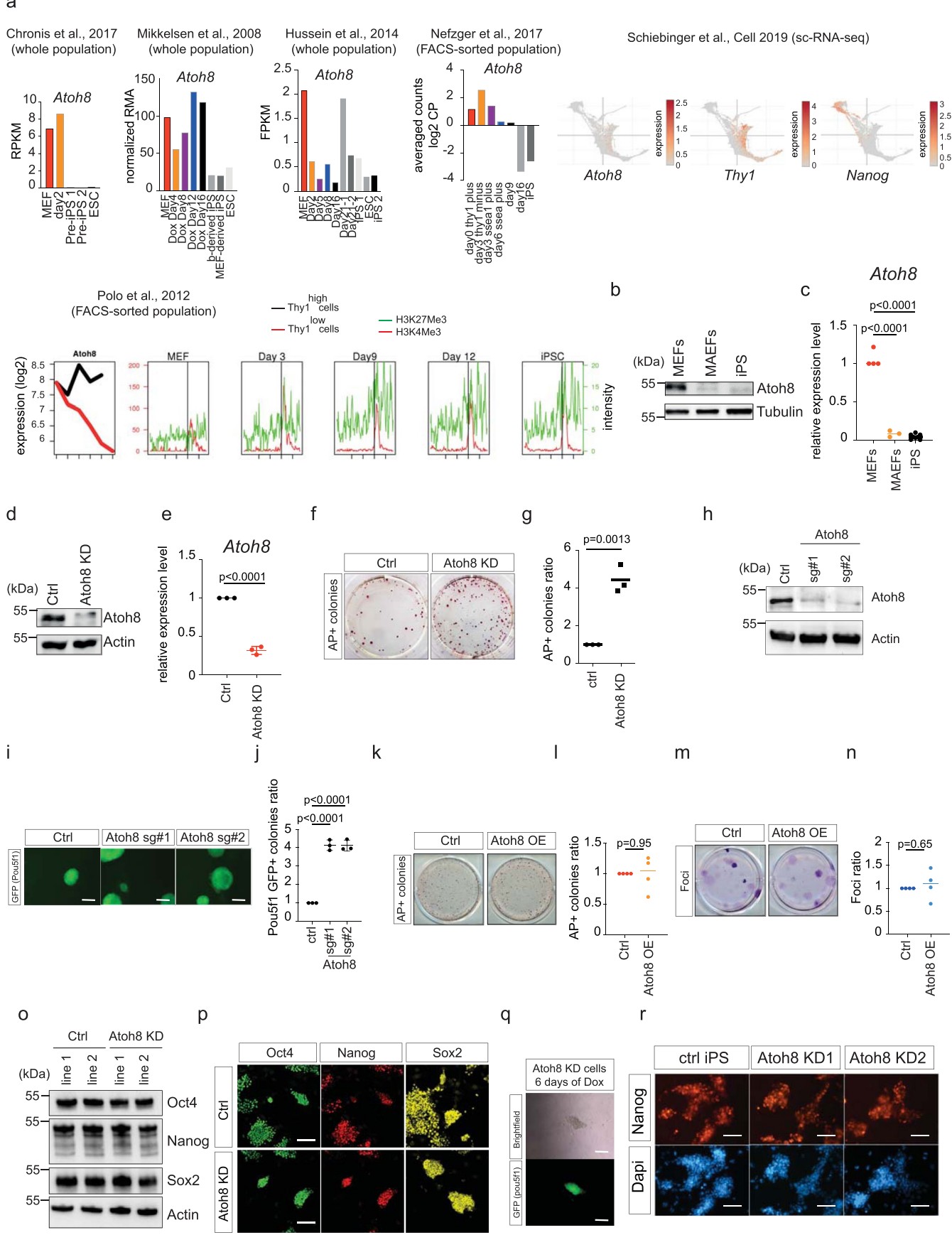

**Extended Data Fig. 6 | See next page for caption.**

**Extended Data Fig. 6 | Atoh8 constrains MEF reprogramming.** (a) The graphs depict Atoh8 expression levels during MEF reprogramming and iPS/ES cells from six independent datasets. For the sc-RNA-seq data from Schiebinger et al., 2019, Thy1 and Nanog expression are shown to indicate the location of MEFs and iPS cells. ChIP-seq data for H3K4me3 (red) and H3K27me3 (green) methylation in MEFs and during iPS cells generation are also presented. (b) Western blot. (c) Q-RTPCR. (d) Western blot of Atoh8 level. (e) Q-RTPCR of Atoh8 levels in similar settings as (d). Data, normalized to control, are the mean +/- sd of 3 independent experiments. Two-tailed student T-test. (f) Picture representing AP + colonies, representative of three independent experiments. (g) AP + iPS colony counting. Data are the mean ± s.d. (n = 3 independent experiments). Two-tailed student T-test. (h) Western blot showing Atoh8 expression level following KO with 2 independent guides (sg#1 and sg#2). (i) Picture representing Pou5f1-GFP + iPS colonies at day 15 of Repro, representative of three independent experiments. Scale bars=200 μm. (j) Pou5f1-GFP + colony counting. Data are the mean ± s.d. (n = 3 independent experiments). Two-tailed student T-test. (k) Picture representing AP + colonies, representative of four independent experiments. (l) AP + iPS colony counting. Data are the mean ± s.d. (n = 4 independent experiments). Two-tailed student T-test. (m) Picture representing foci, representative of four independent experiments. (n) Foci counting. Data are the mean ± s.d. (n = 4 independent experiments). Two-tailed student T-test. (o) Western blot. (p) Immunofluorescence. Scale bar: 100 μm. (q) Brightfield and GFP images showing Pou5f1-GFP + iPS colonies observed after 6 days of Dox treatment. Scale bars=200 μm. (r) Nanog immunofluorescence in bona fide and Atoh8-KD iPS independent lines. Scale bars=500 μm.

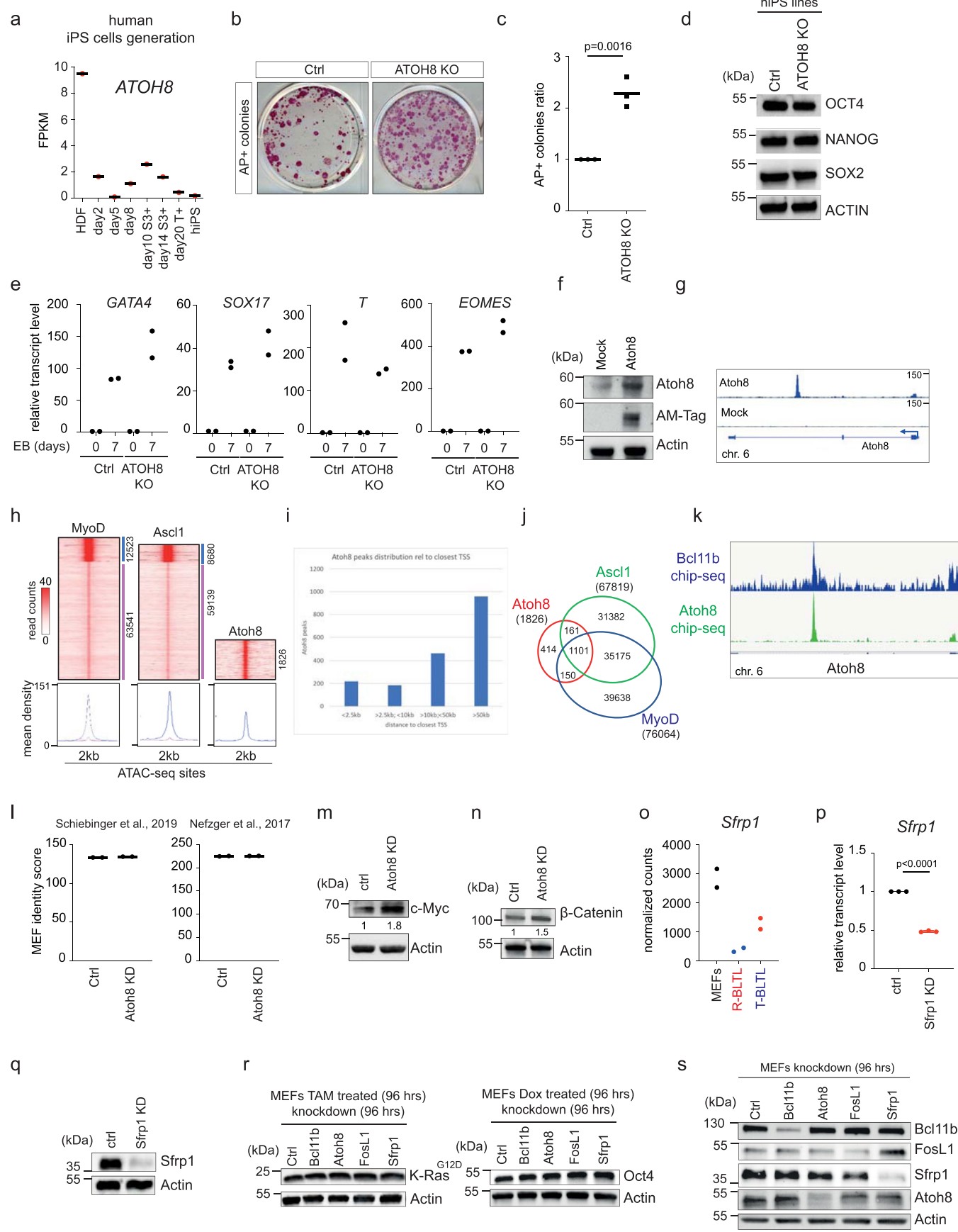

**Extended Data Fig. 7 | See next page for caption.**

**Extended Data Fig. 7 | Atoh8 fine tunes WNT signalling via Sfrp1.** (a) ATOH8 transcript levels in human dermal fibroblasts (HDF), reprogramming intermediates and human iPS cells. Data present transcripts level in log2 FPKM. S3+=SSEA3 positive cells; T+=TRA-1-60 positive cells. (b) Picture representing human AP+ colonies at day 26 of Reprogramming, representative of three independent experiments. (c) Colony counting. Data are the mean ± s.d. (n=3 independent experiments). Two-tailed student's t-test. (d) Western blot. (e) q-RTPCR of transcript levels during EB formation. Data, normalized to day 0 of differentiation. N=2 independent experiments. (f) Western blot depicting Atoh8 and AM-Tag detection following the infection of MEF with Mock or Atoh8-AM flagged particles. (g) Genome browser track showing Atoh8 binding within its own intronic region. (h) Heat map displaying the enrichment in MyoD, Ascl1 or Atoh8 within chromatin accessible sites (ATAC-seq). (i) Atoh8 peaks distribution relative to adjacent TSS. TSS: Transcription start site. (j) Venn Diagram. (k) Example of Bcl11b and Atoh8 co-binding on Atoh8 locus. (l) Ssgsea analysis based on the 2 independent MEF identity scores depicted in Nefzger et al., 2017 and Schiebinger et al., 2019. n=2 independent samples. (m) Western blot. (n) Western blot. (o) Sfrp1 transcript level in MEFs and BLTL cells. RNA-seq data from Fig. 4 are used. (p) Q-RTPCR showing Sfrp1 transcript levels after RNAi. Data, normalized to control, are the mean +/- sd of 3 independent experiments. Two-tailed student's t-test. (q) Western blot. (r) Western blots. Left panel: K-RasG12D expression was induced in MEFs by Tamox treatment. Infection with ctrl or gene-targeting sequences was performed on the same day. Right panel. Oct4 expression was induced in MEFs by Dox treatment. Similar settings as left panel for the depletion. (s) Effect of candidate on each other. Similar settings as (r).

# Reporting Summary

## Statistics

For all statistical analyses, confirm that the following items are present in the figure legend, table legend, main text, or Methods section.

| n/a | Confirmed | |
|---|---|---|
| ☐ | ☒ | The exact sample size (*n*) for each experimental group/condition, given as a discrete number and unit of measurement |
| ☐ | ☒ | A statement on whether measurements were taken from distinct samples or whether the same sample was measured repeatedly |
| ☐ | ☒ | The statistical test(s) used AND whether they are one- or two-sided *Only common tests should be described solely by name; describe more complex techniques in the Methods section.* |
| ☒ | ☐ | A description of all covariates tested |
| ☒ | ☐ | A description of any assumptions or corrections, such as tests of normality and adjustment for multiple comparisons |
| ☐ | ☒ | A full description of the statistical parameters including central tendency (e.g. means) or other basic estimates (e.g. regression coefficient) AND variation (e.g. standard deviation) or associated estimates of uncertainty (e.g. confidence intervals) |
| ☐ | ☒ | For null hypothesis testing, the test statistic (e.g. *F*, *t*, *r*) with confidence intervals, effect sizes, degrees of freedom and *P* value noted *Give P values as exact values whenever suitable.* |
| ☒ | ☐ | For Bayesian analysis, information on the choice of priors and Markov chain Monte Carlo settings |
| ☐ | ☒ | For hierarchical and complex designs, identification of the appropriate level for tests and full reporting of outcomes |
| ☒ | ☐ | Estimates of effect sizes (e.g. Cohen's *d*, Pearson's *r*), indicating how they were calculated |

*Our web collection on statistics for biologists contains articles on many of the points above.*

## Software and code

Policy information about availability of computer code

| Data collection | Western blot acquisition was performed using a Image Lab 6.0.1 Bio-Rad software. Q-RTPCR data were acquired using the Lightcycler 4.1 software. Microscopy images were acquired with axiovision 4.8.2 software. FACS data were acquired on a FACSDiva v8.0 software. |
|---|---|
| Data analysis | All the analyses conducted are described in the methods section. Quantification of western blot data was performed using ImageJ 1.53r. Most of the data analysis and statistics were performed using Graphpad prism 8. Next-generation sequencing data were analysed using FASTQC (v0.11.5), STAR (v2.5.2b), DESeq2 (v1.30.1) package in R (v4.0.3), GSVA R-package v1.44.0, Cellranger Single-Cell Software Suite (v3.0.2), Cerebro visualization tool v1.3, Bowtie 2.1.0, MACS 2.1.1, seqMINER v1.3.4., MEME Suite v5.4.1, IGV genome browser (v2.4.15), ChromHMM v1.14. |

For manuscripts utilizing custom algorithms or software that are central to the research but not yet described in published literature, software must be made available to editors and reviewers. We strongly encourage code deposition in a community repository (e.g. GitHub). See the Nature Portfolio guidelines for submitting code & software for further information.

## Data

Policy information about availability of data

All manuscripts must include a data availability statement. This statement should provide the following information, where applicable:
- Accession codes, unique identifiers, or web links for publicly available datasets
- A description of any restrictions on data availability
- For clinical datasets or third party data, please ensure that the statement adheres to our policy

Data availability
NGS data were deposited on GEO (record number series GSE137050).

# Field-specific reporting

Please select the one below that is the best fit for your research. If you are not sure, read the appropriate sections before making your selection.

☒ Life sciences          ☐ Behavioural & social sciences          ☐ Ecological, evolutionary & environmental sciences

For a reference copy of the document with all sections, see nature.com/documents/nr-reporting-summary-flat.pdf

# Life sciences study design

All studies must disclose on these points even when the disclosure is negative.

| | |
|---|---|
| Sample size | Sample sizes were determined without statistical measures, but based on prior experience with the specific experiments and widely used sizes in relevant publications (Polo. et al., 2012, Ozmadenci D. et al., 2015) within this field of research in order to ensure that it will be appropriate for statistical analysis. See Figures legends for each experiment. |
| Data exclusions | No data exclusion was used in the study. |
| Replication | The vast majority of the presented results come from three independent biological replicates and some results from two independent biological replicates. Next-generation-sequencing data are coming from 2 or 3 independent biological replicates. In all cases, all attempts at replication were successful. |
| Randomization | No randomization was used in the study. |
| Blinding | Experiments execution, data collection and result analysis were carried out by the same researcher, therefore no blinding was used. |

# Reporting for specific materials, systems and methods

We require information from authors about some types of materials, experimental systems and methods used in many studies. Here, indicate whether each material, system or method listed is relevant to your study. If you are not sure if a list item applies to your research, read the appropriate section before selecting a response.

| Materials & experimental systems | | | Methods | |
|---|---|---|---|---|
| n/a | Involved in the study | | n/a | Involved in the study |
| ☐ | ☒ Antibodies | | ☐ | ☒ ChIP-seq |
| ☐ | ☒ Eukaryotic cell lines | | ☐ | ☒ Flow cytometry |
| ☒ | ☐ Palaeontology and archaeology | | ☒ | ☐ MRI-based neuroimaging |
| ☐ | ☒ Animals and other organisms | | | |
| ☒ | ☐ Human research participants | | | |
| ☒ | ☐ Clinical data | | | |
| ☒ | ☐ Dual use research of concern | | | |

## Antibodies

| | |
|---|---|
| Antibodies used | The following antibodies were used: Rat anti-mouse Thy1 (17090283, Invitrogen, 1:1000), mouse anti-OCT4 (sc-5279, Santa Cruz, 1:1000), rabbit anti-SOX2 (ab97959, Abcam, 1:1000, mouse anti-c-MYC (sc-42, Santa-Cruz, 1:200), rabbit-anti ATOH8 (PA5-20710, Termofisher, 1:1000), rabbit anti-ID4 (BCH-9/82-12, BioCheck, 1:1000), MAP2 (Sigma Aldrich, M4403), rabbit anti-NANOG (RCAB002P, Reprocell, 1:1000), mouse anti-SSEA1 (sc-101462, Santa Cruz, 1:1000), mouse AM-Tag (active motif 9111), mouse anti-CDH1 (610181, BD, 1:1000), rabbit anti-SNAIL (3895S, Cell signaling, 1:1000), rabbit anti-VIM (R28, Cell signaling, 1:1000), mouse anti-TWIST1 (ab50887, Abcam, 1:250), goat anti-hSOX2 (AF2018, R&D, 1:1000), rabbit anti-hNANOG (3580, Cell signaling, 1:1000), mouse anti-ACTIVE β-CATENIN (8E7, Millipore, 1:1000), rabbit anti-GAPDH (sc-25778, Santa-Cruz, 1:4000),rat anti-BCL11B (Abcam, ab18465, 1:500), horse radish peroxidase (HRP)-conjugated anti-ACTIN (Sigma Aldrich, A3854, 1:10,000), anti-TWIST2 (Abnova H007581-MO1, 1:1000), anti-FOSL1 (Santa Cruz sc376148, 1:1000), anti-b-Tubulin (Sigma T5293, 1:1000), anti-Sfrp1 (Abcam ab267466, 1:1000), anti-K-RasG12D (Cell signaling 8955, 1:1000), anti-Bcl11b (Bethyl A300-385A), phospho-Histone H2A.X (Cell Signaling Technology 2577). Live SSEA4 immunostaining was carried out with the GloLIVE Human Pluripotent Stem Cell Live Cell Imaging Kit (SC023B, R&D). |
| Validation | The information of all antibodies we used can be found in their official website. |

# Eukaryotic cell lines

Policy information about cell lines

| | |
|---|---|
| Cell line source(s) | All the cell lines sources can be found in the methods. Primary MEFs were derived from E13.5 embryos in the Lavial lab, as previously described (Ozmadenci D. et al., 2015). MAEFs were derived from adult mice at 6-8 weeks old. Mouse iPS cells were derived from MEFs following exposure to OSKM expression (dox-inducible or retroviral, as indicated). Transformed cells (TC) were generated by the serial passaging of MEFs induced to express the indicated oncogenes. Human dermal fibroblasts were pruchased from Sigma (Merck 106-05A). Primary T cells were derived from the spleen and lymph nodes of adult mice (6-8 weeks old). 293T cells were purchased from ATCC (CRL-1573) and Plat-E cells from Cellbiolabs (RV-101). |
| Authentication | The differentiation potential of the mouse iPS cell lines was authenticated by performing in vivo differentiation in teratoma assays. The differentiation ability of human iPS cell lines was evaluated in vitro by embryoid body formation assay. The malignant fetaures of TC cells was evaluated in vitro by soft agar assays and in vivo by conducting mouse xenograft and chick chorioallantoic membrane assays. The other cell lines were not authenticated. |
| Mycoplasma contamination | All cells are tested negative for mycoplasma contamination using MycoAlert Mycoplasma detection kit from Lonza. |
| Commonly misidentified lines (See ICLAC register) | None of the cell lines used in this study is listed in the database of commonly misidentified cell lines maintained by ICLAC. |

# Animals and other organisms

Policy information about studies involving animals; ARRIVE guidelines recommended for reporting animal research

| | |
|---|---|
| Laboratory animals | Laboratory animals used in the study are described in the methods. MEFs were derived from E13.5 embryos of repro-transformable C57Bl mice models. Pou5f1-GFP C57Bl mice were also used. Teratoma assays were performed with 7-week-old severe combined immunodeficient (SCID) female mice (CB17/SCID, Charles River). Xenograft assays were performed with 7-week old immunocompromised SCID mice. CAM assays were conducted with E11 chicken embryos (local supplier élevage avicole du grand buisson). |
| Wild animals | The study did not involve wild animals. |
| Field-collected samples | The study did not involve samples collected from the field. |
| Ethics oversight | Animals were maintained in a specific pathogen free (SPF) animal facility P-PAC at the Cancer Research Center of Lyon (CRCL) at the Center Léon Bérard (CLB), Lyon, France. All the experiments were performed in accordance with the animal care guidelines of the European Union and were validated by the local Animal Ethic Evaluation Committee (C2EA 15 agreed by the French Ministry of High Scool and Research). |

Note that full information on the approval of the study protocol must also be provided in the manuscript.

# ChIP-seq

## Data deposition

☒ Confirm that both raw and final processed data have been deposited in a public database such as GEO.

☒ Confirm that you have deposited or provided access to graph files (e.g. BED files) for the called peaks.

| | |
|---|---|
| Data access links *May remain private before publication.* | Record number series GSE137050, secure token for reviewers access yvmpscquvhybzgl. |
| Files in database submission | Raw and processed data files: Input-Atoh8-rep1 AM-empty-rep1 AM-Atoh8-rep1 Input-Atoh8-rep2 AM-empty-rep2 AM-Atoh8-rep2 Input-Bcl11b-rep1 Bcl11b-rep1 Bcl11b-rep2 |
| Genome browser session (e.g. UCSC) | No |

## Methodology

| | |
|---|---|
| Replicates | For Atoh8 ChIP-seq, we conducted 2 independent experiments. Each of them includes one input file, one file of MEFs infected with an empty-AM-Tag vector and one file of MEFs infected with an Atoh8-AM-tag vector. For Bcl11b, we conducted two independent ChIP-seq experiments. We provide one input file and 2 Bcl11b-chip-seq files. |

| Sequencing depth | Sequencing depth per sample:<br>-Input: 38,668,812 sequenced reads<br>-AM-rep1: 27,773,977 sequenced reads<br>-AM-Atoh8-rep1: 34,110,739 sequenced reads<br>-AM-Atoh8 rep2: 40,109,516 sequenced reads<br>-AM-rep2: 34,110,739 sequenced reads<br>-Input_for_Atoh8_rep2: 38,001,912 sequenced reads<br><br>-Bcl11b_rep1: 38,397,339 sequenced reads<br>-Bcl11b_rep2: 32,917,302 sequenced reads<br>-Input_for_Bcl11b: 38,237,984 sequenced reads |
| --- | --- |
| Antibodies | mouse AM-Tag (active motif 9111), Bcl11b (Abcam ab18465), Bcl11b (Bethyl A300-385A) |
| Peak calling parameters | Peak caller MACS2 (version 2.2.7.1), following by default parameters, including the following options: --nomodel, --extsize 150, --control Input_dataset. |
| Data quality | Excellent. Quality assessment has been performed with the NGS-QC Generator tool (Mendoza-Parra et al. NAR 2013). |
| Software | Bowtie2 (alignement); MACS2 (peak calling); bedtools (peaks comparison); ChromHMM (peaks comparison with public datasets). |

# Flow Cytometry

## Plots

Confirm that:

☒ The axis labels state the marker and fluorochrome used (e.g. CD4-FITC).

☒ The axis scales are clearly visible. Include numbers along axes only for bottom left plot of group (a 'group' is an analysis of identical markers).

☒ All plots are contour plots with outliers or pseudocolor plots.

☒ A numerical value for number of cells or percentage (with statistics) is provided.

## Methodology

| Sample preparation | MEF cells were washed with PBS, trypsinized and resuspended in PBS with 5% FBS for FACS. |
| --- | --- |
| Instrument | Flow cytometry analysis was performed using the BD Fortessa and sorting was performed on the BD FACSAria II sorter. |
| Software | BD FACSDiva (v8.0) was used for data collection and FlowJo (v10) was used for data analysis. |
| Cell population abundance | A total of at least 10,000 events were quantified. |
| Gating strategy | Stringent gatings were always used, leaving a significant gap in between negative/postive population. Cells sorted by FACS were reanalyzed with BD Fortessa to confirm the purity. A detailed gating strategy can be found in Extended Data Fig. 3c. |

☒ Tick this box to confirm that a figure exemplifying the gating strategy is provided in the Supplementary Information.

