## [Peer Review File · Nature Cell Biology]

Peer Review Information

Journal: Nature Cell Biology

Manuscript Title: Comparative roadmaps of reprogramming and oncogenic transformation identify Bcl11b and Atoh8 as broad regulators of cellular plasticity.

Corresponding author name(s): Fabrice Laval

Reviewer Comments & Decisions:

Decision Letter, initial version:

Dear Dr Laval,

Your manuscript, "The comparative roadmaps of reprogramming and transformation unveiled that cellular plasticity is broadly controlled by a c-Myc/Atoh8/Sfrp axis", has now been seen by 3 referees, who are experts in somatic reprogramming (referee 1 and 2), and stem cells, cancer, oncogenes (referee 3). As you will see from their comments (attached below) they find this work of potential interest, but have raised substantial concerns, which in our view would need to be addressed with considerable revisions before we can consider publication in Nature Cell Biology.

Nature Cell Biology editors discuss the referee reports in detail within the editorial team, including the chief editor, to identify key referee points that should be addressed with priority, and requests that are overruled as being beyond the scope of the current study. To guide the scope of the revisions, I have listed these points below. We are committed to providing a fair and constructive peer-review process, so please feel free to contact me if you would like to discuss any of the referee comments further.

In particular, it would be essential to:

A) Clarify the discrepancy with the relevant literature as questioned by Reviewer 1:

"The role of Atoh8 in reprogramming is already reported by Divvela et al (Cells, 2019) as the authors referred. Surprisingly, the result shown in this previous report is completely opposite to the result shown in this manuscript. Divvela et al demonstrated that Atoh8/Math6 is up-regulated during reprogramming and Atoh8/Math6 KO MEFs show lower efficiency of reprogramming although they analyzed the same reprogramming event (MEF with Oct4/Sox2/Klf4/Myc). Since there are many data sets available for the transcriptomic change in the reprogramming process (Kaji, Hochedlinger etc),

the authors should confirm the universality of their finding and address the reason of the controversy to the result by Divvela et al."

B) Address the concerns related to the working mechanism for Bcl11b and Atoh 8;

Reviewer 2

"The authors uncovered that Bcl11b is highly expressed in MEFs and becomes downregulated during reprogramming and oncogenic transformation. This raises the intriguing possibility that Bcl11b controls somatic cell identity and acts as a barrier to cell fate change. It would be helpful if the authors could gather more mechanistic insights into the role of Bcl11b. Although they use it prevalently as a marker, Bcl11b is a transcription factor (mostly known as lymphoid and leukemia factor) and could have an active role in controlling MEF identity. Indeed, previous work has shown that depletion of Bcl11b causes loss of T cell identity (Li et al., Science 2010). I would suggest performing knockdown (KD) and overexpression (OE) experiments to determine whether it plays a role in cell fate changes. I expect that the KD accelerates cell fate transitions. In the presence of a phenotype, a Bcl11b ChIP experiment would also help gather some mechanistic insights. My suspect is that it could play a more substantial role than Atoh8."

"The effect of Atoh8 on reprogramming and transdifferentiation is relatively minor, and the author could include some experiments to corroborate the data presented in Figure 5. For example, the authors could check whether Atoh8 KD specifically affects the population RI4 by flow cytometry. The authors should also include experiments to prove that Atoh8 OE impairs cell fate changes. Given its role in Wnt signaling regulation, the authors should test the effect of GSK3Bi (a known enhancer of reprogramming) together with Atoh8 KD (if Atoh8 depletion activates Wnt, the authors should not see an increase with GSK3Bi treatment)."

Reviewer 3

"Bcl11b downregulation was identified as a marker of transformation and reprogramming at very early time points and the sub-population sorting experiment in Figure 2 suggests that Bcl11b downregulation is a key event in both processes. Given that a) Bcl11b has not been explored in the context of MEF cell identity or transformation/reprogramming, b) Bcl11b is a well-known regulator of T cell differentiation with similar properties (constraining cell fate) and c) the claim of uncoupling between loss of cell identity and gain of plasticity is based on Bcl11b-defined populations, functional perturbation of Bcl11b would increase the significance of this finding and these claims. The experimental system appears sufficiently established to feasibly test the impact of Bcl11b knock-down (and possibly rescue) on transformation and reprogramming."

"In this study, multiple factors were identified as correlating with (Thy1, Bcl11b) or directly impacting (FosL1, Atoh8, Srfp1/2) MEF cell reprogramming and transformation, however whether and how these factors functionally interact with each other was not discussed. Moreover, how these factors interact with the factors that actually drive each process (e.g. OSKM for reprogramming and oncogenic Kras for transformation) was not explored or discussed. Although this would not necessarily impact the main conclusions of the paper, some attempt at tying these diverse factors together within the context of these central cellular processes would improve the broader relevance to the field. Some of these points are easily explored with the data already generated. For example, Atoh8 knock-down hastened

2activation of endogenous OSKM genes, suggesting Atoh8 might repress these genes. However, Atoh8 ChIP-seq showed that it predominantly binds enhancers. Does Atoh8 bind OSKM genes? Atoh8 presumably suppresses Wnt signaling by upregulating Wnt inhibitors. Is there any evidence of Atoh8 binding to these genes?"

C) Improve the data regarding MEF identity as raised by Reviewer 3:

"The authors report the finding that acquisition of cellular plasticity precedes loss of cellular identity as a major conceptual advance. This is based on the observation that RI4/TI4 cells did not downregulate MEF identity (Figure 3f). However, in Fig 1j the MEF identity score seems to decrease pretty quickly, especially along the transformation trajectory. As the RI4/TI4 intermediates are defined by loss of both Bcl11b and Thy1 which, according to Figure 2c, would place them towards the end of the trajectories where the MEF identity has been lost, can the authors add clarity where these group 4 intermediates might lie along the trajectory? Importantly, how MEF identity is defined and what makes up the MEF identity score should be clearly stated, even if it is derived from other publications."

D) All other referee concerns pertaining to strengthening existing data, providing controls, methodological details, clarifications and textual changes as applicable should also be addressed.

E) Finally please pay close attention to our guidelines on statistical and methodological reporting (listed below) as failure to do so may delay the reconsideration of the revised manuscript. In particular please provide:

We would be happy to consider a revised manuscript that would satisfactorily address these points, unless a similar paper is published elsewhere, or is accepted for publication in Nature Cell Biology in the meantime.

- ensure that it conforms to our format instructions and publication policies (see below and www.nature.com/nature/authors/).

- provide a point-by-point rebuttal to the full referee reports verbatim, as provided at the end of this letter.

3- provide the completed Editorial Policy Checklist (found here <https://www.nature.com/authors/policies/Policy.pdf>), and Reporting Summary (found here <https://www.nature.com/authors/policies/ReportingSummary.pdf>). This is essential for reconsideration of the manuscript and these documents will be available to editors and referees in the event of peer review. For more information see <http://www.nature.com/authors/policies/availability.html> or contact me.

Nature Cell Biology is committed to improving transparency in authorship. As part of our efforts in this direction, we are now requesting that all authors identified as 'corresponding author' on published papers create and link their Open Researcher and Contributor Identifier (ORCID) with their account on the Manuscript Tracking System (MTS), prior to acceptance. ORCID helps the scientific community achieve unambiguous attribution of all scholarly contributions. You can create and link your ORCID from the home page of the MTS by clicking on 'Modify my Springer Nature account'. For more information please visit www.springernature.com/orcid.

[REDACTED]

We would like to receive a revised submission within six months. We would be happy to consider a revision even after this timeframe, however if the resubmission deadline is missed and the paper is eventually published, the submission date will be the date when the revised manuscript was received.

We hope that you will find our referees' comments, and editorial guidance helpful. Please do not hesitate to contact me if there is anything you would like to discuss.

Best wishes,
Zhe Wang

Zhe Wang, PhD
Senior Editor
Nature Cell Biology

Tel: +44 (0) 207 843 4924
email: zhe.wang@nature.com

Reviewers' Comments:

4Reviewer #1:

Remarks to the Author:

In this manuscript, the authors reported a novel genetic pathway that is shared by the initial steps of the reprogramming and transformation. They generated a mouse model in which both reprogramming and transformation events can be induced: the reprogramming of somatic cells to pluripotent stem cells by tet-inducible Oct4/Sox2/Klf4/Myc and the transformation of somatic cells to cancer-like cells by tamoxifen-inducible K-Ras G12D in combination with viral transfection of Myc. Using this system, they found that the cell populations with high plasticity can be separated by the Bcl11-low/Thy1-low expressions at the initial steps of both cases. Then they found that Atoh8 is commonly down-regulated in these populations. Functional validation revealed that Atoh8 depletion enhances both transformation and reprogramming. The analysis of Atoh8 binding site by ChIP-seq combined with other data set deduced the Myc/Atoh8/Sfrp pathway driving the Wnt signal is functionally important as a roadblock of both reprogramming and transformation.

The mechanistic overlap between reprogramming and transformation events at the molecular level is interest question to be addressed in cell biology. However, the wide-range variations of both events make it difficult to reveal the general answer. The reprogramming of somatic cells to induced pluripotent stem (iPS) cells has been well analyzed and it was shown that the process depends on the reprogramming methods and the types of somatic cells. Here the authors applied the canonical reprogramming model, reprogramming of MEF by Oct4/Sox2/Klf4/Myc, but they never addressed whether the finding is applicable for other reprogramming events with different starting cells and different reprogramming methods. For the transformation of MEF, the combination of K-Ras G12D and Myc was applied. The problem is the common use of Myc in both processes. As the result, they found the Myc/Atoh8/Sfrp pathway as a roadblock, but it may depend on the function of Myc in both reprogramming and transformation. What will be observed if the reprogramming and transformation events without Myc are analyzed?

The role of Atoh8 in reprogramming is already reported by Divvela et al (Cells, 2019) as the authors referred. Surprisingly, the result shown in this previous report is completely opposite to the result shown in this manuscript. Divvela et al demonstrated that Atoh8/Math6 is up-regulated during reprogramming and Atoh8/Math6 KO MEFs show lower efficiency of reprogramming although they analyzed the same reprogramming event (MEF with Oct4/Sox2/Klf4/Myc). Since there are many data sets available for the transcriptomic change in the reprogramming process (Kaji, Hochedlinger etc), the authors should confirm the universality of their finding and address the reason of the controversy to the result by Divvela et al.

Because of the points described above, it is difficult to find the general importance and interest in this manuscript that is required for the publication in Nature Cell Biology, I think.

Reviewer #2:

Remarks to the Author:

Huyghe and colleagues provide a comprehensive view of reprogramming and oncogenic transformation at single-cell resolution. They generate a novel mouse model, which helps them identify novel regulators of somatic cell identity and cellular plasticity. Mechanistically, the authors identified Bcl11b as a new marker of fibroblast identity and Atoh8 as a novel modulator of the Wnt pathway and cellular plasticity. Overall, the idea is original, the findings are well presented, the

5datasets and transgenic models they generated are helpful for the scientific community, and the paper is of general interest.

I have some suggestions and minor points to help improve the manuscript:

1. The authors uncovered that Bcl11b is highly expressed in MEFs and becomes downregulated during reprogramming and oncogenic transformation. This raises the intriguing possibility that Bcl11b controls somatic cell identity and acts as a barrier to cell fate change. It would be helpful if the authors could gather more mechanistic insights into the role of Bcl11b. Although they use it prevalently as a marker, Bcl11b is a transcription factor (mostly known as lymphoid and leukemia factor) and could have an active role in controlling MEF identity. Indeed, previous work has shown that depletion of Bcl11b causes loss of T cell identity (Li et al., Science 2010). I would suggest performing knockdown (KD) and overexpression (OE) experiments to determine whether it plays a role in cell fate changes. I expect that the KD accelerates cell fate transitions. In the presence of a phenotype, a Bcl11b ChIP experiment would also help gather some mechanistic insights. My suspect is that it could play a more substantial role than Atoh8.

2. Figure 2l, m is a bit puzzling. The reprogramming and transformation plots at day 5 resemble the plots at days 10-17 in Figure 2e. So, the reprogramming and transformation in Figure 2l,m seem massively accelerated. Moreover, based on the authors' finding on Bcl11b⁻ cells, I would expect that the population RI2 (Bcl11b⁻/Thy-1⁺) reprograms better than the population RI3 (Bcl11b⁺/Thy-1⁻). The author should explain these discrepancies.

3. In Figure 2, the authors mainly focused on downregulated markers during the reprogramming and transformation processes. It would be great to discuss whether transient upregulated markers mark cell populations poised to change fate. This could indicate whether intermediates can be sorted during the process for further characterization in future studies.

4. The effect of Atoh8 on reprogramming and transdifferentiation is relatively minor, and the author could include some experiments to corroborate the data presented in Figure 5. For example, the authors could check whether Atoh8 KD specifically affects the population RI4 by flow cytometry. The authors should also include experiments to prove that Atoh8 OE impairs cell fate changes. Given its role in Wnt signaling regulation, the authors should test the effect of GSK3Bi (a known enhancer of reprogramming) together with Atoh8 KD (if Atoh8 depletion activates Wnt, the authors should not see an increase with GSK3Bi treatment).

5. In Figure S1i, S2d, S2m, 5m, 5q, 6j, the authors present statistical differences and mention "n=3" in the figure legends; however, only two samples per condition are shown. The authors should include the missing data points.

Minor points:

1. The authors should show the actual numbers throughout the Figures and not fold-change/percentage/fold ratio (for example, figure 2F, 4C).

2. In Figure S1g, no error bars are presented for two conditions. The authors should include the

6missing data.

3. The cell cycle is only affected in “transfo” conditions but not in the “repro+transfo.” The author should clarify this discrepancy and show the original FACS plots.

4. Figure 3f does not help and can be removed, as the populations RI4 and TI4 still have MEF identity. Moreover, the axis is starting at 3.5 while it should begin at 0.

Reviewer #3:

Remarks to the Author:

Review of Huyghe et al NCB 2021

This study interrogated the early stages of cellular reprogramming and cellular transformation, to identify the molecular changes that define and drive these processes, and to examine whether loss of cellular identity is coupled to gain of plasticity. The authors developed and utilized a mouse model that enabled inducible reprogramming or transformation of mouse embryonic fibroblasts (MEFs) *ex vivo*. Reprogramming of MEFs to induced pluripotent stem cells (iPSCs) was achieved by doxycycline-inducible expression of Oct4, Sox2, Klf4 and Myc (OSKM) whereas oncogenic transformation was achieved by tamoxifen-induced activation of oncogenic K-ras (sometimes with cMyc overexpression and trp53 knockdown). Established functional readouts were used to assess each model (teratoma formation and alkaline phosphatase positive colonies by reprogrammed iPSCs, and soft agar colony formation/anchorage independent growth and *in vivo* tumor formation for transformed MEFs). Multiple molecular modalities were used to characterize early time points of each process (single cell and bulk RNA-seq, ATAC-seq). The main findings include identification of Bcl11b as a novel marker of MEF cell identity and whose downregulation (together with the previously described marker Thy1) can be used to prospectively isolate cells prone to (or already undergoing) reprogramming/transformation; use of ATAC-seq data to identify FosL1 as having divergent roles in transformation (loss blocks) and reprogramming (loss enhances); purported uncoupling of loss of cell identity and gain of plasticity; identification of shared gene expression changes that occur during both reprogramming and transformation; and identification of the Atoh8 transcription factor as a nonspecific constraint to reprogramming and transformation through its presumed role in suppressing Wnt signaling. The evidence that Bcl11b is a faithful marker of transformation and reprogramming is well supported by the cell sorting experiments presented in Figure 2, although experimental perturbation would strengthen the functional relevance of this marker even further. Identification of Atoh8 as a relatively nonspecific constrainer of cellular plasticity is also well supported by the data, including examining Atoh8 knock-down in additional models of transformation and trans-differentiation. Overall, this study illustrates the power of examining dynamic cellular processes in a highly controlled, defined manner using time-resolved high-resolution molecular read-outs. The main conclusions of the study could be significantly strengthened by additional experiments, clarifying aspects of the experimental design, and analyzing the wealth of data in additional ways, as outlined below.

Major comments

1. The authors report the finding that acquisition of cellular plasticity precedes loss of cellular identity as a major conceptual advance. This is based on the observation that RI4/TI4 cells did not

7downregulate MEF identity (Figure 3f). However, in Fig 1j the MEF identity score seems to decrease pretty quickly, especially along the transformation trajectory. As the RI4/TI4 intermediates are defined by loss of both Bcl11b and Thy1 which, according to Figure 2c, would place them towards the end of the trajectories where the MEF identity has been lost, can the authors add clarity where these group 4 intermediates might lie along the trajectory? Importantly, how MEF identity is defined and what makes up the MEF identity score should be clearly stated, even if it is derived from other publications.

2. Bcl11b downregulation was identified as a marker of transformation and reprogramming at very early time points and the sub-population sorting experiment in Figure 2 suggests that Bcl11b downregulation is a key event in both processes. Given that a) Bcl11b has not been explored in the context of MEF cell identity or transformation/reprogramming, b) Bcl11b is a well-known regulator of T cell differentiation with similar properties (constraining cell fate) and c) the claim of uncoupling between loss of cell identity and gain of plasticity is based on Bcl11b-defined populations, functional perturbation of Bcl11b would increase the significance of this finding and these claims. The experimental system appears sufficiently established to feasibly test the impact of Bcl11b knock-down (and possibly rescue) on transformation and reprogramming.

3. In this study, multiple factors were identified as correlating with (Thy1, Bcl11b) or directly impacting (FosL1, Atoh8, Srfp1/2) MEF cell reprogramming and transformation, however whether and how these factors functionally interact with each other was not discussed. Moreover, how these factors interact with the factors that actually drive each process (e.g. OSKM for reprogramming and oncogenic Kras for transformation) was not explored or discussed. Although this would not necessarily impact the main conclusions of the paper, some attempt at tying these diverse factors together within the context of these central cellular processes would improve the broader relevance to the field. Some of these points are easily explored with the data already generated. For example, Atoh8 knock-down hastened activation of endogenous OSKM genes, suggesting Atoh8 might repress these genes. However, Atoh8 ChIP-seq showed that it predominantly binds enhancers. Does Atoh8 bind OSKM genes? Atoh8 presumably suppresses Wnt signaling by upregulating Wnt inhibitors. Is there any evidence of Atoh8 binding to these genes?

Minor comments and suggestions

1. Experimental details are missing in the following places

- How was the MEF identity score identified and applied to the data generated in this manuscript?
- page 10 mentions a result showing expression change of Dmrtc2 and Pou3f1 but these genes are not shown in Extended Data Fig. 4i (and the gene Shisa8 is shown in ED 4i but not listed in the results section).
- Page 11 mentions a "protein class analysis with pantherdb" was used to find master regulators of a transcriptional program, but unclear how this analysis was performed, and it is not mentioned in the Methods section (what genes were used as input? How were they compiled? How did this analysis lead to the subsequent focus on the genes highlighted in Cluster 2 of fig. 3k?)
- Figure 6i: how long after Atoh8 knockdown were cells taken for RNA-seq? This is important for interpreting that the MEF identity score did not change. In many figures, the time point of the experiment was difficult to track down, so adding the timepoint to the figure itself or including in the figure legend each time would be helpful.

2. Based on the wide-spread use of the term 'cellular plasticity' it would be good to clearly define this term within the context of the read-outs presented in this study.

3. Adding Bcl11B/Thy1 expression (high/low) to figure 2n would aid interpretation.
4. A suggestion is to analyze the joint Bcl11B/Thy1 expression in the single cell data from Figure 1 as support for the transition kinetics inferred in Figure 2 (how does the frequency of Bcl11b/Thy1 high/low cells change along each trajectory? What about the pattern in the bifurcation area, which is relatively under-discussed in this manuscript?)
5. An additional suggestion would be to identify genes that significantly correlate with Bcl11b expression during reprogramming and transformation to identify other genes putatively regulated by or regulating Bcl11b expression during these processes. Are they similar in the repro/transfo context? Particularly given the GO terms of the common set in Figure 3h, it is likely worthwhile to describe a putative Bcl11b controlled gene expression program in each context.
6. Unsupervised hierarchical clustering of the ATAC-seq data to identify clusters in an unsupervised manner is another way to identify potentially interesting clusters and see how it compares to the supervised classification of C1-C6 in Fig. 3c,d
7. Extended data fig. 4i should also show Dmrtc2 and Pou3f1 but these are missing. Shisa8 is shown but isn't listed in the main text (p.10)
8. Fig. 3a: add the % variance explained by each component
9. Why is p53 knock-down used in the experiments shown in Figure 4, but not in previous experiments of the same experimental system?
10. The significance of the gene expression results presented in Figure 4f-j is unclear; gene expression differences can be seen, but how they support the conclusion that "Atoh8 constrains cellular plasticity" is not clear. The functional read outs on the other hand are very convincing.
11. Also figure 4: the colony assay and western blots show that Atoh8 KD has an impact at early time points (d.3 and d.6) but analysis of gene expression was performed at day >30, so it's unclear if the time points directly impacted by Atoh8 are missed (e.g. the results in Figs4f,h,I,j are simply reflecting emergence of different cell states/types that are consequential to Atoh8 knock-down).
12. Figure 4j. References for Ube2c and Top2a as "promoting cancer cell invasion and migration"?
13. In the discussion, it is stated that the authors identified "the existence of a molecular program that commonly emerges during both processes" but this program remains obscure after reading the manuscript. Is this program all of the genes mentioned (Thy1, Bcl11b, Atoh8, Wnt regulators, etc.?). Is this program the "bifurcation area" shown in Figure 1k?
14. Page 6: "let us" should be "led us""
15. Page 7: "reflect the ability" should be "reflects the ability"
16. Fig4 legend: "Cells were splited" should be "Cells were split" or "Cells were passaged at least 10 times"

REFERENCES – are limited to a total of 70 for Articles, Resources, Technical Reports; and 40 for Letters. This includes references in the main text and Methods combined. References must be

numbered sequentially as they appear in the main text, tables and figure legends and Methods and must follow the precise style of Nature Cell Biology references. References only cited in the Methods should be numbered consecutively following the last reference cited in the main text. References only associated with Supplementary Information (e.g. in supplementary legends) do not count toward the total reference limit and do not need to be cited in numerical continuity with references in the main text. Only published papers can be cited, and each publication cited should be included in the numbered reference list, which should include the manuscript titles. Footnotes are not permitted.

Methods should be written concisely, but should contain all elements necessary to allow interpretation and replication of the results. As a guideline, Methods sections typically do not exceed 3,000 words. The Methods should be divided into subsections listing reagents and techniques. When citing previous methods, accurate references should be provided and any alterations should be noted. Information must be provided about: antibody dilutions, company names, catalogue numbers and clone numbers for monoclonal antibodies; sequences of RNAi and cDNA probes/primers or company names and catalogue numbers if reagents are commercial; cell line names, sources and information on cell line identity and authentication. Animal studies and experiments involving human subjects must be reported in detail, identifying the committees approving the protocols. For studies involving human subjects/samples, a statement must be included confirming that informed consent was obtained. Statistical analyses and information on the reproducibility of experimental results should be provided in a section titled "Statistics and Reproducibility".

All Nature Cell Biology manuscripts submitted on or after March 21 2016 must include a Data availability statement at the end of the Methods section. For Springer Nature policies on data availability see <http://www.nature.com/authors/policies/availability.html>; for more information on this particular policy see <http://www.nature.com/authors/policies/data/data-availability-statements-data-citations.pdf>. The Data availability statement should include:

- Accession codes for primary datasets (generated during the study under consideration and designated as "primary accessions") and secondary datasets (published datasets reanalysed during the study under consideration, designated as "referenced accessions"). For primary accessions data should be made public to coincide with publication of the manuscript. A list of data types for which submission to community-endorsed public repositories is mandated (including sequence, structure, microarray, deep sequencing data) can be found here <http://www.nature.com/authors/policies/availability.html#data>.
- Unique identifiers (accession codes, DOIs or other unique persistent identifier) and hyperlinks for datasets deposited in an approved repository, but for which data deposition is not mandated (see here for details <http://www.nature.com/sdata/data-policies/repositories>).
- At a minimum, please include a statement confirming that all relevant data are available from the authors, and/or are included with the manuscript (e.g. as source data or supplementary information),

11listing which data are included (e.g. by figure panels and data types) and mentioning any restrictions on availability.

- If a dataset has a Digital Object Identifier (DOI) as its unique identifier, we strongly encourage including this in the Reference list and citing the dataset in the Methods.

We recommend that you upload the step-by-step protocols used in this manuscript to the Protocol Exchange. More details can found at www.nature.com/protocolexchange/about.

All imaging data should be accompanied by scale bars, which should be defined in the legend. Cropped images of gels/blots are acceptable, but need to be accompanied by size markers, and to retain visible background signal within the linear range (i.e. should not be saturated). The boundaries of panels with low background have to be demarked with black lines. Splicing of panels should only be considered if unavoidable, and must be clearly marked on the figure, and noted in the legend with a statement on whether the samples were obtained and processed simultaneously. Quantitative comparisons between samples on different gels/blots are discouraged; if this is unavoidable, it should only be performed for samples derived from the same experiment with gels/blots were processed in parallel, which needs to be stated in the legend.

12- We accept PowerPoint (.PPT) files if they are fully editable. However, please refrain from adding PowerPoint graphical effects to objects, as this results in them outputting poor quality raster art. Text used for PowerPoint figures should be Helvetica (preferred) or Arial.

Supplementary items should relate to a main text figure, wherever possible, and should be mentioned

sequentially in the main manuscript, designated as Supplementary Figure, Table, Video, or Note, and numbered continuously (e.g. Supplementary Figure 1, Supplementary Figure 2, Supplementary Table 1, Supplementary Table 2 etc.).

The total number of Supplementary Figures (not including the “unprocessed scans” Supplementary Figure) should not exceed the number of main display items (figures and/or tables (see our Guide to Authors and March 2012 editorial <http://www.nature.com/ncb/authors/submit/index.html#suppinfo>; <http://www.nature.com/ncb/journal/v14/n3/index.html#ed>). No restrictions apply to Supplementary Tables or Videos, but we advise authors to be selective in including supplemental data.

GUIDELINES FOR EXPERIMENTAL AND STATISTICAL REPORTING

REPORTING REQUIREMENTS – To improve the quality of methods and statistics reporting in our papers we have recently revised the reporting checklist we introduced in 2013. We are now asking all life sciences authors to complete two items: an Editorial Policy Checklist (found here <https://www.nature.com/authors/policies/Policy.pdf>) that verifies compliance with all required editorial policies and a reporting summary (found here <https://www.nature.com/authors/policies/ReportingSummary.pdf>) that collects information on experimental design and reagents. These documents are available to referees to aid the evaluation of the manuscript. Please note that these forms are dynamic ‘smart pdfs’ and must therefore be downloaded and completed in Adobe Reader. We will then flatten them for ease of use by the reviewers. If you would like to reference the guidance text as you complete the template, please access these flattened versions at <http://www.nature.com/authors/policies/availability.html>.

STATISTICS – Wherever statistics have been derived the legend needs to provide the n number (i.e. the sample size used to derive statistics) as a precise value (not a range), and define what this value represents. Error bars need to be defined in the legends (e.g. SD, SEM) together with a measure of centre (e.g. mean, median). Box plots need to be defined in terms of minima, maxima, centre, and percentiles. Ranges are more appropriate than standard errors for small data sets. Wherever statistical significance has been derived, precise p values need to be provided and the statistical test

14used needs to be stated in the legend. Statistics such as error bars must not be derived from $n < 3$. For sample sizes of $n < 5$ please plot the individual data points rather than providing bar graphs. Deriving statistics from technical replicate samples, rather than biological replicates is strongly discouraged. Wherever statistical significance has been derived, precise p values need to be provided and the statistical test stated in the legend.

Author Rebuttal to Initial comments

Reviewers' Comments:

Reviewer #1:

Remarks to the Author:

In this manuscript, the authors reported a novel genetic pathway that is shared by the initial steps of the reprogramming and transformation. They generated a mouse model in which both reprogramming and transformation events can be induced: the reprogramming of somatic cells to pluripotent stem cells by tet-inducible Oct4/Sox2/Klf4/Myc and the transformation of somatic cells to cancer-like cells by tamoxifen-inducible K-Ras G12D in combination with viral transfection of Myc. Using this system, they found that the cell populations with high plasticity can be separated by the Bcl11-low/Thy1-low expressions at the initial steps of both cases. Then they found that Atoh8 is commonly down-regulated in these populations. Functional validation revealed that Atoh8 depletion enhances both transformation and reprogramming. The analysis of Atoh8 binding site by ChIP-seq combined with other data set deduced the Myc/Atoh8/Sfrp pathway driving the Wnt signal is functionally important as a roadblock of both reprogramming and transformation.

15The mechanistic overlap between reprogramming and transformation events at the molecular level is an interesting question to be addressed in cell biology. However, the wide-range variations of both events make it difficult to reveal the general answer.

We thank Reviewer 1 (R1) for his/her interest for exploring “the mechanistic overlap between reprogramming and transformation...”. We improved significantly the manuscript by addressing his/her constructive comments:

The reprogramming of somatic cells to induced pluripotent stem (iPS) cells has been well analyzed and it was shown that the process depends on the reprogramming methods and the types of somatic cells. Here the authors applied the canonical reprogramming model, reprogramming of MEF by Oct4/Sox2/Klf4/Myc, but they never addressed whether the finding is applicable for other reprogramming events with different starting cells and different reprogramming methods. For the transformation of MEF, the combination of K-Ras G12D and Myc was applied. The problem is the common use of Myc in both processes. As the result, they found the Myc/Atoh8/Sfrp pathway as a roadblock, but it may depend on the function of Myc in both reprogramming and transformation. What will be observed if the reprogramming and transformation events without Myc are analyzed?

We thank R1 for his/her constructive comment. We conducted different approaches to assess the relevance of our findings in alternative somatic cell types and with other reprogramming/transformation/transdifferentiation methods.

1-We assessed first whether the emergence of Bcl11b^{low}/Thy1^{low} (BLTL) cells constitutes a broad hallmark of reprogramming and transformation rather than an event specifically linked to the use of c-Myc and MEFs.

-We assessed whether BLTL cells emerged when alternative reprogramming and transformation methods excluding c-Myc were used. For reprogramming, we compared OSKM¹ with (i) OSK + Wnt inhibitor IWP2 and (ii) Sall4, Nanog, Esrrb and Lin28 (SNEL)². For transformation, we compared K-Ras^{G12D}/c-Myc with (i) K-Ras^{G12D}+c-Myc+p53KD, (ii) H-Ras^{G12V}+CyclinE+p53KD and (iii) H-Ras^{G12V}+p53KD. Strikingly, we found that BLTL cells emerged in all of these conditions at different efficiencies. Please see Figure 3e and Extended Data Fig. 3e. In addition, we showed by FACS isolation followed by replating that BLTL cells generated with OSK+IWP2 were more efficient at forming pluripotent derivatives. Data are presented in Figure 3g and Extended Data Fig. 3e.

-We next derived mouse adult ear fibroblasts (MAEFs) to conduct reprogramming and transformation experiments because we found that they express both Bcl11b and Thy1. Of note, even

if MAEFs and MEFs are mesenchymal cells, their molecular and functional features are heavily different³. We found that BLTL cells emerged when Bcl11b-tdTomato knock-in MAEFs were induced for reprogramming (OSKM) and transformation (K-Ras^{G12D}/c-Myc), as observed with MEFs. Moreover, FACS isolation demonstrated that these BLTL cells were prone to generate iPS colonies. These data are incorporated in Figure 3f, h and Extended Data Fig. 3f.

2-We attempted to evaluate Atoh8 function during reprogramming in other cell types but it was not expressed in MAEFs or in T lymphocytes (Extended Data Fig. 6b-c). However, during the review, we identified the Bcl11b TF as a regulator of cell fate in MEFs. In line with R1 request, we also evaluated Bcl11b function in reprogramming of T lymphocytes as an alternative cell type, as well as during MEFs to neuron transdifferentiation. Data are incorporated in an entirely new Figure 2, panels k-n.

The role of Atoh8 in reprogramming is already reported by Divvela et al (Cells, 2019) as the authors referred. Surprisingly, the result shown in this previous report is completely opposite to the result shown in this manuscript. Divvela et al demonstrated that Atoh8/Math6 is up-regulated during reprogramming and Atoh8/Math6 KO MEFs show lower efficiency of reprogramming although they analyzed the same reprogramming event (MEF with Oct4/Sox2/Klf4/Myc). Since there are many data sets available for the transcriptomic change in the reprogramming process (Kaji, Hochedlinger etc), the authors should confirm the universality of their finding and address the reason of the controversy to the result by Divvela et al.

We thank R1 for this very important comment. We summarized below the four main controversies with the potential explanations for the discrepancies:

Controversy 1: Atoh8 is not expressed in mouse adult ear fibroblasts (MAEF) (Divvela) – Atoh8 is expressed in mouse embryonic fibroblasts (MEF) (our study)

Of high importance, Divvela et al. used **mouse adult ear fibroblasts (MAEFs)** for reprogramming (please see page 4 of the publication “To evaluate the expression of Math6 during somatic cell reprogramming, we prepared fibroblasts from the ears of adult Math6Flag-tag mice and subjected them to reprogramming using Oct4, Sox2, Klf4 and c-Myc...”). In contrast, we used mouse embryonic fibroblasts (MEFs). Therefore, the two studies work on completely different reprogramming events, and it might explain the difference of Atoh8 expression. To assess it, we compared Atoh8 expression in MAEFs, MEFs and iPS cells by Q-RT-PCR and WB. We showed that Atoh8 is indeed expressed at very low levels in MAEFs, when compared with MEFs. These data are included in Extended Data Fig. 6b-c. **The first discrepancy might therefore be explained by the use of different cells-of-origin in the two studies.**

17As an additional note, Divvela et al. used a **Math6-Flag knock-in** MAEFs and assessed the expression of a fused Atoh8-flag protein. In contrast, we analyzed the expression of the **endogenous native Atoh8** using a commercial antibody (that we validated in gain-and loss-of-function approaches). Therefore, we cannot exclude the possibility that the presence of the flag fused to Atoh8 affects its expression, localization and function.

Controversy 2: Atoh8 promotes MAEFs reprogramming (Divvela) – Atoh8 constrains MEFs reprogramming (our study)

As stated before, the two studies used different cells-of-origin (MAEFs vs MEFs) to evaluate Atoh8 function. Therefore, we cannot rule out the possibility that Atoh8 functions in a cellular context-dependent manner, as found for the NuRD complex during reprogramming⁴. **The discrepancy here might be explained by the different cells-of-origin in the two studies.**

To reinforce our findings, we asked Jose Polo's lab to reproduce independently in his own lab the experiments of Atoh8 depletion in MEFs prior to reprogramming. As shown below in Reviewer Figure 1, using MEFs coming from 3 different embryos, an independent lab confirmed that Atoh8 constrains MEF reprogramming efficiency.

Reviewer Figure 1: Atoh8 constrains MEF reprogramming. (a) Pictures showing reprogramming experiments conducted by treating MEFs derived from 3 different embryos (Embryo 1 or E1) treated or not with Dox to induce OSKM expression. MEFs were also non infected (no virus) or infected with lentiviral particles expressing scramble shRNA, GFP or Atoh8 shRNA. (b) Countings of iPS colonies from (a). Data are the mean \pm s.d. (n=3 independent experiments).

Controversy 3: Atoh8 is induced during MAEFs reprogramming (Divvela) – Atoh8 is downregulated during MEFs reprogramming (our study)

A very important difference is that Divvela et al. analyzed Atoh8 expression **at the whole population level** while we focused on **FACS-sorted subset of reprogramming cells**. Just as a reminder, reprogramming events are quite rare in the culture and can be obscured by the non-reprogramming cells when analyzing whole population of cells. To assess whether the discrepancy comes from that difference, we interrogated 6 independent public RNA-seq datasets for *Atoh8* expression, including data from total populations (Alex Meissner lab⁵, Andreas Nagy lab⁶ and Kathrin Plath lab⁷), FACS-sorted subpopulations

of cells (from Konrad Hochedlinger Lab⁸ and Jose Polo Lab⁹) and individual cells (from Erik Lander lab¹⁰) (see Extended Data Fig. 6a) during MEFs reprogramming. Our conclusions are:

-At the whole population level, *Atoh8* transcript level is variable across studies. However, the main trend is a downregulation during MEFs reprogramming.

-In reprogramming (FACS-sorted) cells, *Atoh8* is robustly downregulated.

Therefore, the discrepancy in the dynamic expression of *Atoh8* is explained by (i) the different cell-of-origin and by (ii) the fact that we analyzed rare FACS-sorted reprogramming cells when Divvela et al. analyzed the whole population, mainly composed of non-reprogramming cells.

Controversy 4: *Atoh8* is highly expressed in mouse iPS cells – *Atoh8* is not expressed in mouse iPS cells (our study)

In the 6 independent datasets, *Atoh8* expression is **completely undetectable in iPS cells, at the bulk but also single-cell levels** (Extended Data Fig. 6a). In addition, we conducted WB and Q-RTPCR analyses on MEFs, MAEFs and iPS cells that confirmed that ***Atoh8* is not expressed in naïve pluripotent cells** (Extended Data Fig. 6b-c). We hypothesize that the expression detected by Divvela et al. might come from (i) the feeder cells (MEFs) on which iPS cells are grown and/or (ii) by the genetic modification of the *Atoh8* locus that disturbs its expression.

In conclusion, the use of **different somatic cells-of -origin** (MEFs versus MAEFs), combined with the fact that Divvela et al. worked with a **flagged** protein on **whole population of cells**, might explain the discrepancies. After exchanging with Stylianos Lefkopoulos from Nature Cell Biology, we decided to include the published and newly generated data in the Extended Data Figure 6a-c. We also discuss the controversy in the main text page 14 and in the discussion page 18.

As a personal note, by assessing carefully the data from Figure 1 of Divvela et al. (see below), we are concerned by the following points:

-On the WB figure 1B, the different lanes are clearly not from the same membrane and the loading control (alpha-tubulin) is not constant between the samples.

-On the IF Figure 1C, *Atoh8*-Flag staining is exclusively cytoplasmic, which is not expected for a bHLH transcription factor, supporting the hypothesis that the modification of the protein might impact its function.

Figure 1. Math6 expression during somatic cell reprogramming. (A) mRNA expression of *Math6* on Day 0 (Untransduced fibroblasts), 5, 11 and 17 of reprogramming generated by RT-PCR. The data shown were normalized to housekeeping gene *GAPDH*. The data shown is the average of three similar individual sets of experiments. From here on this applies to all gene expression data. (B) Expression of Flag-tagged Math6 on Day 0, 5, 11 and 17 of reprogramming analysed by Western blot. α -tubulin was used as a control. (C) Immunofluorescence images showing Flag-tagged Math6 expression in fibroblasts, Day 0, 5, 11, 17 of reprogramming.

Because of the points described above, it is difficult to find the general importance and interest in this manuscript that is required for the publication in Nature Cell Biology, I think.

We thank R1 because her/his comments allowed us to demonstrate that our findings are not strictly limited to c-Myc and MEFs reprogramming but rather constitute a broad hallmark of reprogramming/transformation/transdifferentiation in various somatic cell types (MEFs, MAEFs, T-lymphocytes, Human dermal fibroblasts). We believe that the data generated to address them improved significantly the impact and interest of the work. We hope that she/he will agree.

Reviewer #2:

Remarks to the Author:

Huyghe and colleagues provide a comprehensive view of reprogramming and oncogenic transformation at single-cell resolution. They generate a novel mouse model, which helps them identify novel regulators of

22somatic cell identity and cellular plasticity. Mechanistically, the authors identified Bcl11b as a new marker of fibroblast identity and Atoh8 as a novel modulator of the Wnt pathway and cellular plasticity. Overall, the idea is original, the findings are well presented, the datasets and transgenic models they generated are helpful for the scientific community, and the paper is of general interest. I have some suggestions and minor points to help improve the manuscript:

We first would like to thank R2 for her/his strong support of our work, especially for noticing the originality and general interest of our paper.

1. The authors uncovered that Bcl11b is highly expressed in MEFs and becomes downregulated during reprogramming and oncogenic transformation. This raises the intriguing possibility that Bcl11b controls somatic cell identity and acts as a barrier to cell fate change. It would be helpful if the authors could gather more mechanistic insights into the role of Bcl11b. Although they use it prevalently as a marker, Bcl11b is a transcription factor (mostly known as lymphoid and leukemia factor) and could have an active role in controlling MEF identity. Indeed, previous work has shown that depletion of Bcl11b causes loss of T cell identity (Li et al., Science 2010). I would suggest performing knockdown (KD) and overexpression (OE) experiments to determine whether it plays a role in cell fate changes. I expect that the KD accelerates cell fate transitions. In the presence of a phenotype, a Bcl11b ChIP experiment would also help gather some mechanistic insights. My suspect is that it could play a more substantial role than Atoh8.

We sincerely thank R2 for his/her pertinent comment. As suggested, we conducted a large number of approaches to dissect Bcl11b function during reprogramming, transformation and transdifferentiation that are now presented in an entirely new Figure 2:

1-We started by modulating Bcl11b expression prior to induce reprogramming or transformation. Bcl11b gain- and loss-of-function approaches in MEFs (using RNA interference but also Bcl11b conditional KO MEFs¹¹) led to demonstrate that Bcl11b depletion significantly increased reprogramming and transformation efficiencies, as suggested by R2 (please see Fig. 2e-i). Even if not requested, we next conducted teratoma formation assays to show that Bcl11b depletion has no major detrimental effect on the acquisition of *in vivo* multilineage differentiation potential (Fig. 2j). Therefore, we demonstrated that Bcl11b functionally acts as a gatekeeper of the somatic state.

2-We also assessed, as requested, whether Bcl11b accelerates cell state transitions. We showed unexpectedly that Bcl11b depletion does not accelerate the emergence of cells that activated the endogenous Oct4 reporter, in contrast to Atoh8. Data are presented below in Reviewer Figure 2 as we decided not to include them in the revised draft.

Reviewer Figure 2: Bcl11b depletion does not accelerate the activation of the pluripotent network. (a) The FACS plots presents the dynamic emergence of Pou5f1-GFP + cells during reprogramming in ctrl and Bcl11b KD settings.

3-As indicated by R2, Bcl11b is highly expressed in T lymphocytes. Therefore, even if not requested, and in order to broaden the impact of our finding, we evaluated Bcl11b role in the reprogramming of T lymphocytes. Using OSKM Dox-inducible primary T cells, we demonstrated that Bcl11b constrains reprogramming of this alternative somatic cell type. Data are presented in Figure 2k-l. We also showed that Bcl11b constrains MEFs to neuron transdifferentiation triggered by Brn2, Ascl1 and Myt1l (Fig. 2m-n).

4-Next, in order to understand Bcl11b function, we combined RNA-seq and ChIP-seq analyses. We compared the transcriptomes of control, Bcl11b-overexpressing (OE) and Bcl11b-knockdown (KD) MEFs. In parallel, we optimized the Bcl11b ChIP-seq conditions by combining two commercial antibodies (Abcam, ab18465, Bethyl, A300-385A). We identified 7430 specific peaks of Bcl11b binding to the MEF genome (Fig. 2p). We found that Bcl11b modulation impacts 979 genes (adjusted p-value < 5.10⁻²; log₂ FC > 0.5 or < -0.5) with 122 genes (13%) presenting a Bcl11b binding site within 10kb of the TSS (Fig. 2s). Finally, the fact that several Bcl11b targets were related to the Mapk signalling pathway led us to demonstrate that Bcl11b directly regulates pErk1/2 levels in MEFs (Fig. 2t), providing a potential explanation of its mode of action during reprogramming¹².

Overall, we invested important efforts to address the pertinent suggestion of R2. We believe that the results presented in Figure 2 improves significantly the impact of the manuscript by identifying Bcl11b as

a new broad-range gatekeeper of somatic states from reprogramming, transdifferentiation and transformation.

2. Figure 2l, m is a bit puzzling. The reprogramming and transformation plots at day 5 resemble the plots at days 10-17 in Figure 2e. So, the reprogramming and transformation in Figure 2l,m seem massively accelerated.

We thank R2 for his/her pertinent comment and we apologize for the lack of clarity that led to that misunderstanding. For Figure 2e (now Fig. 3b), we FACS-sorted $Bcl11b^{high}/Thy1^{high}$ (BHTH) cells prior to induce reprogramming/transformation (as indicated in the initial draft). Therefore, by starting with this subset of highly refractory cells, we significantly slowed down the reprogramming and transformation processes. Similar enrichment procedures are commonly used in the reprogramming field, for example for Thy1 by K. Hochedlinger lab¹³. In contrast, for the panels Fig. 2l and 2m (now Fig. 3l-m), we started the experiments with unsorted MEFs. This difference of experimental design explains that the process appeared somewhat accelerated in those panels. We sincerely apologize for this lack of accuracy, as we forgot to provide this crucial information. In the revised draft, we removed the initial misleading panels and indicated the difference of experimental design in the figure legends of Figure 3b and 3l-m on page 26.

Moreover, based on the authors finding on $Bcl11b^{-}$ cells, I would expect that the population RI2 ($Bcl11b^{-}/Thy1^{+}$) reprograms better than the population RI3 ($Bcl11b^{+}/Thy1^{-}$). The author should explain these discrepancies.

We thank R2 for noticing this point that becomes even more important now that we showed that $Bcl11b$ functionally restrains reprogramming and transformation (new Figure 2). As noticed by R2, when we FACS sorted cells based on the combined expression of $Bcl11b$ and $Thy1$, the downregulation of $Bcl11b$ in RI2 (BLTH) cells does not correlate with a significant improvement of reprogramming/transforming efficiency. To understand this apparent discrepancy, we hypothesized that, even if BLTH cells downregulated $Bcl11b$ expression, they still harbor molecular differences with BLTL cells. To answer this question, we FACS sorted the 4 subpopulations at day5 of reprogramming and transformation and conducted Q-RTPCR analyses. We noticed that BLTH cells still harbor high levels of *Atoh8* transcript (equivalent to refractory BHTH cells) when compared with BLTL cells during both reprogramming and transformation, which might explain their functional properties. The data are presented in the Reviewer Figure 3 below because we could not find a pertinent way to integrate them within the manuscript. Indeed, *Atoh8* identification is presented in Figure 3 after the capture of the cellular intermediates in Figure 2.

Reviewer Figure 3: Q-RTPCR depicts *Atoh8* expression level in cellular intermediates during reprogramming and transformation. BHTH, BLTH, BHTL and BLTL cells were FACS sorted at day5 of reprogramming and transformation and subjected to Q-RTPCR analyses. Data are the mean \pm s.d. (n=2-3 independent experiments).

3. In Figure 2, the authors mainly focused on downregulated markers during the reprogramming and transformation processes. It would be great to discuss whether transient upregulated markers mark cell populations poised to change fate. This could indicate whether intermediates can be sorted during the process for further characterization in future studies.

We thank R2 for his/her pertinent comment. We now discussed the transient induction of some cell surface markers, such as CD14, CD53, CD72 and CD84, that might be used to refine the tracking of reprogramming/transforming intermediates (see discussion page 18).

4. The effect of *Atoh8* on reprogramming and transdifferentiation is relatively minor, and the author could include some experiments to corroborate the data presented in Figure 5. For example, the authors could check whether *Atoh8* KD specifically affects the population RI4 by flow cytometry.

We thank R2 for this interesting comment. We analyzed by FACS the dynamic emergence of RI4 (now called BLTL) cells during transformation and reprogramming in “Ctrl” and “*Atoh8*-KD” settings. We found that *Atoh8* depletion significantly accelerates the appearance of RI4 cells during transformation but not reprogramming. For reprogramming, we already showed that *Atoh8* depletion accelerates the arrival of Pou5f1-GFP+ cells. To reconcile this apparent discrepancy, we hypothesize that, even if the number of BLTL cells is not increased by *Atoh8* depletion, the *Atoh8*-depleted BLTL cells might be more “advanced” in their reprogramming stage than control BLTL cells. However, addressing this hypothesis in detail will require additional approaches such as transcriptomic characterization that are, we believe, out of the scope of the study. The data are now presented in Figure 5c-d.

The authors should also include experiments to prove that *Atoh8* OE impairs cell fate changes.

26

We performed the required experiments but found that Atoh8 OE does not significantly affect reprogramming or transformation efficiencies (see Reviewer Fig. 4 below). For reprogramming, we hypothesize that the massive epigenetic reorganization induced by OSKM in the early days of the process, especially on the Atoh8-bound regions (as demonstrated in Figure 7), is not sufficiently counteracted by the sole exogenous Atoh8 expression. Of note, in the context of the review, we observed a similar lack of effect with the exogenous expression of Bcl11b, in agreement with the hypothesis that the exogenous expression of these single TFs is not sufficient to counteract OSKM action.

Reviewer Figure 4: Atoh8 overexpression is not sufficient to constrain reprogramming and transformation. (a) Image representing AP+ colonies at day 15 of reprogramming in control and Atoh8 OE settings, representative of four independent experiments. (b) Colony counting. Data are the mean \pm s.d. (n=4 independent experiments). Student's t-test was used, and two-sided p-values are indicated. (c) Image representing immortalized foci at day 20 of oncogenic transformation in control and Atoh8 OE settings, representative of four independent experiments. (d) Foci counting. Data are the mean \pm s.d. (n=4 independent experiments). Student's t-test was used, and two-sided p-values are indicated.

Given its role in Wnt signaling regulation, the authors should test the effect of GSK3Bi (a known enhancer of reprogramming) together with Atoh8 KD (if Atoh8 depletion activates Wnt, the authors should not see an increase with GSK3Bi treatment).

We thank R2 for this comment. We employed different approaches to assess whether Atoh8 effect is strictly due to Wnt signalling modulation. We attempted to use GSK3Bi, as proposed by R2, but we observed a strong cellular toxicity on MEFs. Therefore, we showed first that Atoh8 depletion has no additive effect on reprogramming/transformation when combined with Wnt signalling activation *via*

Sfrp1 KD (Fig. 7o). Moreover, we demonstrated that the effect of Atoh8 depletion was completely abrogated if Wnt signalling activation is blocked by recombinant Sfrp1 during reprogramming/transformation. These data are now incorporated in Fig. 7p-q. Finally, even if not requested, we also demonstrated the existence of a feedback loop linking Atoh8 and Wnt signalling, with Wnt3a treatment that activates Atoh8 expression in MEFs (Fig. 7r).

Minor points:

1. The authors should show the actual numbers throughout the Figures and not fold-change/percentage/fold ratio (for example, figure 2F, 4C).

We attempted to modify the figures but it was very difficult to present the actual numbers because of the important variability observed between experiments and MEF batches, as reported by other labs^{14,15}. We maintained the ratio but we can easily provide the detailed numbers of each experiment as raw data if requested by R2.

2. In Figure S1g, no error bars are presented for two conditions. The authors should include the missing data.

Data have been included accordingly in the Extended Data Figure 1e.

3. The cell cycle is only affected in “transfo” conditions but not in the “repro+transfo.” The author should clarify this discrepancy and show the original FACS plots.

We thank R2 for this observation. We now included the FACS plots in Extended Data Fig. 1f. We also commented on this result by enlarging the protective effect of OSKM on DNA damage and cell cycle features in the main text page 5.

4. Figure 3f does not help and can be removed, as the populations RI4 and TI4 still have MEF identity. Moreover, the axis is starting at 3.5 while it should begin at 0.

As requested, we removed Fig. 3f from the main figures.

5. In Figure S1i, S2d, S2m, 5m, 5q, 6j, the authors present statistical differences and mention “n=3” in the

figure legends; however, only two samples per condition are shown. The authors should include the missing data points.

We modified the figure legends and/or figures accordingly.

Reviewer #3:

Remarks to the Author:

Review of Huyghe et al NCB 2021

This study interrogated the early stages of cellular reprogramming and cellular transformation, to identify the molecular changes that define and drive these processes, and to examine whether loss of cellular identity is coupled to gain of plasticity. The authors developed and utilized a mouse model that enabled inducible reprogramming or transformation of mouse embryonic fibroblasts (MEFs) *ex vivo*. Reprogramming of MEFs to induced pluripotent stem cells (iPSCs) was achieved by doxycycline-inducible expression of Oct4, Sox2, Klf4 and Myc (OSKM) whereas oncogenic transformation was achieved by tamoxifen-induced activation of oncogenic K-ras (sometimes with cMyc overexpression and trp53 knockdown). Established functional readouts were used to assess each model (teratoma formation and alkaline phosphatase positive colonies by reprogrammed iPSCs, and soft agar colony

29formation/anchorage independent growth and in vivo tumor formation for transformed MEFs). Multiple molecular modalities were used to characterize early time points of each process (single cell and bulk RNA-seq, ATAC-seq). The main findings include identification of Bcl11b as a novel marker of MEF cell identity and whose downregulation (together with the previously described marker Thy1) can be used to prospectively isolate cells prone to (or already undergoing) reprogramming/transformation; use of ATAC-seq data to identify FosL1 as having divergent roles in transformation (loss blocks) and reprogramming (loss enhances); purported uncoupling of loss of cell identity and gain of plasticity; identification of shared gene expression changes that occur during both reprogramming and transformation; and identification of the Atoh8 transcription factor as a nonspecific constraint to reprogramming and transformation through its presumed role in suppressing Wnt signaling.

The evidence that Bcl11b is a faithful marker of transformation and reprogramming is well supported by the cell sorting experiments presented in Figure 2, although experimental perturbation would strengthen the functional relevance of this marker even further. Identification of Atoh8 as a relatively nonspecific constrainer of cellular plasticity is also well supported by the data, including examining Atoh8 knock-down in additional models of transformation and trans-differentiation. Overall, this study illustrates the power of examining dynamic cellular processes in a highly controlled, defined manner using time-resolved high-resolution molecular read-outs. The main conclusions of the study could be significantly strengthened by additional experiments, clarifying aspects of the experimental design, and analyzing the wealth of data in additional ways, as outlined below.

We sincerely thank R3 for his/her strong support of the work and for her/his constructive comments.

Major comments

1. The authors report the finding that acquisition of cellular plasticity precedes loss of cellular identity as a major conceptual advance. This is based on the observation that RI4/TI4 cells did not downregulate MEF identity (Figure 3f). However, in Fig 1j the MEF identity score seems to decrease pretty quickly, especially along the transformation trajectory. As the RI4/TI4 intermediates are defined by loss of both Bcl11b and Thy1 which, according to Figure 2c, would place them towards the end of the trajectories where the MEF identity has been lost, can the authors add clarity where these group 4 intermediates might lie along the trajectory? Importantly, how MEF identity is defined and what makes up the MEF identity score should be clearly stated, even if it is derived from other publications.

We sincerely apologize for this lack of accuracy. In the initial draft, the MEF identity score was defined by a list of 298 MEF genes defined by Schiebinger et al., 2019⁷. The color code that we used in Fig. 1j was confusing and gave the feeling that the identity was decreasing pretty quickly, as noticed by R3. We apologize for this inaccuracy and proceeded as follow to correct:

1-With the help of our bioinformatics department, we modified the color code to render it more accurate and to show that the MEF identity score is decreasing far more slowly than initially presented and is still maintained in some cells after 10 days of reprogramming or transformation (please see new Fig. 1k).

2-We also added subsequent representations (Pseudo-time analysis in Fig. 1l and Violin plots encompassing the final iPS and cancer cells samples in Extended Data Fig. 1n) to reinforce the idea that the MEF identity is not rapidly and homogeneously lost during both processes.

3-Moreover, in order to strengthen this important result, we reproduced the analyses with a second independent MEF identity score (defined in Neftzger et al., 2017⁶) that consists of 395 genes. Similar results were obtained with the two independent MEF scores and integrated in Extended Data Fig. 1l-m but also throughout the manuscript (Extended Data Fig. 2m and 7l).

As requested by R3, we also attempted to position the RI4 and TI4 (now called BLTL) cells on their respective single cell trajectories. The direct comparison of sc-RNA-seq data with bulk-RNA-seq is hindered by the difference in resolution between the two techniques. We circumvented this issue in 2 ways:

1-We first located the cells that downregulated both *Bcl11b* and *Thy1* on the single cell trajectories and observed that they appeared at day5 of both processes, especially in the bifurcation area that we identified (see Diffusion map in Fig. 3a and Violin plots in Extended Data Fig. 3b).

2-We used a trick by finding differentially expressed genes between each intermediate (RI4-BLTL and TI4-BLTL) and untreated MEFs in bulk-RNA-seq, and converting them into a “signature score”. These scores were next calculated for each individual cell and projected on the sc-RNA-seq diffusion map. For reprogramming and to a lesser extent transformation, we noticed the emergence of a reduced number of cells harboring a high activity of the BLTL score in the bifurcation area, suggesting that these cells might correspond to the bulk intermediates (please see Fig. 4g).

2. *Bcl11b* downregulation was identified as a marker of transformation and reprogramming at very early time points and the sub-population sorting experiment in Figure 2 suggests that *Bcl11b* downregulation is a key event in both processes. Given that a) *Bcl11b* has not been explored in the context of MEF cell identity or transformation/reprogramming, b) *Bcl11b* is a well-known regulator of T cell differentiation with similar properties (constraining cell fate) and c) the claim of uncoupling between loss of cell identity and gain of plasticity is based on *Bcl11b*-defined populations, functional perturbation of *Bcl11b* would increase the significance of this finding and these claims. The experimental system appears sufficiently established to feasibly test the impact of *Bcl11b* knock-down (and possibly rescue) on transformation and reprogramming.

We thank R3 for this inspiring comment, redundant with R2. As suggested, we conducted a large number of approaches to dissect *Bcl11b* function during reprogramming, transformation and transdifferentiation that are now presented in a fully new Figure 2:

1-We started by modulating Bcl11b expression prior to induce reprogramming or transformation. Bcl11b gain- and loss-of-function approaches in MEFs (using RNA interference but also Bcl11b conditional KO MEFs¹¹) led to demonstrate that Bcl11b depletion significantly increased reprogramming and transformation efficiencies (please see Fig. 2e-i). Even if not requested by R3, we next conducted teratoma formation assays to show that Bcl11b depletion has no major detrimental effect on the acquisition of *in vivo* multilineage differentiation potential (Fig. 2j). Therefore, we demonstrated that Bcl11b functionally acts as a gatekeeper of the somatic state.

2-Bcl11b is highly expressed in T lymphocytes. Therefore, even if not requested, and in order to broaden the impact of our finding, we evaluated Bcl11b role in the reprogramming of T lymphocytes. Using OSKM Dox-inducible primary T cells, we demonstrated that Bcl11b constrains reprogramming of this alternative somatic cell type. Data are presented in Figure 2k-l. We also showed that Bcl11b also constrains MEFs to neuron transdifferentiation triggered by Brn2, Ascl1 and Myt1l (Fig. 2m-n).

3-Next, in order to understand Bcl11b function, we combined RNA-seq and ChIP-seq analyses. We compared the transcriptomes of control, Bcl11b-overexpressing (OE) and Bcl11b-knockdown (KD) MEFs. In parallel, we optimized the Bcl11b ChIP-seq conditions by combining two commercial antibodies (Abcam, ab18465, Bethyl, A300-385A). We identified 7430 specific peaks of Bcl11b binding to the MEF genome (Fig. 2p). We found that Bcl11b modulation impacts 979 genes (adjusted p-value < 5.10⁻²; log₂ FC > 0.5 or < -0.5) with 122 genes (13%) presenting a Bcl11b binding site within 10kb of the TSS (Fig. 2s). Finally, the fact that several Bcl11b targets were related to the Mapk signalling pathway led us to demonstrate that Bcl11b directly regulates pErk1/2 levels in MEFs (Fig. 2t), providing a potential explanation of its mode of action during reprogramming¹².

Overall, we invested massive efforts to address the pertinent suggestion of R3 that led to improve significantly the impact of the manuscript by identifying Bcl11b as a new broad-range gatekeeper of somatic states from reprogramming, transdifferentiation and transformation.

3. In this study, multiple factors were identified as correlating with (Thy1, Bcl11b) or directly impacting (FosL1, Atoh8, Srfp1/2) MEF cell reprogramming and transformation, however whether and how these factors functionally interact with each other was not discussed. Moreover, how these factors interact with the factors that actually drive each process (e.g. OSKM for reprogramming and oncogenic Kras for transformation) was not explored or discussed. Although this would not necessarily impact the main conclusions of the paper, some attempt at tying these diverse factors together within the context of these central cellular processes would improve the broader relevance to the field.

We than R3 for this pertinent comment. To address this point, we conducted different approaches:

1-We focused in particular on *Atoh8* and *Bcl11b* and, by comparing the ChIP-seq profiles, we revealed the existence of a co-binding on 533 peaks on the MEF genome (see Fig. 7e). We also noticed an overlap of 173 genes that are commonly deregulated by *Bcl11b* and *Atoh8* KD.

2- We individually knockdown the expression of *Bcl11b*, *Atoh8*, *FosL1* and *Sfrp1* in MEFs and assessed the effect on the expression of the other factors. This approach showed that most of the factors do not regulate each other, at least in MEFs, except *Sfrp1* that appear to repress *FosL1* expression. Deeper investigations will be required but we believe that they are out of the scope of the present study. These data are presented in Extended Data Fig. 7s.

3-We assessed whether the expression of *Oct4* (reprogramming) and *K-Ras^{G12D}* (transformation) was impacted by *Bcl11b*, *Atoh8*, *FosL1* and *Sfrp1* KD. No significant impact was observed, ruling out the possibility that the phenotypes observed were due to a direct action of the candidates on the factors that drive the processes. These data are presented in Extended Data Fig. 7r.

4-Finally, we significantly extended the exploration of the interplay between WNT signalling and *Atoh8*. We showed in particular that *Atoh8* effect is abrogated by recombinant *Sfrp1* (see Fig. 7p-q) and we also revealed the existence of a feedback loop by which WNT signalling promotes *Atoh8* expression. These data are presented in Fig. 7r. The interplays between the actors are now presented in the model Fig. 7s.

Some of these points are easily explored with the data already generated. For example, *Atoh8* knock-down hastened activation of endogenous OSKM genes, suggesting *Atoh8* might repress these genes. However, *Atoh8* ChIP-seq showed that it predominantly binds enhancers. Does *Atoh8* bind OSKM genes? *Atoh8*, presumably suppresses Wnt signaling by upregulating Wnt inhibitors. Is there any evidence of *Atoh8* binding to these genes?

We thank R3 for this suggestion. We interrogated the ChIP-seq datasets and no direct binding of *Atoh8* was detected on OSKM, Ras and Wnt-associated genes. We believe that the fact that *Atoh8* hinders the activation of the endogenous *Oct4* gene might be due to indirect effects, but additional approaches, that appear to us out of the scope of the present study, will be required.

Minor comments and suggestions

1. Experimental details are missing in the following places

- How was the MEF identity score identified and applied to the data generated in this manuscript?

We apologize for this inaccuracy. These informations can now be found in the main text page 6 but also in the methods section on page 41.

- page 10 mentions a result showing expression change of *Dmrtc2* and *Pou3f1* but these genes are not shown in Extended Data Fig. 4i (and the gene *Shisa8* is shown in ED 4i but not listed in the results section).

We apologize for this inaccuracy and thank R3 for noticing it. We corrected the sentence accordingly on page 12.

- Page 11 mentions a “protein class analysis with pantherdb” was used to find master regulators of a transcriptional program, but unclear how this analysis was performed, and it is not mentioned in the Methods section (what genes were used as input? How were they compiled? How did this analysis lead to the subsequent focus on the genes highlighted in Cluster 2 of fig. 3k?)

We corrected the revised draft on page 12 by removing the sentence.

- Figure 6i: how long after *Atoh8* knockdown were cells taken for RNA-seq? This is important for interpreting that the MEF identity score did not change. In many figures, the time point of the experiment was difficult to track down, so adding the timepoint to the figure itself or including in the figure legend each time would be helpful.

We apologize for this inaccuracy and added time points in a large number of panels throughout the figures and/or in the figure legends. The consequences of *Atoh8* depletion were analyzed 5 days after the infection with the lentiviral particles inducing the KD. Even if we cannot rule out that this timing was too short to induce changes of MEF identity, it was sufficient to induce severe transcriptomic changes, suggesting that at least part of the GRN controlled by *Atoh8* is impacted at this timing. However, because we believe that this comment is important, we modified the text accordingly claiming that “at this time point, *Atoh8* depletion has no significant impact on MEF identity...” on page 16.

2. Based on the wide-spread use of the term ‘cellular plasticity’ it would be good to clearly define this term within the context of the read-outs presented in this study.

We now included a definition of the term cell plasticity and cited relevant literature in the Introduction page 3.

3. Adding *Bcl11B*/*Thy1* expression (high/low) to figure 2n would aid interpretation.

We fully agree with R3 that the previous nomenclature was difficult to track. In order to gain clarity, we modified the nomenclature for the reprogramming and transforming intermediates throughout the manuscript. The nomenclature now includes the Bcl11b/Thy1 expression levels with initials (BHTH for Bcl11b^{high}/Thy1^{high}) in the new figure 3n.

4. A suggestion is to analyze the joint Bcl11B/Thy1 expression in the single cell data from Figure 1 as support for the transition kinetics inferred in Figure 2 (how does the frequency of Bcl11b/Thy1 high/low cells change along each trajectory? What about the pattern in the bifurcation area, which is relatively under-discussed in this manuscript?)

We thank R3 for this pertinent suggestion. We analyzed the joint expression of *Bcl11b* and *Thy1* in the sc-RNA-seq data to visualize the transition inferred in Fig. 2. As shown in Fig. 3a and Extended Data Fig. 3b, we visualized and quantified the emergence of Bcl11b^{low}/Thy1^{low} cells along each trajectory. We also discussed in more detail the existence and significance of this area in the discussion page 18.

5. An additional suggestion would be to identify genes that significantly correlate with Bcl11b expression during reprogramming and transformation to identify other genes putatively regulated by or regulating Bcl11b expression during these processes. Are they similar in the repro/transfo context? Particularly given the GO terms of the common set in Figure 3h, it is likely worthwhile to describe a putative Bcl11b controlled gene expression program in each context.

As presented in the new Figure 2, we deciphered the GRN controlled by Bcl11b by conducting RNA-seq in gain- and loss-of-function settings with ChIP-seq analyses. For this specific question, we compared the genes responding to Bcl11b KD in MEFs with the genes that significantly correlate with Bcl11b during reprogramming and transformation. This analysis led to reveal a significant overlap, in agreement with the fact that these intermediates lost Bcl11b expression. These data are presented in Figure 4f.

6. Unsupervised hierarchical clustering of the ATAC-seq data to identify clusters in an unsupervised manner is another way to identify potentially interesting clusters and see how it compares to the supervised classification of C1-C6 in Fig. 3c,d

We included a heatmap visualizing relative peak intensities in the data (see Extended Data Fig. 4b) for differential loci. It allows indeed to visualize the high-dimensional structure of the data as well as our cluster group assignments. However, in this case, the data appear a bit too complex to do a single hierarchical clustering to identify peak groups, potentially explaining why the clusters don't just show up as neat blocks due to the high dimensionality.

7. Extended data fig. 4i should also show Dmrtc2 and Pou3f1 but these are missing. Shisa8 is shown but isn't listed in the main text (p.10)

We apologize for this inaccuracy and thank R3 for noticing it. We corrected the sentence accordingly on page 12.

8. Fig. 3a: add the % variance explained by each component

We corrected this part and added the % variance in figure 4a.

9. Why is p53 knock-down used in the experiments shown in Figure 4, but not in previous experiments of the same experimental system?

We included p53 KD to accelerate the transformation process in several experiments but we obtained similar results with or without p53.

10. The significance of the gene expression results presented in Figure 4f-j is unclear; gene expression differences can be seen, but how they support the conclusion that “Atoh8 constrains cellular plasticity” is not clear. The functional read outs on the other hand are very convincing.

We agree with R3. We modified the main text in order to focus more specifically on the adhesion properties of the cells that are modulated by Atoh8, especially *via* Cdh1. We also modified the title of the paragraph accordingly for “Atoh8 regulates the acquisition of malignant features” page 13.

11. Also figure 4: the colony assay and western blots show that Atoh8 KD has an impact at early time points (d.3 and d.6) but analysis of gene expression was performed at day >30, so it's unclear if the time points directly impacted by Atoh8 are missed (e.g. the results in Figs4f,h,I,j are simply reflecting emergence of different cell states/types that are consequential to Atoh8 knock-down).

We understand R3 comment and separated the 2 biological questions in the revised draft page 13. We focused first on the “early” effects of Atoh8 depletion on the efficiency and pace of oncogenic transformation. In a second paragraph, we asked a different question, namely whether “Atoh8 depletion leads to the emergence of different cell states”. We also removed the kinetic assays showing the early induction of cdh1 because they were potentially confusing for the overall message.

12. Figure 4j. References for Ube2c and Top2a as “promoting cancer cell invasion and migration”?

We added a reference for Ube2c on page 14.

13. In the discussion, it is stated that the authors identified “the existence of a molecular program that commonly emerges during both processes” but this program remains obscure after reading the manuscript. Is this program all of the genes mentioned (Thy1, Bcl11b, Atoh8, Wnt regulators, etc.?). Is this program the “bifurcation area” shown in Figure 1k?

We apologize for this lack of clarity. The discussion was modified accordingly page 18.

14. Page 6: “let us” should be “led us”

Sentence has been modified accordingly.

15. Page 7: “reflect the ability” should be “reflects the ability”

Sentence has been corrected.

16. Fig4 legend: “Cells were splited” should be “Cells were split” or “Cells were passaged at least 10 times”

Sentence has been corrected accordingly.

References:

- 1 Wernig, M. *et al.* A drug-inducible transgenic system for direct reprogramming of multiple somatic cell types. *Nat Biotechnol* **26**, 916-924, doi:10.1038/nbt1483 (2008).
- 2 Buganim, Y. *et al.* Single-cell expression analyses during cellular reprogramming reveal an early stochastic and a late hierarchic phase. *Cell* **150**, 1209-1222, doi:10.1016/j.cell.2012.08.023 (2012).
- 3 Takahashi, K. & Yamanaka, S. Induction of pluripotent stem cells from mouse embryonic and adult fibroblast cultures by defined factors. *Cell* **126**, 663-676, doi:10.1016/j.cell.2006.07.024 (2006).
- 4 dos Santos, R. L. *et al.* MBD3/NuRD facilitates induction of pluripotency in a context-dependent manner. *Cell Stem Cell* **15**, 102-110, doi:10.1016/j.stem.2014.04.019 (2014).
- 5 Mikkelsen, T. S. *et al.* Dissecting direct reprogramming through integrative genomic analysis. *Nature* **454**, 49-55, doi:10.1038/nature07056 (2008).
- 6 Hussein, S. M. *et al.* Genome-wide characterization of the routes to pluripotency. *Nature* **516**, 198-206, doi:10.1038/nature14046 (2014).
- 7 Chronis, C. *et al.* Cooperative Binding of Transcription Factors Orchestrates Reprogramming. *Cell* **168**, 442-459 e420, doi:10.1016/j.cell.2016.12.016 (2017).
- 8 Polo, J. M. *et al.* A molecular roadmap of reprogramming somatic cells into iPS cells. *Cell* **151**, 1617-1632, doi:10.1016/j.cell.2012.11.039 (2012).
- 9 Nefzger, C. M. *et al.* Cell Type of Origin Dictates the Route to Pluripotency. *Cell Rep* **21**, 2649-2660, doi:10.1016/j.celrep.2017.11.029 (2017).
- 10 Schiebinger, G. *et al.* Optimal-Transport Analysis of Single-Cell Gene Expression Identifies Developmental Trajectories in Reprogramming. *Cell* **176**, 928-943 e922, doi:10.1016/j.cell.2019.01.006 (2019).
- 11 Li, P. *et al.* Reprogramming of T cells to natural killer-like cells upon Bcl11b deletion. *Science* **329**, 85-89, doi:10.1126/science.1188063 (2010).
- 12 Prieto, J. *et al.* Early ERK1/2 activation promotes DRP1-dependent mitochondrial fission necessary for cell reprogramming. *Nat Commun* **7**, 11124, doi:10.1038/ncomms11124 (2016).
- 13 Stadtfeld, M., Maherali, N., Breault, D. T. & Hochedlinger, K. Defining molecular cornerstones during fibroblast to iPS cell reprogramming in mouse. *Cell Stem Cell* **2**, 230-240, doi:10.1016/j.stem.2008.02.001 (2008).
- 14 Mahmoudi, S. *et al.* Heterogeneity in old fibroblasts is linked to variability in reprogramming and wound healing. *Nature* **574**, 553-558, doi:10.1038/s41586-019-1658-5 (2019).
- 15 Kida, Y. S. *et al.* ERRs Mediate a Metabolic Switch Required for Somatic Cell Reprogramming to Pluripotency. *Cell Stem Cell* **16**, 547-555, doi:10.1016/j.stem.2015.03.001 (2015).

Decision Letter, first revision:

7th April 2022

Dear Fabrice,

Thank you for submitting your revised manuscript "The comparative roadmaps of reprogramming and transformation unveiled that cellular plasticity is broadly controlled by Bcl11b and Atoh8" (NCB-L46074A). It has now been seen by the original referees and their comments are below. The reviewers find that the paper has improved in revision, and therefore we'll be happy in principle to publish it in Nature Cell Biology, pending minor revisions to satisfy the referees' final requests and to comply with our editorial and formatting guidelines.

Please, note that we wish you to address ALL the remaining referee points, EXCEPT for point 2 of referee 1, asking you to address the commonality of the initial events of reprogramming and transformation (which currently mostly relies on the gene expression profiles and the roles of few regulators) that -although interesting and potentially insightful- was not raised in the initial round of review and we do not consider necessary for publication of your article at Nature Cell Biology.

Thank you again for your interest in Nature Cell Biology Please do not hesitate to contact me if you have any questions.

Best wishes,
Stelios

Stylios Lefkopoulos, PhD
He/him/his
Associate Editor
Nature Cell Biology
Springer Nature
Heidelberger Platz 3, 14197 Berlin, Germany

E-mail: stylios.lefkopoulos@springernature.com
Twitter: @s_lefkopoulos

Reviewer #1 (Remarks to the Author):

39In this revised manuscript, the authors addressed the points raised by this reviewer very carefully. Reprogramming and transformation events without Myc were analyzed and ruled out the possibility that the role of the Myc-Atoh8-Sfrp1 axis does not depend to the exogenous Myc. In addition, the controversy for the role of Atoh8 is also carefully dissected and they revealed that Atoh8 function is inhibitory for reprogramming/transformation of MEF. These revisions wipe out my concern, so now I agree with the publication of this manuscript in Nat Cell Biol after few minor revisions.

1. Although the authors showed that Atoh8 expression is weak in adult ear fibroblasts and T lymphocytes (Ext Fig 6), their statement in discussion part still sounds the role of Atoh8 as a broad range regulator. Clear statement for the differential roles of Bcl11 as a broad range regulator and Atoh8 as a cell-type-specific (MEF-specific) regulator will be required.
2. Commonality of the initial events of reprogramming and transformation is only based on the gene expression profiles and the roles of few regulators. It will be nice if the authors show it in the functional way. What will happen if the initial state triggered by oncogenic genes is followed by the activation of the reprogramming factors (and vice versa)?

Reviewer #2 (Remarks to the Author):

The authors have adequately addressed my comments. I would still suggest including the ATOH8 overexpression data in the manuscript (now presented as Rev Figure 4). Overall, the findings presented in this study represent a significant conceptual advance in the reprogramming/cancer fields and warrant publication in NCB.

Reviewer #3 (Remarks to the Author):

The authors have sufficiently addressed the main concerns through additional functional experiments, most important of which was to demonstrate that Bcl11b knock-down enhances reprogramming and transformation which supports a main conclusion of the original draft. It is interesting that forced expression of Bcl11b did not block reprogramming, suggesting that the exogenous OSKM factors can overcome Bcl11b in that context. There are many interesting follow-up question that this study raises but they are outside of the scope of the current work.

One outstanding comment: Was Bcl11b chip-seq was done only on unmodified MEFs, or also in Bcl11b overexpression or knock down? It is not clear from the Figure 2 legend/methods.

21st April 2022

Dear Dr. Laval,

40Thank you for your patience as we've prepared the guidelines for final submission of your Nature Cell Biology manuscript, "The comparative roadmaps of reprogramming and transformation unveiled that cellular plasticity is broadly controlled by Bcl11b and Atoh8" (NCB-L46074A). Please carefully follow the step-by-step instructions provided in the attached file, and add a response in each row of the table to indicate the changes that you have made. Please also check and comment on any additional marked-up edits we have proposed within the text. Ensuring that each point is addressed will help to ensure that your revised manuscript can be swiftly handed over to our production team.

We would like to start working on your revised paper, with all of the requested files and forms, as soon as possible (preferably within one week). Please get in contact with us if you anticipate delays.

In recognition of the time and expertise our reviewers provide to Nature Cell Biology's editorial process, we would like to formally acknowledge their contribution to the external peer review of your manuscript entitled "The comparative roadmaps of reprogramming and transformation unveiled that cellular plasticity is broadly controlled by Bcl11b and Atoh8". For those reviewers who give their assent, we will be publishing their names alongside the published article.

Nature Cell Biology offers a Transparent Peer Review option for new original research manuscripts submitted after December 1st, 2019. As part of this initiative, we encourage our authors to support increased transparency into the peer review process by agreeing to have the reviewer comments, author rebuttal letters, and editorial decision letters published as a Supplementary item. When you submit your final files please clearly state in your cover letter whether or not you would like to participate in this initiative. Please note that failure to state your preference will result in delays in accepting your manuscript for publication.

Cover suggestions

As you prepare your final files we encourage you to consider whether you have any images or illustrations that may be appropriate for use on the cover of Nature Cell Biology.

41If your image is selected, we may also use it on the journal website as a banner image, and may need to make artistic alterations to fit our journal style.

Nature Cell Biology has now transitioned to a unified Rights Collection system which will allow our Author Services team to quickly and easily collect the rights and permissions required to publish your work. Approximately 10 days after your paper is formally accepted, you will receive an email in providing you with a link to complete the grant of rights. If your paper is eligible for Open Access, our Author Services team will also be in touch regarding any additional information that may be required to arrange payment for your article.

Please note that *Nature Cell Biology* is a Transformative Journal (TJ). Authors may publish their research with us through the traditional subscription access route or make their paper immediately open access through payment of an article-processing charge (APC). Authors will not be required to make a final decision about access to their article until it has been accepted. Find out more about Transformative Journals

Please use the following link for uploading these materials:
[REDACTED]

Best regards,

42Nyx Hills
Staff
Nature Cell Biology

On behalf of

Stylios Lefkopoulos, PhD
He/him/his
Associate Editor
Nature Cell Biology
Springer Nature
Heidelberger Platz 3, 14197 Berlin, Germany

E-mail: stylios.lefkopoulos@springernature.com
Twitter: @s_lefkopoulos

Reviewer #1:

Remarks to the Author:

In this revised manuscript, the authors addressed the points raised by this reviewer very carefully. Reprogramming and transformation events without Myc were analyzed and ruled out the possibility that the role of the Myc-Atoh8-Sfrp1 axis does not depend to the exogenous Myc. In addition, the controversy for the role of Atoh8 is also carefully dissected and they revealed that Atoh8 function is inhibitory for reprogramming/transformation of MEF. These revisions wipe out my concern, so now I agree with the publication of this manuscript in Nat Cell Biol after few minor revisions.

1. Although the authors showed that Atoh8 expression is weak in adult ear fibroblasts and T lymphocytes (Ext Fig 6), their statement in discussion part still sounds the role of Atoh8 as a broad range regulator. Clear statement for the differential roles of Bcl11 as a broad range regulator and Atoh8 as a cell-type-specific (MEF-specific) regulator will be required.
2. Commonality of the initial events of reprogramming and transformation is only based on the gene expression profiles and the roles of few regulators. It will be nice if the authors show it in the functional way. What will happen if the initial state triggered by oncogenic genes is followed by the activation of the reprogramming factors (and vice versa)?

Reviewer #2:

Remarks to the Author:

The authors have adequately addressed my comments. I would still suggest including the ATOH8 overexpression data in the manuscript (now presented as Rev Figure 4).

Overall, the findings presented in this study represent a significant conceptual advance in the reprogramming/cancer fields and warrant publication in NCB.

43Reviewer #3:

Remarks to the Author:

The authors have sufficiently addressed the main concerns through additional functional experiments, most important of which was to demonstrate that Bcl11b knock-down enhances reprogramming and transformation which supports a main conclusion of the original draft. It is interesting that forced expression of Bcl11b did not block reprogramming, suggesting that the exogenous OSKM factors can overcome Bcl11b in that context. There are many interesting follow-up question that this study raises but they are outside of the scope of the current work.

One outstanding comment: Was Bcl11b chip-seq was done only on unmodified MEFs, or also in Bcl11b overexpression or knock down? It is not clear from the Figure 2 legend/methods.

Author Rebuttal, first revision:

Reviewer #1:

Remarks to the Author:

In this revised manuscript, the authors addressed the points raised by this reviewer very carefully. Reprogramming and transformation events without Myc were analyzed and ruled out the possibility that the role of the Myc-Atoh8-Sfrp1 axis does not depend to the exogenous Myc. In addition, the controversy for the role of Atoh8 is also carefully dissected and they revealed that Atoh8 function is inhibitory for reprogramming/transformation of MEF. These revisions wipe out my concern, so now I agree with the publication of this manuscript in Nat Cell Biol after few minor revisions.

We sincerely thank R1 for his/her comment on the revised manuscript and we are delighted that he got convinced by our additional efforts.

1. Although the authors showed that Atoh8 expression is weak in adult ear fibroblasts and T lymphocytes (Ext Fig 6), their statement in discussion part still sounds the role of Atoh8 as a broad range regulator. Clear statement for the differential roles of Bcl11 as a broad range regulator and Atoh8 as a cell-type-specific (MEF-specific) regulator will be required.

We thank R1 and corrected the text accordingly on page 14.

2. Commonality of the initial events of reprogramming and transformation is only based on the gene expression profiles and the roles of few regulators. It will be nice if the authors show it in the functional

way. What will happen if the initial state triggered by oncogenic genes is followed by the activation of the reprogramming factors (and vice versa)?

We did not raise this point that -although interesting and potentially insightful- was not raised in the initial round of review and was not considered necessary for publication by Nature Cell Biology editor.

Reviewer #2:

Remarks to the Author:

The authors have adequately addressed my comments. I would still suggest including the ATOH8 overexpression data in the manuscript (now presented as Rev Figure 4).

Data are now presented in Extended Data Figure 6.

Overall, the findings presented in this study represent a significant conceptual advance in the reprogramming/cancer fields and warrant publication in NCB.

We sincerely thank R2 for his/her support of the manuscript as well as for his/her pertinent comments that improved the quality of our work.

Reviewer #3:

Remarks to the Author:

The authors have sufficiently addressed the main concerns through additional functional experiments, most important of which was to demonstrate that Bcl11b knock-down enhances reprogramming and transformation which supports a main conclusion of the original draft. It is interesting that forced expression of Bcl11b did not block reprogramming, suggesting that the exogenous OSKM factors can overcome Bcl11b in that context. There are many interesting follow-up question that this study raises but they are outside of the scope of the current work.

We warmly thank R3 for his/her constant support of the manuscript.

One outstanding comment: Was Bcl11b chip-seq was done only on unmodified MEFs, or also in Bcl11b overexpression or knock down? It is not clear from the Figure 2 legend/methods.

We added this information in Figure 2 legend.

Final Decision Letter:

Dear Fabrice,

I am pleased to inform you that your manuscript, "Comparative roadmaps of reprogramming and oncogenic transformation identify Bcl11b and Atoh8 as broad regulators of cellular plasticity.", has now been accepted for publication in Nature Cell Biology. Congratulations to you and your team!

Please note that *Nature Cell Biology* is a Transformative Journal (TJ). Authors may publish their research with us through the traditional subscription access route or make their paper immediately open access through payment of an article-processing charge (APC). Authors will not be required to

46make a final decision about access to their article until it has been accepted. Find out more about Transformative Journals

If you have not already done so, we strongly recommend that you upload the step-by-step protocols used in this manuscript to the Protocol Exchange (www.nature.com/protocolexchange), an open online resource established by Nature Protocols that allows researchers to share their detailed experimental know-how. All uploaded protocols are made freely available, assigned DOIs for ease of citation and are fully searchable through nature.com. Protocols and Nature Portfolio journal papers in which they are used can be linked to one another, and this link is clearly and prominently visible in the online versions of both papers. Authors who performed the specific experiments can act as primary authors for the Protocol as they will be best placed to share the methodology details, but the Corresponding Author of the present research paper should be included as one of the authors. By uploading your Protocols to Protocol Exchange, you are enabling researchers to more readily reproduce or adapt the methodology you use, as well as increasing the visibility of your protocols and papers. You can also establish a dedicated page to collect your lab Protocols. Further information can be found at www.nature.com/protocolexchange/about

With kind regards,
Stelios

47Stylianos Lefkopoulos, PhD
He/him/his
Associate Editor
Nature Cell Biology
Springer Nature
Heidelberger Platz 3, 14197 Berlin, Germany

E-mail: stylianos.lefkopoulos@springernature.com
Twitter: @s_lefkopoulos

** Visit the Springer Nature Editorial and Publishing website at www.springernature.com/editorial-and-publishing-jobs for more information about our career opportunities. If you have any questions please click here.**